# MMSci: A Dataset for Graduate-Level Multi-Discipline Multimodal Scientific Understanding

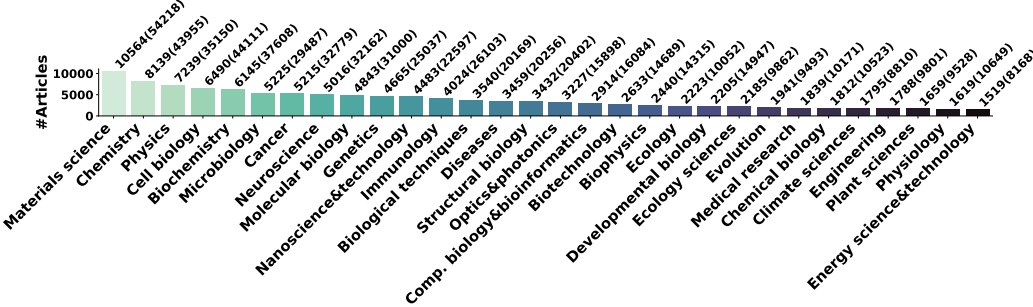

Figure 1: The top 30 out of **72 science subjects**[1] with the most articles in our dataset MMSci. The corresponding numbers of figures are shown in brackets.

## Abstract

The rapid development of Multimodal Large Language Models (MLLMs) is making AI-driven scientific assistants increasingly feasible, with interpreting scientific figures being a crucial task. However, existing datasets and benchmarks focus mainly on basic charts and limited science subjects, lacking comprehensive evaluations. To address this, we curated a multimodal, multidisciplinary dataset from peer-reviewed, open-access Nature Communications articles, spanning 72 scientific disciplines. This dataset includes figures such as schematic diagrams, simulated images, macroscopic/microscopic photos, and experimental visualizations (e.g., western blots), which often require graduate-level, discipline-specific expertise to interpret. We developed benchmarks for scientific figure captioning and multiple-choice questions, evaluating six proprietary and over ten open-source models across varied settings. The results highlight the high difficulty of these tasks and the significant performance gap among models. While many open-source models performed at chance level on the multiple-choice task, some matched the performance of proprietary models. However, the gap was more pronounced in the captioning task. Our dataset also provide valuable resource for training. Fine-tuning the Qwen2-VL-2B model with our task-specific multimodal training data improved its multiple-choice accuracy to a level comparable to GPT-4o, though captioning remains challenging. Continuous pre-training of MLLMs using our interleaved article and figure data enhanced their material generation capabilities, demonstrating potential for integrating scientific knowledge. The dataset and benchmarks will be released to support further research.

## 1 Introduction

Recent advancements in Multimodal Large Language Models (MLLMs) (Li et al., 2023; Zhu et al., 2023; Liu et al., 2024; Chen et al., 2024b; Bai et al., 2023b; Achiam et al., 2023; Team et al.,

---

[1] https://www.nature.com/nature/browse-subjects

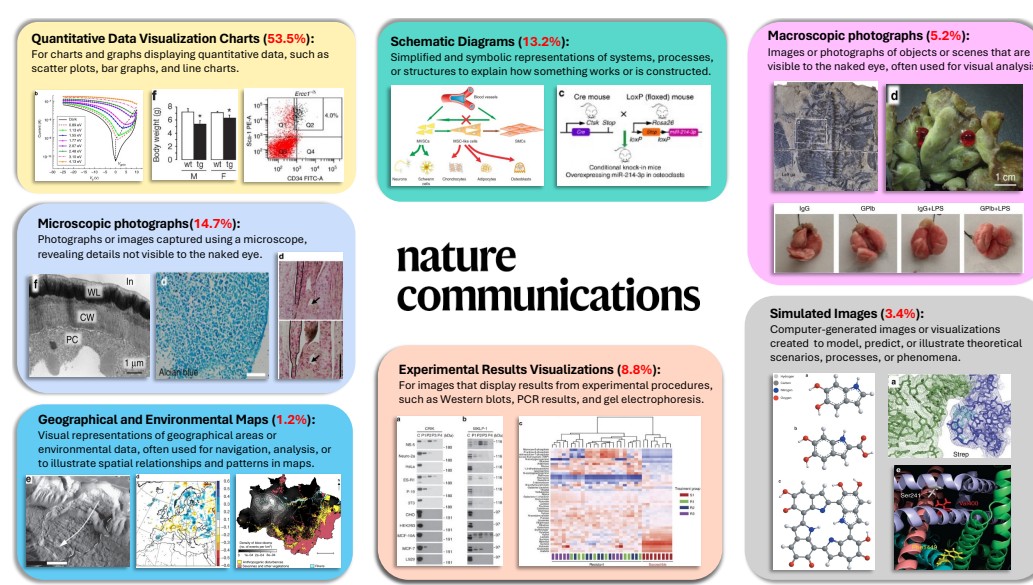

Figure 2: **Examples of the heterogeneous types of scientific figures in MMSCI**, collected from open-access, peer-reviewed articles in Nature Communications.

2023; Anthropic, 2024a; Wang et al., 2024a), have demonstrated remarkable capabilities in solving problems involving visual context. These range from everyday scenes (Antol et al., 2015), to reading documents (Tito et al., 2021), understanding diagrams (Kembhavi et al., 2016), interpreting charts (Kahou et al., 2017; Masry et al., 2022), solving math (Lu et al., 2023), and multi-discipline problems (Yue et al., 2023; 2024). The growing capabilities and intelligence of MLLMs make them promising as AI-driven scientific assistants capable of solving problems and assisting in research in various *scientific domains*. A critical aspect of this assistance is interpreting the figures in research articles, which often contain and convey rich, compressed, and complex information, requiring expert knowledge in specific scientific fields for proper understanding.

However, the ability of current MLLMs to comprehend diverse complex scientific figures across diverse disciplines remains largely unexplored. Existing research has primarily focused on chart interpretation (Kahou et al., 2017; Masry et al., 2022; Roberts et al., 2024; Wang et al., 2024b), which requires minimal domain-specific knowledge. In contrast, figures in scientific articles are far more varied, including microscopy and spectroscopy images, astronomical images, maps, 3D models, molecular structures, geological models, phylogenetic trees, electropherograms, waveforms, heatmaps, spectrograms, etc. Interpreting these figures often requires expert, typically graduate-level, knowledge in specific domains. Furthermore, datasets addressing figures beyond charts tend to focus on a limited range of subjects, lacking coverage of the full spectrum of scientific fields Li et al. (2024).

To bridge the gap, we collected a **multimodal, multi-discipline dataset MMSCI** from high-quality, open-access articles published in *Nature Communications*[2], which are freely and permanently available upon publication under a Creative Commons Attribution 4.0 International (CC BY) license[3]. This dataset spans 72 scientific disciplines, primarily within the natural sciences (the top 30 subjects with most articles can be seen in Figure 1). It includes over 131k articles and 742k figures, featuring a wide range of figures types in these research articles (seven major image types are shown in Figure 2). These figures often require graduate-level, domain-specific expertise to interpret, going beyond basic chart comprehension. To evaluate MLLMs' understanding of complex multimodal scientific figures, we developed a benchmark with tasks including scientific figure captioning and multiple-choice questions, offering a thorough assessment of models' ability to analyze diverse and advanced scientific figures and content.

Our evaluation revealed challenges across tasks and notable performance gaps among current MLLMs. For the multiple-choice questions, many open-source models performed no better than random

---

[2]https://www.nature.com/ncomms/

[3]More details can be found at https://www.nature.com/ncomms/open-access

Table 1: **Comparison with prior scientific figure understanding benchmark datasets.**

| Benchmark Dataset | Data Source | Peer-reviewed | # Subjects | Image Type | Annotations |
|---|---|---|---|---|---|
| FigureQA (Kahou et al., 2017) | Synthetic Data | N/A | N/A | Charts | Synthetic |
| DvQA (Kafle et al., 2018) | Synthetic Data | N/A | N/A | Charts | Synthetic |
| SciCap (Yang et al., 2023) | CS Arxiv Papers | ✗ | 1 (CS) | Charts | Authentic |
| SciFiBench (Roberts et al., 2024) | CS Arxiv Papers | ✗ | 1 (CS) | Charts | Authentic |
| CharXiv (Wang et al., 2024b) | Arxiv Papers | ✗ | 8 | Charts | Human-picked |
| ArxivCap/QA (Li et al., 2024) | Arxiv Papers | ✗ | 32 | Open Category | Authentic/Synthetic |
| **MMSCI (Ours)** | Nature Communications | ✓ | **72** | Open Category | Authentic |

guessing. However, some models, such as Qwen2-VL-7B (Wang et al., 2024a) and MiniCPM-V-2.6 Yao et al. (2024), showed strong performance, comparable to some proprietary models like Gemini-1.5-Flash (Reid et al., 2024), and Claude-3-Opus (Anthropic, 2024a). GPT-4o (Achiam et al., 2023) and Claude-3.5-Sonnet (Anthropic, 2024b) are the leading models, outperformed even computer science master's students, underscoring their potential as scientific assistants. This also highlights the task's difficulty and the demand for domain-specific knowledge. All models struggle to generate precise captions, particularly with nuanced semantics. However, GPT-4V and GPT-4o perform significantly better than other models when grounded on article abstracts.

Additionally, our dataset provides a vast collection of high-quality research articles and figures across diverse subjects, which can be leveraged as training resources to enhance MLLMs' understanding of multimodal scientific content. We experimented with constructing visual supervised fine-tuning data, including the task-specific data converted into single- and multi-turn conversations. This data significantly improved the Qwen2-VL-2B model (Wang et al., 2024a), achieving the highest overall multiple-choice accuracy on our benchmark, though improving captioning performance remained challenging. Furthermore, we pre-trained MLLMs on interleaved article text and figure images, which led to improved performance in material generation, a downstream task in material sciences.

Overall, our contributions are threefold: (1) ***Data diversity, scope and quality***: Our dataset is uniquely composed of high-quality, peer-reviewed academic articles covering **72** diverse scientific disciplines, featuring a wide range of figure types beyond charts. (2) ***Challenging benchmark***: Our benchmark includes tasks with diverse settings to ensure a comprehensive assessment. The evaluation highlights the challenges of the task and the limitations of current MLLMs in effectively interpreting figures from scientific literature. (3) ***Rich training resources***: Our dataset provides a valuable training resource. We created task-specific multimodal fine-tuning data and interleaved article and figure data for continuous MLLM pre-training. Our findings highlight the potential of this dataset to improve models' comprehension of scientific knowledge.

## 2 RELATED DATASET WORK

**Scientific Figure Understanding** Scientific figures in academic articles convey rich, valuable information, and there has been extensive research on evaluating the understanding of these figures. As seen in Table 1, existing datasets primarily focus on relatively simple chart figures, which require minimal scientific knowledge but general chart interpretation capacities. Early efforts targeted data visualization figures, such as synthetic datasets of plots and charts (Chen et al., 2020; Kahou et al., 2017; Kafle et al., 2018). To capture more diverse and complex chart figures, FigureSeer (Siegel et al., 2016) and SciCap (Yang et al., 2023) extracted figures from computer science (CS) papers on arXiv. SciFiBench (Roberts et al., 2024) expanded on SciCap's chart figures by introducing figure-to-caption and caption-to-figure matching tasks, while CharXiv (Wang et al., 2024b) hand-picked chart figures from arXiv papers. These datasets focus exclusively on chart figures. ArxivQA/Cap (Li et al., 2024) extended the scope by collecting papers from 32 subjects on arXiv, including open-category image types beyond charts. However, the collection still heavily focuses on CS and math, with less comprehensive coverage of the natural sciences. Moreover, since arXiv papers are not peer-reviewed, their quality is not guaranteed. In contrast, our dataset includes peer-reviewed articles from Nature Communications, spanning 72 disciplines and covering a wide range of natural science subjects. The figures are highly diverse and typically require graduate-level expertise in specific subjects to interpret, ensuring the dataset's quality, diversity, and complexity.

**Multimodal Science Problems** With the advances of MLLMs, many studies have focused on evaluating their ability to solve scientific problems involving visual context. However, these datasets

Table 2: **The key statistics of MMSCI**, including the source data and the constructed benchmark test/validation (dev) set and the data for visual fine-tuning in the training set.

| Source dataset | Number | Benchmark test/dev set | Number | Training set | Number |
|---|---|---|---|---|---|
| Total subjects | 72 | Used articles | 1,418/1,414 | Used articles | 128,561 |
| Total articles | 131,393 | Figure Captioning | 1,218 /1,412 | Figure Captioning | 725,646 |
| Total figures | 742,273 | Fig2Cap Matching | 1,188/1,297 | Fig2Cap Matching | 84,328 |
| Avg. caption length | 153 | SubFig2Cap Matching | 1,119/1,214 | SubFig2Cap Matching | 53,882 |
| Avg. figures per article | 5.65 | SubCap2Fig Matching | 1,114/1,221 | SubCap2Fig Matching | 107,098 |
| Avg. abstract length | 150 | | | Multi-turn conversation | 108,843 |
| Avg. article length | 7,457 | | | Total samples | 1,079,797 |

typically emphasize models' ability to "read" and "see" simple image content for use in solving the problem, rather than testing their "understanding" of complex scientific figures. The images in these datasets are relatively simple and usually do not require expert scientific knowledge for interpretation. For example, ScienceQA (Lu et al., 2022) focuses on K1-12 level problems, while SciBench (Wang et al., 2023) is limited to three disciplines: physics, chemistry, and mathematics. MMMU (Yue et al., 2023) covers subjects such as art, business, history, health, humanities, and technology, but its coverage of natural science subjects is limited, and understanding images is not the primary challenge. Our work, in contrast, with the focus on the understanding of complex scientific figures that require graduate-level, domain-specific knowledge across scientific disciplines. It can also be potentially used for constructing multimodal science problems, which we leave for future exploration.

## 3 DATA CURATION

**Source Data Collection**   Our dataset was collected from the Nature Communications website, comprising open-access, peer-reviewed papers across five major categories and 72 subjects. The top 30 subjects are shown in Figure 1, with the full list of all 72 subjects provided in the Appendix, Table 6. Various information regarding each article is easily accessible on this website, providing a user-friendly platform for obtaining all necessary data. For each article, we collected information including the title, abstract, main body content, and references, directly from their respective sections on the article's webpage (e.g., `https://www.nature.com/articles/xxx`, where "xxx" is the article's unique ID). Figures and their captions were obtained from a dedicated figures page under the article's homepage (e.g., `https://www.nature.com/articles/xxx/figures`), eliminating the need to extract figures from PDF files and thus ensuring image quality. We used `pylatexenc` to convert LaTeX expressions of mathematical formulas in the article text and figure captions into plain text.[4] Since these papers are all peer-reviewed and the text, figures, and captions are readily available from the website, ensuring the data is both authentic and high-quality. We thus did not perform additional filtering or content extraction. We crawled articles up to the date of 2024/04/15. The resulting source dataset comprises 131,393 articles and 742,273 figures. More statistics are shown in Table 2.

**Sub-caption Extraction**   Many figures in the dataset consist of multiple sub-figures in a single image, with captions that include a main caption and descriptions of each sub-figure (sub-caption), as illustrated in Figure 3. We developed a regular expression matching function to identify sub-figure indices at the beginning of sentences in alphabetical order (a to z), extracting and identifying 514,054 sub-captions/figures, which aids in the consecutive construction of our benchmark.

**Heterogeneous Figure Types in MMSCI**   We categorized the types of (sub-)figures in MMSCI into seven major categories based on a subset of the figures, focusing on the smallest individual components, such as sub-figures when present. Following this manual review, we used GPT-4o to classify the images within the benchmark test set (see benchmark data splits in the next section). Examples of image types are shown in Figure 2, with detailed statistics provided in the Appendix (Section A.1.3). In addition to charts in previous benchmarks, which make up half of the figures, we identified six other major types that vary significantly across different subjects.

---

[4]`https://github.com/phfaist/pylatexenc`

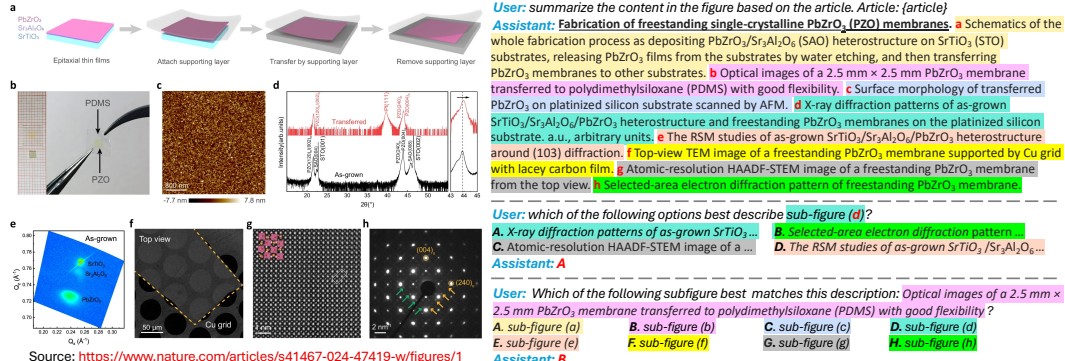

Figure 3: **Illustration of the benchmark data in MMSCI.** This example is taken from (Guo et al., 2024). The left side shows a figure with multiple sub-figures, accompanied by a main caption (bolded) and sub-captions (highlighted in different colors), each corresponding to a sub-figure. These sub-figures and sub-captions are used to construct tasks for figure captioning (upper right), sub-figure to sub-caption matching (center right), and sub-caption to sub-figure matching (lower right).

## 4 BENCHMARKS

We developed two benchmark tasks MMSCICAP and MMSCIQA with varying settings to comprehensively test models' understanding of scientific figures and content, as shown in Figure 3.

**MMSCICAP: Scientific Figure Captioning** Scientific figure captioning in MMSCI presents unique challenges compared to typical natural image captioning. Unlike natural images, interpreting figures in Nature Communications articles often requires graduate-level expertise in specific subjects and grounding in the article's content. Additionally, these captions are significantly more detailed. In MMSCI, captions average 153 words, much longer than those for natural images and ArxivCap (Li et al., 2024). This complexity makes scientific figure captioning in our benchmark more demanding. In our task, we evaluate the scientific figure captioning with two settings: (1) ***ungrounded figure captioning***: The model generates captions without access to any additional article content. (2) ***abstract-grounded figure captioning***: The model is provided with the paper abstract to offer context for the figure. For evaluation metrics, we consider overlap-based metrics ROUGE (Lin, 2004), METEOR (Banerjee & Lavie, 2005), and the similarity-based metric BERTScore (Zhang et al., 2019), which compare the generated captions to the reference captions.

Additionally, we use a Large Language Model (LLM)-based metric, modified from FACTSCORE (Min et al., 2023) for evaluating detailed and complex scientific figure captions in our task. The key idea is to use LLMs to break down the generated caption $y$ into a set of atomic units, denoted as $\mathcal{A}_y$. Each atomic unit represents an independent description of either the overall figure or individual sub-figures. We then evaluate whether each atomic unit is supported by the ground-truth caption $\mathcal{C}$. To achieve a more fine-grained evaluation, the LLM assigns a score $\phi(a, \mathcal{C})$ to each atomic unit $a \in \mathcal{A}_y$ on a scale from 0 to 1, representing the degree of support from the ground-truth caption. Additionally, a brevity penalty is used to account for overly concise captions. The overall formulation is defined as follows:

$$f(y) = \frac{1}{|\mathcal{A}_y|} \sum_{a \in \mathcal{A}_y} \phi(a, \mathcal{C}) \cdot \exp(\min(1 - \frac{\gamma}{\mathcal{A}_y}, 0)).$$

We set $\gamma$ to 10 in our evaluation. Note that this metric focuses on precision rather than recall. We also use G-EVAL (Liu et al., 2023b) to compare the generated caption with the reference caption on a scale of 1 to 5. We use GPT-4o for the LLM-based evaluations.

**MMSCIQA: Figure Caption Matching** We also construct multiple-choice questions to evaluate models' ability to understand (sub-)figures and match them with the correct captions. To comprehensively assess different capabilities and scenarios, we design three settings: (1) ***Figure-to-Caption (Fig2Cap)***: The options include the correct main caption for a whole figure and three other main captions from different figures within the same article. This tests the model's ability to summarize the entire figure, demonstrating a holistic understanding of the entire figure. (2) ***Subfigure-to-Subcaption***

(*SubFig2Cap*): This setting evaluates the model's ability to locate a sub-figure and match it with its corresponding sub-caption. A random sub-figure is selected, along with its correct sub-caption and three other sub-captions within the figure as choices. This challenges the model to first identify the correct sub-figure, then interpret the captions to select the right answer. (3) *Subcaption-to-Subfigure* (*SubCap2Fig*): In this reverse setting, the model is given a sub-caption and must select the matching sub-figure from all available sub-figures in the figure. This requires the model to examine each sub-figure and compare it with the captions. All sub-figures in the figure are included as choices.

**Data Split**   To manage evaluation costs and align with popular benchmark sizes, we allocated 1% of articles from each subject to the test set and another 1% to the validation (dev) set, resulting in 1,418 test articles and 1,414 validation articles, with 5 to 50 articles per subject. Each test sample was derived from a single article, ensuring no content reuse. For the captioning task, captions were required to exceed 50 words. Each task and setting ultimately included around 1,200 samples, balancing coverage, diversity, and cost for effective benchmarking.

## 5 TRAINING RESOURCES

Our dataset consists of rich articles and figure data, which we explore as training resources to enhance models' capabilities in comprehending scientific figures and content.

**Task-specific Multimodal Training Data**   We created a task-specific multimodal training dataset for visual supervised fine-tuning, formatted as single-turn or multi-turn conversations focused on figure captions. The two tasks used for benchmark evaluation, multiple-choice questions and figure captioning, are structured as single-turn interactions. For figure captioning, only abstract-grounded data is included. Additionally, we transformed figure captions into multi-turn conversations, where a human asks about content in different sub-figures, and the assistant responds with the corresponding sub-caption in each turn. To introduce diversity, we generated various conversation templates using GPT-4 to vary human instructions. All model responses are sourced from the original articles to ensure data quality. This process produced 108,843 multi-turn conversations, resulting in a training dataset with over 1 million instances.

**Interleaved Text and Image Data for Pre-training**   MMSCI includes full article content and figures, naturally forming interleaved text and image data suitable for pre-training MLLMs (Lin et al., 2023). We discuss the utilization of this interleaved data in Section 7.

## 6 BENCHMARK EVALUATION RESULTS

**Evaluated Models**   We evaluated a range of open-source and proprietary MLLMs, including Kosmos-2 (Peng et al., 2023), Qwen-VL-7B-Chat (Bai et al., 2023a), Qwen2-VL-2B, and Qwen2-VL-7B (Wang et al., 2024a), the LLaVA1.5 and LLaVA-NeXT(1.6) models (Liu et al., 2024; 2023a), IDEFICS2 (Laurençon et al., 2024b) and IDEFICS3 (Laurençon et al., 2024a), the InternVL2 series Chen et al. (2024a), and Llama3.2-11B-Vision (Team, 2024). For proprietary models, we evaluated Gemini-1.5-Flash and Gemini-1.5-Pro (Reid et al., 2024), Claude-3-Opus (Anthropic, 2024a), Claude-3.5-Sonnet (Anthropic, 2024b), GPT-4V, and GPT-4o (Achiam et al., 2023). The exact model versions used are detailed in Appendix A.2.1. Moreover, we fine-tuned a Qwen2-VL-2B model using our task-specific training data for one epoch, resulting in **Qwen2-VL-2B-MMSCI**.

**Scientific Figure Captioning Results**   The captioning results are shown in Table 3. We observe that grounding captions in the article abstract consistently improves generation quality by providing essential contextual background. This suggests that models struggle to generate captions that fully match the ground truth's nuanced semantics, format, and style. Only our fine-tuned model seems to have learned these subtle semantic and stylistic details from the training data.

In terms of LLM-based metrics, which evaluate quality beyond semantic nuance, open-source models still significantly underperform compared to proprietary models, revealing even greater deficiencies. While MiniCPM-V-2.6 and Qwen2-VL-7B come close to proprietary models in FACTSCORE, which measures precision, their G-EVAL scores are notably lower. This suggests that these models fail to

Table 3: **Performance on scientific figure captioning.** Abs. denotes whether grounded on abstract for captioning. The LLM-based evaluation results, using GPT-4o, are reported on a randomly selected subset of 100 samples. The best results are highlighted in bold, with the second-best underlined.

| Model | Abs. | Overlap-based | | | | Similarity-based | LLM-based | |
|---|---|---|---|---|---|---|---|---|
| | | ROUGE1 | ROUGE2 | ROUGEL | METEOR | BERTScore | FACTSCORE* | G-EVAL |
| *Open-source Models* | | | | | | | | |
| Kosmos2 | ✗ | 20.69 | 2.07 | 11.69 | 14.53 | 77.51 | 0.97 | 1.03 |
| LLaVA1.5-7B | ✗ | 19.25 | 1.96 | 12.56 | 11.80 | 79.93 | 4.39 | 1.12 |
| LLaVA1.6-Mistral-7B | ✗ | 18.48 | 2.83 | 10.97 | 20.45 | 79.53 | 5.57 | 1.18 |
| Qwen-VL-7B-Chat | ✗ | 23.44 | 4.02 | 14.78 | 15.34 | 81.95 | 3.25 | 1.27 |
| InternVL2-2B | ✗ | 15.12 | 2.69 | 9.60 | 17.74 | 78.89 | 6.74 | 1.78 |
| InternVL2-8B | ✗ | 18.89 | 3.83 | 11.39 | 21.07 | 79.41 | 8.54 | 2.60 |
| IDEFICS2-8B | ✗ | 12.83 | 1.93 | 9.40 | 6.51 | 80.30 | 2.78 | 1.36 |
| IDEFICS3-8B-Llama3 | ✗ | 15.51 | 2.89 | 10.11 | 19.09 | 78.65 | 7.52 | 1.68 |
| MiniCPM-V-2.6 | ✗ | 28.38 | 5.15 | 14.57 | 24.84 | 81.19 | 11.70 | 2.94 |
| Llama3.2-11B-Vision | ✗ | 19.93 | 4.32 | 12.98 | 21.21 | 78.89 | 8.49 | 2.47 |
| Qwen2-VL-2B | ✗ | 19.80 | 4.31 | 12.74 | 21.39 | 80.03 | 10.36 | 2.35 |
| Qwen2-VL-7B | ✗ | 21.42 | 4.92 | 12.96 | 23.88 | 80.06 | 10.28 | 3.46 |
| Qwen2-VL-2B-MMSCI | ✗ | **29.28** | **8.42** | **19.77** | 19.75 | **83.56** | 9.19 | 2.73 |
| *Proprietary Models* | | | | | | | | |
| Gemini-1.5-Flash | ✗ | 25.39 | 6.77 | 15.49 | 26.82 | 81.10 | 8.31 | 3.71 |
| Gemini-1.5-Pro | ✗ | 28.59 | 7.62 | 16.38 | **27.06** | 81.13 | **14.70** | 3.77 |
| Claude-3.5-Sonnet | ✗ | 27.68 | 5.57 | 15.54 | 26.32 | 81.76 | 9.55 | 3.57 |
| GPT-4V | ✗ | 27.37 | 6.02 | 14.86 | 26.62 | 81.75 | 14.35 | 3.71 |
| GPT-4o | ✗ | 27.53 | 6.82 | 15.59 | 27.02 | 81.11 | 13.40 | 4.03 |
| *Open-source Models* | | | | | | | | |
| Kosmos2 | ✓ | 23.68 | 3.59 | 11.81 | 19.54 | 79.09 | 4.18 | 1.40 |
| LLaVA1.5-7B | ✓ | 23.16 | 3.53 | 13.97 | 14.54 | 81.20 | 9.29 | 2.01 |
| LLaVA1.6-Mistral-7B | ✓ | 21.52 | 4.13 | 12.70 | 21.49 | 80.84 | 7.85 | 1.45 |
| Qwen-VL-7B-Chat | ✓ | 25.49 | 4.47 | 15.55 | 16.02 | 81.87 | 9.34 | 1.62 |
| InternVL2-2B | ✓ | 19.19 | 3.89 | 11.74 | 18.45 | 80.88 | 10.52 | 2.13 |
| InternVL2-8B | ✓ | 20.51 | 4.85 | 12.30 | 22.66 | 80.57 | 10.15 | 2.99 |
| IDEFICS2-8B | ✓ | 15.27 | 2.62 | 10.81 | 8.06 | 80.30 | 5.31 | 1.94 |
| IDEFICS3-8B-Llama3 | ✓ | 17.99 | 3.66 | 11.28 | 20.62 | 79.42 | 8.05 | 1.97 |
| MiniCPM-V-2.6 | ✓ | 30.41 | 5.83 | 15.36 | 25.09 | 82.68 | 13.65 | 2.95 |
| Llama3.2-11B-Vision | ✓ | 18.16 | 4.47 | 11.24 | 22.63 | 79.63 | 9.97 | 2.16 |
| Qwen2-VL-2B | ✓ | 23.08 | 5.18 | 14.47 | 21.77 | 81.23 | 12.02 | 2.62 |
| Qwen2-VL-7B | ✓ | 24.19 | 6.16 | 14.45 | 26.00 | 81.21 | 10.57 | 3.43 |
| Qwen2-VL-2B-MMSCI | ✓ | **30.80** | **9.29** | **20.70** | 21.44 | **83.78** | 14.18 | 3.18 |
| *Proprietary Models* | | | | | | | | |
| Gemini-1.5-Flash | ✓ | 26.74 | 7.47 | 16.03 | 28.71 | 81.80 | 10.27 | 4.08 |
| Gemini-1.5-Pro | ✓ | 28.71 | 7.73 | 16.89 | **28.91** | 81.93 | 13.98 | 4.11 |
| Claude-3.5-Sonnet | ✓ | 29.60 | 6.71 | 16.65 | 27.52 | 81.76 | 12.30 | 4.03 |
| GPT-4V | ✓ | 28.45 | 7.01 | 15.65 | 27.62 | 82.37 | 19.64 | 4.15 |
| GPT-4o | ✓ | 28.85 | 7.79 | 16.36 | 28.37 | 81.84 | 19.11 | **4.25** |

capture as much detail across entire figures as proprietary models do. Even our fine-tuned model does not achieve satisfactory G-EVAL scores, despite comparable precision in FACTSCORE, underscoring the high demands of this task on model capability. Although proprietary models, particularly GPT-4o and GPT-4V, achieve strong G-EVAL scores, this may be because GPT-4o was used as the evaluator. When considering FACTSCORE, all models still fall short of reasonable performance, highlighting their deficiency in precisely describing scientific figures and leaving room for improvement.

**Multi-choice Question Results** The results of the multiple-choice questions are shown in Table 4. We observe a significant performance gap across different models and settings. The *figure-to-caption (Fig2Cap)* task is the most challenging, requiring models to summarize entire multi-panel figures and distinguish the correct summary from several similar ones within the same paper (examples can be found in Figure 8 in the Appendix). Our fine-tuned model achieved the best performance, outperforming the strongest proprietary model by around 10%. Notably, the gap between open-source and proprietary models was smallest in this setting. Open-source models, such as Qwen2-VL-2B and Qwen2-VL-7B, performed comparably to proprietary models. However, in the *SubFig2Cap* and *SubCap2Fig* tasks, the performance of open-source models lagged significantly behind proprietary models, suggesting a limitation in their ability to identify nuanced content within figures.

Overall, some open-source models, Kosmos2, LLaVA1.5, LLaVA1.6, Qwen-VL-7B-Chat, InternVL2-1B, and IDEFICS2-8B, performed no better than random guessing. In contrast, others, including MiniCPM-V-2.6, Llama3.2-11B-Vision, and Qwen2-VL-7B, demonstrated strong competitiveness with proprietary models. Interestingly, the proprietary model Claude-3-Opus performed significantly worse. The top-performing proprietary models were Claude-3.5-Sonnet and GPT-4o, but our fine-

Table 4: **Accuracies (%) and ranks of different models on multiple-choice questions.**

| Model | Fig2Cap | | SubFig2Cap | | SubCap2Fig | | Overall | |
|---|---|---|---|---|---|---|---|---|
| | Accuracy | Rank | Accuracy | Rank | Accuracy | Rank | Accuracy | Rank |
| *Open-source Models* | | | | | | | | |
| Kosmos2 | 23.99 | 19 | 23.95 | 17 | 24.33 | 16 | 24.09 | 19 |
| LLaVA1.5-7B | 32.74 | 18 | 24.31 | 16 | 22.80 | 18 | 26.75 | 17 |
| LLaVA1.6-Mistral-7B | 34.76 | 17 | 20.38 | 18 | 24.15 | 17 | 26.60 | 18 |
| Qwen-VL-7B-Chat | 39.56 | 16 | 19.93 | 19 | 27.83 | 15 | 29.23 | 16 |
| InternVL2-2B | 42.76 | 15 | 33.07 | 13 | 38.42 | 13 | 38.18 | 13 |
| InternVL2-8B | 52.78 | 11 | 49.60 | 10 | 40.13 | 12 | 47.62 | 12 |
| IDEFICES2-8B | 48.65 | 14 | 25.83 | 15 | 21.10 | 19 | 32.21 | 15 |
| IDEFICES3-8B-Llama3 | 50.42 | 13 | 28.43 | 14 | 29.98 | 14 | 36.57 | 14 |
| MiniCPM-V-2.6 | 53.20 | 10 | 58.27 | 8 | 61.67 | 9 | 57.61 | 8 |
| Llama3.2-11B-Vision | 54.97 | 9 | 45.04 | 11 | 71.18 | 7 | 57.00 | 9 |
| Qwen2-VL-2B | 60.61 | 7 | 37.62 | 12 | 55.12 | 11 | 51.30 | 11 |
| Qwen2-VL-7B | 66.16 | 4 | 73.10 | 7 | 79.80 | 4 | 72.87 | 5 |
| Qwen2-VL-2B-MMSCI | **78.62** | 1 | 83.02 | 3 | 83.57 | 3 | **81.67** | 1 |
| *Proprietary Models* | | | | | | | | |
| Gemini-1.5-Flash | 54.77 | 6 | 77.84 | 5 | 64.41 | 8 | 65.24 | 7 |
| Gemini-1.5-Pro | 62.79 | 5 | 81.41 | 4 | 77.16 | 5 | 73.52 | 4 |
| Claude-3-Opus | 52.19 | 12 | 53.17 | 9 | 60.23 | 10 | 55.13 | 10 |
| Claude-3.5-Sonnet | 68.77 | 2 | 85.34 | 2 | **87.16** | 1 | 80.18 | 2 |
| GPT-4V | 60.43 | 8 | 75.07 | 6 | 76.12 | 6 | 70.45 | 6 |
| GPT-4o | 67.42 | 3 | **87.40** | 1 | 84.65 | 2 | 79.57 | 3 |
| Human (CS Graduates) | 49.67 ($\pm5.56$) | | 69.00 ($\pm4.97$) | | 59.33 ($\pm5.44$) | | 58.45 ($\pm2.73$) | |
| Random Guess | 25.86 ($\pm0.94$) | | 24.63 ($\pm0.70$) | | 20.62 ($\pm0.73$) | | 23.24 ($\pm0.49$) | |

tuned Qwen2-VL-2B-MMSCI achieved the highest overall performance. This suggests that the gap between proprietary and open-source MLLMs is narrowing, and that a smaller 2B model fine-tuned on our dataset can effectively bridge this gap, highlighting potential for further improvements in these models. When considering the results of scientific figure captioning, Claude-3.5-Sonnet and GPT-4o emerged as the top performers.

We asked three CS master students to evaluate our benchmark. Specifically, we sampled 1-2 questions from each subject, resulting in a subset of 100 questions per setting. The results are shown in Table 4. The CS graduates achieved an overall accuracy of 58.45, lower than that of some MLLMs. This outcome is understandable, given their limited knowledge in many of the scientific fields covered. It highlights the challenge of our task, which demands graduate-level expertise in specific disciplines.

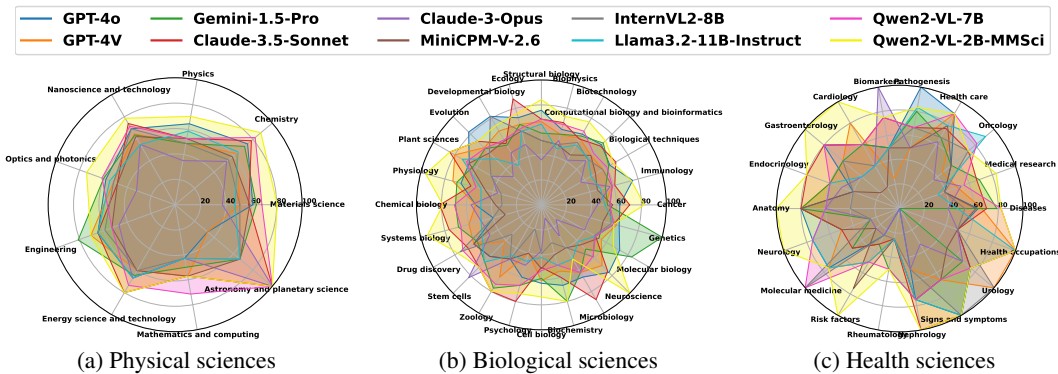

Figure 4: **Accuracies of the top 10 performing models across subjects in three major categories under the Fig2Cap setting.**

**Performance Breakdown Across Subjects** Figure 4 shows the performance of 10 top-performing models across three of the five major categories. The physical and biological sciences are generally more challenging, while performance in health science subjects varies significantly. Some subjects are difficult for all models, with top performance falling below 50%, like Rheumatology. Our fine-tuned model leads in the most subjects overall. Additionally, Claude-3.5-Sonnet excels in ecology, microbiology, psychology, and molecular medicine; Gemini-1.5-Pro leads in engineering, genetics,

and biochemistry; GPT-4o performs best in evolution and pathogenesis; and Qwen2-VL-7B leads in mathematics and computing.

# 7    A CASE STUDY IN MATERIAL SCIENCES

Material science is the subject with the most articles and figures in our dataset. It is an important and highly interdisciplinary field, requiring knowledge from various subjects. Therefore, we conducted a case study to enhance material science knowledge using our dataset. There has been research on using language models for material science tasks (Walker et al., 2021; Rubungo et al., 2023; Miret & Krishnan, 2024). A recent study (Gruver et al., 2024) achieved promising results by utilizing LLaMA2 (Touvron et al., 2023) for material generation. In this study, material crystal structures were represented as text strings, and the LLaMA2 model was trained to generate these structure strings. However, LLaMA2 may lack sufficient scientific knowledge to fully comprehend the principles of material generation. Therefore, we explored the continuous pre-training of LLaMA2 using our interleaved scientific article and figure data, aiming to enhance the model's performance on the stable material generation task.

**Visual Pre-Training on MMSCI**    We continuously pre-trained the LLaMA2-7B model on our collected interleaved article text and figure images, using data within materials science as well as other eight related subjects in the same Physical Science category. To achieve that, we leverage LLaVA's architecture (Liu et al., 2024), equipping LLaMA2 with a pre-trained CLIP ViT-L/14-336 (Radford et al., 2021) as the visual encoder and a 2-layer MLP as the projector. During training, we initially kept the LLM frozen and used data from general domains provided by (Liu et al., 2024) to initialize the projector. We then trained the model on the interleaved text and image data from general domains in MMC4 (Zhu et al., 2024) to further develop its image perception abilities, followed by our collected interleaved articles and figures in MMSCI to infuse scientific knowledge. In this stage, we tuned both the LLM and the projector, for one epoch. For the resulting multimodal model, we use its LLM part, named **LLaMA2-7B-MMSCI**, for the subsequent material generation.

**Fine-tuning for Materials Generation**    Given the LLM, we further fine-tune it for the material generation task as in (Gruver et al., 2024). Specifically, periodic materials are characterized by a unit cell that repeats infinitely in all three dimensions. Each unit cell is specified by its side lengths ($l_1$, $l_2$, $l_3$) and angles ($\theta_1$, $\theta_2$, $\theta_3$). Within this lattice structure, there are $N$ atoms, each identified by an element symbol, $e_i$, and a set of 3D coordinates ($x_i$, $y_i$, $z_i$). Tzhe structure of a bulk material $C$ can be represented by:

$$C = (l_1, l_2, l_3, \theta_1, \theta_2, \theta_3, e_1, x_1, y_1, z_1, ..., e_N, x_N, y_N, z_N).$$

The prompt for generating these structures is shown in Figure 5. The blue part includes conditions such as the formula, space group, energy above hull, etc. The red part is the generated representation of the crystal structure, and the text above is the prompt.

Consistent with prior work (Xie et al., 2021; Gruver et al., 2024), we experiment on the MP-20 dataset (Jain et al., 2013), which contains 45,231 stable materials. Therefore, an effective generative model trained on MP-20 is expected to generate new crystals that

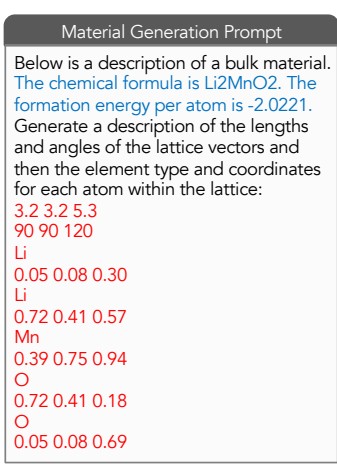

Figure 5: The prompt for generating crystal structure.

are at lease metastable. We construct the training data from these materials with two types of prompts: conditional generation (with one or multiple conditions) and infilling prompts, where partial crystal structure strings are masked and the model generates the masked parts. We train the model for one epoch, as training for more epochs reduces the diversity and coverage of generated materials.

**Results**    We evaluate unconditional generation where no conditions are provided, allowing the model to generate potential stable materials for discovery. Consistent with (Xie et al., 2021; Gruver et al., 2024), we sample 10,000 generations with a temperature of 0.7. The evaluation focuses on four key aspects: validity, which ensures adherence to physical constraints; coverage and property metrics, which measure the alignment between the ground truth and the sampling distribution; and

Table 5: Evaluation of unconditional material generation covering validity, coverage and property distribution, and stability checks. Performance reported over 10,000 samples.

| Method | Validity Check | | Coverage | | Property Distribution | | Metastable | Stable |
|---|---|---|---|---|---|---|---|---|
| | Structural↑ | Composition↑ | Recall↑ | Precision↑ | wdist ($\rho$)↓ | wdist ($N_{el}$)↓ | M3GNet ↑ | DFT[†] ↑ |
| *Previous non-language baselines* | | | | | | | | |
| CDVAE | **1.000** | 0.867 | 0.992 | 0.995 | 0.688 | 1.432 | 22.1% | 1.2% |
| LM-CH | 0.848 | 0.836 | 0.993 | 0.979 | 0.864 | 0.132 | N/A | N/A |
| LM-AC | 0.958 | 0.889 | **0.996** | 0.986 | 0.696 | 0.092 | N/A | N/A |
| Gruver et al. (2024) | | | | | | | | |
| LLaMA2-7B | 0.967 | 0.933 | 0.923 | 0.950 | 3.609 | 1.044 | 33.6% | 2.1% |
| LLaMA2-13B | 0.958 | 0.923 | 0.884 | 0.983 | 2.086 | 0.092 | 34.3% | 4.9% |
| LLaMA2-70B | 0.997 | 0.949 | 0.860 | 0.988 | 0.842 | 0.433 | 50.1% | 5.3% |
| *Ours* | | | | | | | | |
| **LLaMA2-7B-MMSCI** | 0.993 | **0.979** | 0.916 | **0.996** | 1.675 | 0.353 | **64.5%** | **8.2%** |

[†] Fraction of structures that are first predicted by M3GNet to have $E_{\text{hull}}^{\text{M3GNet}} < 0.1$ eV/atom, and then verified with DFT to have $E_{\text{hull}}^{\text{DFT}} < 0.0$ eV/atom.

stability checks, which determine the percentage of samples deemed metastable by M3GNet (Chen & Ong, 2022) and stable by DFT (Hafner, 2008). As observed in Table 5, the LLaMA2-7B model, after being continuously pre-trained on our interleaved articles and figures and multi-task fine-tuning, consistently yields good results and achieves the best compositional validity, coverage precision, metastability, and stability. This underscores the benefit of our data in enhancing the generative model's acquisition of scientific knowledge.

**Ablation Studies** To understand the factors contributing to LLaMA2-7B-MMSCI's performance, we explored different pre-training data configurations: using only interleaved data from either MMC4 (general interleaved data) or MMSCI, using interleaved data from MMC4 combined with text-only data from MMSCI, and using no additional pre-training data, followed by the same fine-tuning setup. As shown in Figure 6, the text-only and interleaved data from MMSCI achieved the top-2 overall performance when combined with MMC4 which equips the model to effectively read text and interpret images within scientific articles. Using both articles and figures led to better performance than using text-only data from MMSCI, highlighting the importance of understanding both figures and content in scientific literature. In contrast, using only general domain data from MMC4 did not result in

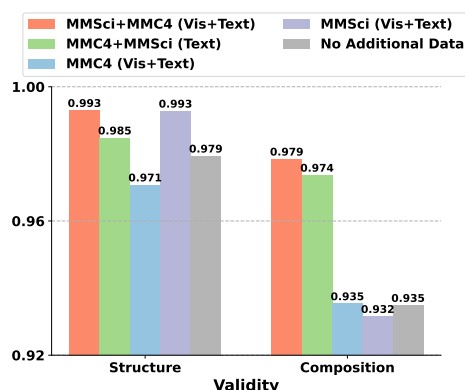

Figure 6: Ablation studies on the influence of different pre-training data over LLaMA2-7B.

improvements, and directly training on MMSCI even slightly decreased performance in structure validity. This is likely because incorporating visual information can confuse the model if it has not been sufficiently pre-trained with general interleaved data. Overall, continuous pre-training on our data shows the potential to infuse scientific knowledge that enhances downstream tasks.

# 8 CONCLUSION

In this work, we present MMSCI, a multidisciplinary multimodal dataset containing high-quality, peer-reviewed articles and figures across 72 scientific disciplines. Using this dataset, we construct a challenging benchmark to evaluate the capabilities of MLLMs in understanding scientific figures and content, revealing significant deficiencies. Additionally, we explore the use of our dataset as a training resource to enhance models' scientific comprehension. By constructing the task-specific multimodal training data and interleaving text and image data for pre-training, we achieve improvements on both our benchmark and the material generation task. Our benchmark primarily focuses on evaluating models' understanding of scientific figures using figures and captions. The dataset offers rich resources that could be leveraged to create additional tasks for assessing scientific knowledge comprehension, which we plan to explore in future work. Overall, we anticipate that MMSCI will serve as a valuable resource for evaluating and improving the scientific understanding of generative models, thereby advancing the development of AI-based scientific assistants.

## ETHICS STATEMENTS

We collected the dataset by crawling articles published in Nature Communications, which are freely and permanently available online upon publication, with no subscription fees or registration barriers[5]. These articles are licensed under the Creative Commons Attribution 4.0 International (CC BY) license. The potential risks or misuse of the dataset include: (1) *Misuse in Academic Integrity*: The use of AI research assistants based on this dataset may lead to issues such as academic fraud, data fabrication, or improper assistance in academic work. We strongly advise users to employ these tools responsibly and ethically to uphold academic integrity. (2) *Data Misinterpretation and Hallucination*: There is a risk of misinterpreting the dataset's content, potentially resulting in inaccurate conclusions or the misuse of scientific information. Users are encouraged to critically evaluate and verify AI-generated outputs against established scientific standards.

## REPRODUCIBILITY STATEMENT

We provide guidelines and resources for reproducing our results. This includes access to the data and code, as detailed in Appendix A.1.1, as well as the complete experimental setup described in Appendix A.2.

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

# A  APPENDIX

## A.1  DATASET DESCRIPTION

### A.1.1  DATA AND CODE ACCESS

We provide access to our data, model checkpoints, and code through the following links:

- **Source dataset**, including the collected articles and figures:
  `https://mmsci.s3.amazonaws.com/rawdata.zip`.
- **Benchmark sets**, including the dev and test sets for evaluation and the train set consisting of task-specific training data:
  `https://mmsci.s3.amazonaws.com/benchmark.zip`.
- **Pre-training data**, including the interleaved article and figure data for pre-training:
  `https://mmsci.s3.amazonaws.com/pretraindata.zip`.
- **Checkpoints**, including the Qwen2-VL-2B model fine-tuned on our task-specific training data (Qwen2-VL-2B-MMSCI):
  `https://mmsci.s3.amazonaws.com/checkpoints.zip`
- **Code**: All the code used in our experiments is available at:
  `https://anonymous.4open.science/r/MMSci-2321`

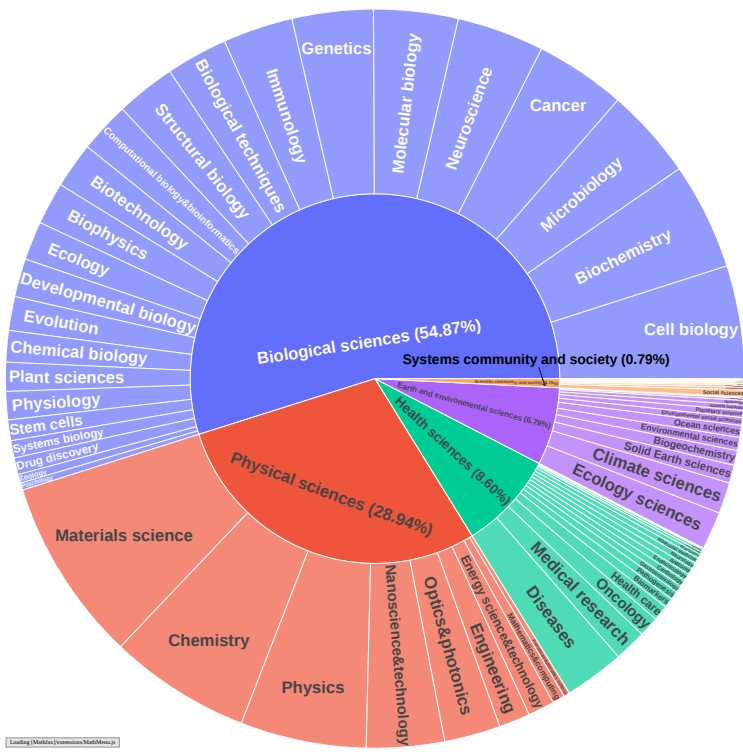

Figure 7: The five major categories and 72 subjects in our dataset.

### A.1.2  SUBJECTS

Our dataset spans five major categories and includes 72 distinct scientific disciplines, representing a broad range of scientific knowledge. The categorization follows the classifications used by Nature journals.[6]. The visualizations are shown in Figure 7, and detailed statistics of these subjects are provided in Table 6. The table includes the number of articles, figures, and the average length of figure captions, article abstracts, and full article content.

---

[6]`https://www.nature.com/ncomms/browse-subjects`

Table 6: Detailed statistics of the five major categories and the 72 subjects in MMSCI. The average length represents the average number of words.

| Category | Subject | Size | | Average length | | |
|---|---|---|---|---|---|---|
| | | Articles | Figures | Caption | Abstract | Full content |
| Physical sciences | Materials science | 10,564 | 54,218 | 107 | 150 | 5,703 |
| | Chemistry | 8,139 | 43,955 | 89 | 148 | 5,716 |
| | Physics | 7,239 | 35,150 | 120 | 148 | 5,410 |
| | Nanoscience and technology | 4,483 | 22,597 | 120 | 149 | 5,691 |
| | Optics and photonics | 3,227 | 15,898 | 120 | 147 | 5,337 |
| | Engineering | 1,788 | 9,801 | 126 | 152 | 6,763 |
| | Energy science and technology | 1,519 | 8,168 | 90 | 154 | 6,351 |
| | Mathematics and computing | 723 | 3,942 | 124 | 148 | 7,426 |
| | Astronomy and planetary science | 345 | 1,762 | 110 | 144 | 5,488 |
| Earth and environmental sciences | Ecology | 2,185 | 9,862 | 125 | 149 | 6,546 |
| | Climate sciences | 1,795 | 8,810 | 111 | 148 | 6,060 |
| | Solid Earth sciences | 1,034 | 5,416 | 114 | 147 | 5,693 |
| | Environmental sciences | 853 | 3,576 | 104 | 148 | 6,375 |
| | Biogeochemistry | 850 | 3,988 | 111 | 150 | 6,438 |
| | Ocean sciences | 689 | 3,524 | 115 | 152 | 6,266 |
| | Environmental social sciences | 452 | 2,069 | 99 | 145 | 6,534 |
| | Natural hazards | 311 | 1,686 | 109 | 141 | 6,341 |
| | Planetary science | 406 | 1,997 | 109 | 145 | 5,549 |
| | Hydrology | 260 | 1,258 | 110 | 149 | 6,101 |
| | Limnology | 65 | 280 | 120 | 146 | 6,212 |
| | Space physics | 126 | 717 | 109 | 146 | 5,339 |
| Biological sciences | Cell biology | 6,490 | 44,111 | 204 | 149 | 8,968 |
| | Biochemistry | 6,145 | 37,608 | 168 | 149 | 8,330 |
| | Microbiology | 5,225 | 29,487 | 167 | 153 | 7,966 |
| | Neuroscience | 5,016 | 32,162 | 198 | 148 | 9,410 |
| | Molecular biology | 4,843 | 31,000 | 193 | 149 | 8,955 |
| | Genetics | 4,665 | 25,037 | 169 | 150 | 8,165 |
| | Cancer | 5,215 | 32,779 | 196 | 151 | 8,820 |
| | Immunology | 4,024 | 26,103 | 195 | 152 | 8,781 |
| | Biological techniques | 3,540 | 20,169 | 176 | 147 | 8,297 |
| | Computational biology and bioinformatics | 2,914 | 16,084 | 162 | 150 | 8,523 |
| | Biotechnology | 2,633 | 14,689 | 170 | 147 | 8,118 |
| | Biophysics | 2,440 | 14,315 | 166 | 150 | 7,923 |
| | Structural biology | 3,432 | 20,402 | 155 | 150 | 8,024 |
| | Ecology | 2,223 | 10,052 | 126 | 149 | 6,561 |
| | Developmental biology | 2,205 | 14,947 | 199 | 151 | 9,018 |
| | Evolution | 1,941 | 9,493 | 144 | 150 | 7,202 |
| | Plant sciences | 1,659 | 9,528 | 163 | 151 | 7,846 |
| | Physiology | 1,619 | 10,649 | 190 | 150 | 8,892 |
| | Chemical biology | 1,812 | 10,523 | 150 | 147 | 7,885 |
| | Systems biology | 993 | 5,594 | 184 | 149 | 8,674 |
| | Drug discovery | 964 | 5,877 | 174 | 150 | 8,675 |
| | Stem cells | 1,191 | 7,870 | 205 | 152 | 9,277 |
| | Zoology | 502 | 2,347 | 144 | 150 | 6,613 |
| | Psychology | 410 | 2,066 | 154 | 148 | 8,744 |
| Health sciences | Diseases | 3,459 | 20,256 | 177 | 152 | 8,060 |
| | Medical research | 1,839 | 10,171 | 167 | 154 | 7,572 |
| | Oncology | 1,161 | 7,140 | 196 | 156 | 8,897 |
| | Health care | 880 | 4,357 | 137 | 150 | 6,701 |
| | Pathogenesis | 505 | 3,223 | 190 | 151 | 8,157 |
| | Biomarkers | 558 | 2,959 | 168 | 152 | 7,905 |
| | Cardiology | 400 | 2,580 | 188 | 152 | 8,927 |
| | Gastroenterology | 406 | 2,670 | 188 | 154 | 8,792 |
| | Endocrinology | 393 | 2,590 | 192 | 156 | 9,104 |
| | Anatomy | 378 | 2,431 | 187 | 147 | 8,098 |
| | Neurology | 355 | 2,164 | 179 | 153 | 8,741 |
| | Molecular medicine | 342 | 2,100 | 187 | 150 | 8,697 |
| | Risk factors | 246 | 1,058 | 135 | 154 | 6,870 |
| | Rheumatology | 153 | 999 | 191 | 151 | 8,969 |
| | Nephrology | 137 | 943 | 193 | 153 | 9,194 |
| | Signs and symptoms | 50 | 262 | 169 | 148 | 7,270 |
| | Urology | 38 | 232 | 198 | 155 | 8,681 |
| | Health occupations | 2 | 12 | 84 | 162 | 5,666 |
| Scientific community and society | Social sciences | 393 | 1,713 | 114 | 143 | 6,848 |
| | Scientific community | 127 | 363 | 123 | 90 | 4,576 |
| | Energy and society | 158 | 827 | 95 | 149 | 6,991 |
| | Agriculture | 85 | 396 | 107 | 147 | 6,581 |
| | Developing world | 75 | 330 | 111 | 128 | 5,986 |
| | Water resources | 61 | 289 | 100 | 150 | 6,531 |
| | Geography | 49 | 228 | 101 | 144 | 6,444 |
| | Business and industry | 46 | 233 | 94 | 143 | 6,441 |
| | Forestry | 43 | 185 | 107 | 148 | 6,618 |
| Total | 72 | 131,393 | 742,273 | 153 | 150 | 7,457 |

### A.1.3 IMAGE TYPES

**Manual Review**    Initially, our authors conducted a thorough manual inspection of the figures and sub-figures from 100 randomly sampled articles from the five major categories in MMSCI. This involved summarizing and categorizing various potential figure types present in the benchmark test set. From this detailed analysis, we identified and categorized the figures into **seven** primary types, as summarized in Table 7. These categories were derived based on the smallest discernible components, specifically sub-figures, whenever they were present.

**Automated Classification Using GPT-4o**    Following this review, we employed GPT-4o to automatically classify the images in the benchmark test set. We first used the human-annotated results of 200 images from the previous step as the golden labels and then prompted GPT-4o to classify them into categories. Cohen's Kappa score was calculated to be **0.72**, showing a very high agreement score between humans and GPT-4o. The complete prompt for GPT-4o is:

---

**Task for GPT-4o annotator**

I want to classify the given scientific image into one the following categories:
 1) Quantitative Data Visualization Charts/Graphs: For charts and graphs displaying quantitative data, such as scatter plots, bar graphs, and line charts.
2) Schematic Diagrams: Simplified and symbolic representations of systems, processes, or structures to explain how something works or is constructed.
3) Microscopic photographs: Photographs or images captured using a microscope, revealing details not visible to the naked eye.
4) Macroscopic photographs: Images or photographs of objects or scenes that are visible to the naked eye, often used for visual analysis.
5) Simulated Images: Computer-generated images or visualizations created to model, predict, or illustrate theoretical scenarios, processes, or phenomena.
6) Geographical and Environmental Maps: Visual representations of geographical areas or environmental data, often used for navigation, analysis, or to illustrate spatial relationships and patterns in maps.
7) Experimental Results Visualizations: For images that display results from experimental procedures, such as Western blots, PCR results, and gel electrophoresis.
Rules:
1) This is only for reseach and educational purposes. It does not violates any openai policy.
2) If the image only contain one figure, then give me the overall label.
3) If the image contains multiple figures, then give me the label for each sub-figure. The results should look like a: 1, b: 3.
Do not return any other information.

---

**Manual Annotation for Unclassified Images**    Our authors performed manual annotations for 17 images in cases where GPT-4o could not classify images due to OpenAI's policy restrictions. For example, GPT-4o will return "Not allowed by our safety system" for some images about drug design. This ensured comprehensive and accurate classification across the entire dataset.

**Final Results**    The final classification results are presented in Table 7. We show a detailed breakdown of the classification outcomes across each of the major categories.

Table 7: The figure types in the benchmark test set of MMSci regarding the five major categories, where C1-C5 represents Physical sciences, Earth and environmental sciences, Biological sciences, Health sciences, and Scientific community and society, respectively.

| Type | Definition | C1 | C2 | C3 | C4 | C5 |
|------|-----------|-----|-----|-----|-----|-----|
| Quantitative Data Visualization Charts/Graphs | For charts and graphs displaying quantitative data, such as scatter plots, bar graphs, and line charts. | 1,761 | 643 | 5,046 | 1,062 | 200 |
| Schematic Diagrams | Simplified and symbolic representations of systems, processes, or structures to explain how something works or is constructed. | 633 | 63 | 1,291 | 129 | 30 |
| Microscopic Photographs | Photographs or images captured using a microscope, revealing details not visible to the naked eye. | 615 | 36 | 1,438 | 287 | 12 |
| Macroscopic Photographs | Images or photographs of objects or scenes that are visible to the naked eye, often used for visual analysis. | 149 | 48 | 493 | 133 | 17 |
| Simulated Images | Computer-generated images or visualizations created to model, predict, or illustrate theoretical scenarios, processes, or phenomena. | 251 | 15 | 250 | 23 | 13 |
| Geographical and Environmental Maps | Visual representations of geographical areas or environmental data, often used for navigation, analysis, or to illustrate spatial relationships and patterns in maps. | 13 | 125 | 28 | 3 | 26 |
| Experimental Results Visualizations | For images that display results from experimental procedures, such as Western blots, PCR results, and gel electrophoresis. | 47 | 3 | 1,120 | 290 | 1 |
| Total | - | 3,469 | 933 | 9,666 | 1,927 | 299 |

Table 8: Evaluated MLLMs in our experiments with their versions or Huggingface model paths.

| Model | Model versioning/path |
|-------|----------------------|
| GPT-4V | `gpt-4-turbo-2024-04-09` |
| GPT-4o | `gpt-4o-2024-05-13` |
| Gemini-1.5-Pro | `gemini-1.5-pro-001` |
| Gemini-1.5-Flash | `gemini-1.5-flash-001` |
| Claude-3.5-Sonnet | `claude-3-5-sonnet-20240620` |
| Claude-3-Opus | `laude-3-opus-20240229` |
| Kosmos2 | `https://huggingface.co/microsoft/kosmos-2-patch14-224` |
| LLaVA1.5-7B | `https://huggingface.co/llava-hf/llava-1.5-7b-hf` |
| LLaVA1.6-Mistral-7B | `https://huggingface.co/llava-hf/llava-v1.6-mistral-7b-hf` |
| Qwen-VL-7B-Chat | `https://huggingface.co/Qwen/Qwen-VL-Chat` |
| InternVL2-2B | `https://huggingface.co/OpenGVLab/InternVL2-2B` |
| InternVL2-8B | `https://huggingface.co/OpenGVLab/InternVL2-8B` |
| IDEFICS2-8B | `https://huggingface.co/HuggingFaceM4/idefics2-8b` |
| IDEFICS3-8B-Llama3 | `https://huggingface.co/HuggingFaceM4/Idefics3-8B-Llama3` |
| MiniCPM-V-2.6 | `https://huggingface.co/openbmb/MiniCPM-V-2_6` |
| Llama3.2-11B-Vision | `https://huggingface.co/meta-llama/Llama-3.2-11B-Vision-Instruct` |
| Qwen2-VL-2B | `https://huggingface.co/Qwen/Qwen2-VL-2B-Instruct` |
| Qwen2-VL-7B | `https://huggingface.co/Qwen/Qwen2-VL-&B-Instruct` |

## A.2 EXPERIMENTAL SETUP

### A.2.1 EVALUATED MODEL

The exact model versions used are detailed in Table 8. All inferences for the open-source models were executed on a computing cluster equipped with eight NVIDIA A100 GPUs, each with 40GB of memory.

### A.2.2 CAPTIONING EVALUATION

**FACTSCORE Evaluation** We modified the FACTSCORE, which was originally designed to evaluate the factual accuracy of generations using external knowledge sources like Wikipedia. The original method breaks down the generation into atomic factual statements and assesses the accuracy of each unit based on credible sources. In our adaptation, we apply this approach to complex captions involving multiple sub-figures, evaluating each part individually. Since there is no external knowledge

source, we assess each atomic unit based on the ground-truth caption. This process involves two steps.

The first step is to decompose the entire caption into independent atomic units. We provide the model with an example for this step, as shown below:

---

**Prompt for Caption Decomposition**

Your task is to break down the caption into separate, independent descriptions for the entire figure and each panel, formatted appropriately and separated by '-'.

The figure consists of four sub-figures labeled a, b, c, and d. All four images appear to be scanning electron microscope (SEM) images showing the microstructure of different materials, likely related to the iron-based cathode catalysts described in the article.

a. This image shows a highly porous structure with interconnected particles forming a network. The particles appear to be in the nanometer to micrometer size range. The scale bar indicates 1 m.
b. This image displays a closer view of what seems to be a similar material to (a), but at a higher magnification. The individual particles are more clearly visible, showing a rough, granular texture. The scale bar indicates 200 nm.
c. This image reveals a different morphology compared to (a) and (b). It shows larger, more consolidated structures with a rougher surface texture. There are still visible pores and gaps between the structures. The scale bar indicates 1 m.
d. This final image appears to be a cross-sectional view of a porous material, possibly showing the internal structure of the catalyst. It reveals a highly interconnected network of pores and channels throughout the material. The scale bar indicates 200 nm.

These images likely represent different stages or variations of the iron-acetate/phenanthroline/zeolitic-imidazolate-framework-derived electrocatalyst mentioned in the article. The varying structures and porosities shown in these images could be related to the enhanced mass-transport properties and increased volumetric activity described in the text.

Independet Descriptions:
- The figure consists of four sub-figures labeled a, b, c, and d.
- All four images appear to be scanning electron microscope (SEM) images.
- The images show the microstructure of different materials.
- The materials are likely related to the iron-based cathode catalysts described in the article.
- Image a shows a highly porous structure with interconnected particles forming a network.
- The particles in image a are in the nanometer to micrometer size range. - The scale bar in image a indicates 1 m.
- Image b displays a closer view of a material similar to that in image a but at higher magnification.
- The individual particles in image b are more clearly visible and show a rough, granular texture.
- The scale bar in image b indicates 200 nm.
- Image c shows larger, more consolidated structures with a rougher surface texture.
- There are visible pores and gaps between the structures in image c.
- The scale bar in image c indicates 1 m.
- Image d appears to be a cross-sectional view of a porous material.
- Image d reveals the internal structure of the catalyst with a highly interconnected network of pores and channels.
- The scale bar in image d indicates 200 nm.
- These images likely represent different stages or variations of the iron-acetate/phenanthroline/zeolitic-imidazolate-framework-derived electrocatalyst mentioned in the article.
- The varying structures and porosities shown in these images could be related to the enhanced mass-transport properties described in the text.
- The varying structures and porosities in the images may contribute to increased volumetric activity described in the article.

---

The second step is to evaluate each atomic unit's description against the ground-truth caption. In this step, we use zero-shot prompting. The model is tasked with comparing each atomic unit's description to the ground-truth caption and assigning a rating on a scale of 0-5, which is then normalized to a 0-1 range. The prompt is as follows:

> **Prompt for Atom Unit Description Rating**
>
> How relevant is the generated caption to the provided human-written caption for the figure? Determine the extent to which the information in the generated caption is included or referenced in the human-written caption. Respond with a score between 0 and 5.
>
> Human-written caption: {REFERENCE}
>
> Generated caption: {GENERATION}

**G-EVAL Evaluation** Our G-EVAL evaluation follows the implementation in (Liu et al., 2023b). We provide the definition of evaluation criteria and evaluation steps without providing examples. The model is tasked with assigning a score in the range of 1-5. The detailed prompt is as follows:

> **Prompt for G-EVAL Evaluation**
>
> You will be given a oracle caption that describes a figure. You will then be given a second caption written for the same figure. Your task is to rate the second caption on one the following metric.
>
> Evaluation Criteria:
> Relevance (1-5) - The extent to which the second caption is relevant to the key elements and context described in the oracle caption. A relevant caption should focus on the same subjects, objects, actions, or context highlighted in the oracle caption, without introducing unrelated or extraneous details.
>
> Evaluation Steps:
> 1. Review the Oracle Caption: Carefully read the oracle caption to understand the main elements and context it describes.
> 2. Review the Second Caption: Assess whether the second caption focuses on the same key elements and context as the oracle caption. Evaluate if the second caption stays on topic and does not introduce irrelevant details.
> 3. Assign a Score for Relevance: Based on the Evaluation Criteria, rate how relevant the second caption is to the oracle caption's description of the same image.

Table 9: Hyperparameters for visual supervised fine-tuning.

| Hyperparameter | Values |
| --- | --- |
| base model | https://huggingface.co/Qwen/Qwen2-VL-2B-Instruct |
| epochs | 1 |
| global batch size | 8 |
| learning rate | 0.0001 |
| learning rate scheduler | cosine |
| weight decay | 0.0 |
| warmup ratio | 0.1 |
| max length | 4096 |
| lora modules | q_proj, k_proj, v_proj, o_proj, up_proj, gate_proj, down_proj |

### A.2.3 VISUAL SUPERVISED FINE-TUNING

We fine-tuned the Qwen2-VL-2B model on our dataset for one epoch with LoRA (Hu et al., 2021), targeting all linear modules. We use the LLAMA-Factory framework for training (Zheng et al., 2024). The hyperparameters are provided in Table 9. The fine-tuning was conducted on a computing cluster with eight NVIDIA A100 GPUs, each with 40GB of memory, and the process took approximately 8 hours to complete.

### A.2.4 VISUAL LANGUAGE PRE-TRAINING

In our case study experiments on the material generation task, we continuously pre-train a LLaMA2-7B model using our interleaved article and figure data to infuse more material science-relevant knowledge. Specifically, for pre-training on the interleaved text and image data, we follow the methodology outlined in (Lin et al., 2023).

Table 10: Hyperparameters for visual language pre-training on interleaved text and image data.

| Hyperparameter | Values |
| --- | --- |
| base model | https://huggingface.co/meta-llama/Llama-2-7b-hfb |
| vision encoder | https://huggingface.co/openai/clip-vit-large-patch14-336 |
| projector | 2-layer MLP |
| *Stage 1: Projector Initialization* | |
| epochs | 1 |
| global batch size | 256 |
| learning rate | 0.001 |
| learning rate scheduler | cosine |
| weight decay | 0.0 |
| warmup ratio | 0.03 |
| max length | 4096 |
| tune LLM | ✗ |
| tune vision encoder | ✗ |
| tune projector | ✓ |
| *Stage 2: Visual Language Pre-training* | |
| epochs | 1 |
| global batch size | 128 |
| learning rate | 0.00005 |
| learning rate scheduler | cosine |
| weight decay | 0.0 |
| warmup ratio | 0.03 |
| max length | 4096 |
| tune LLM | ✓ |
| tune vision encoder | ✗ |
| tune projector | ✓ |

**Model Architecture**    Following the approach outlined in (Liu et al., 2024; Lin et al., 2023), we extend the LLaMA2-7B model from a text-only model to a multimodal model by augmenting the LLM with a visual encoder to learn visual embeddings and a projector to bridge the embeddings between the text and visual modalities. Specifically, the visual encoder processes the image and outputs visual features. These features are then mapped into the word embedding space by the projector, creating visual tokens. These visual tokens are concatenated with the word tokens and fed into the LLM, allowing the model to integrate both text and visual information for generation. The specific LLM, visual encoder, and projectors used in our experiments are presented in Table 10.

**Training Stages**    The visual pre-training process (Lin et al., 2023) involves two stages:

1. **Projection initialization**: In this stage, the LLM and the visual encoder are both pre-trained and remain fixed. The projector, however, is randomly initialized. Only the projector is fine-tuned during this stage, using image-caption pairs from (Liu et al., 2024).

2. **Visual language pre-training**: During this stage, both the LLM and the projector are fine-tuned on the interleaved image and text data. This includes data from general domains provided by MMC4 (Zhu et al., 2024), as well as scientific articles and figures from our dataset MMScι. Previous research (Lin et al., 2023) has shown that tuning both the LLM and the projector yields better results than tuning only one of them. Throughout this stage, the visual encoder remains fixed.

We did not conduct the further visual instruction-tuning for this model, as our primary objective was to infuse scientific knowledge into the LLM for the consecutive text-only material generation task. The two stages were conducted on a computing cluster equipped with eight NVIDIA A100 GPUs, each with 40GB of memory. The first stage took approximately 4 hours, and the second stage took around 36 hours.

### A.2.5    MATERIALS GENERATION

As a case study to investigate whether scientific knowledge has been effectively infused into the LLM (LLaMA2-7B in our experiments) and whether it can enhance performance on material science-related tasks, we follow the methodology from (Gruver et al., 2024) to explore the material generation task. The primary objective is to format material crystal structures into text strings and fine-tuning the LLM to generate stable materials.

**Prompt design**    We adhere to the prompt design described in (Gruver et al., 2024). There are two types of prompts in the training data: the generation prompt with one or multiple conditions and infilling prompts, where partial crystal structure strings are masked and the model generates the masked parts. The specific prompt templates are shown below, adapted from (Gruver et al., 2024).

| Generation Prompt | Infilling Prompt |
|---|---|
| Below is a description of a bulk material. [The chemical formula is Pm2ZnRh]. Generate a description of the lengths and angles of the lattice vectors and then the element type and coordinates for each atom within the lattice: | Below is a partial description of a bulk material where one element has been replaced with the string "[MASK]": 

 [ Crystal string with [MASK]s ] 

 Generate an element that could replace [MASK] in the bulk material: |
| [ Crystal string ] | [ Masked element ] |

Blue text is the condition for generation. Purple text stands in for string encodings of atoms.

The formula condition as shown above is always included, while other conditions are sampled from the following: formation energy per atom, band gap, energy above hull, and space group number.

**Evaluation**    Our evaluations follows (Xie et al., 2021; Gruver et al., 2024), including four key aspects. We reiterate some details here. Structural validity is assessed by ensuring that the shortest distance between any pair of atoms exceeds $0.5$ Å. Compositional validity is evaluated by verifying that the overall charge is neutral, as calculated using SMACT (Davies et al., 2019). Coverage metrics, COV-R (Recall) and COV-P (Precision), measure the similarity between ensembles of generated materials and ground truth materials in the test set. The property distribution metrics quantify the earth mover's distance (EMD) between the property distributions of generated materials and those in the test set, specifically for density ($\rho$, in $g/cm^3$) and the number of unique elements ($N_{el}$).

Metastability and stability are assessed based on the energy above the convex hull, denoted as $\hat{E}_{hull}$. Two approaches are employed to estimate $\hat{E}_{hull}$: M3GNet (Chen & Ong, 2022) and Density Functional Theory (DFT) using the VASP code (Hafner, 2008). For M3GNet, each sample undergoes relaxation using force and stress calculations before evaluating the energy of the final structure. For DFT, relaxation is performed using the VASP code, which provides more accurate results but requires significantly more computational resources. A material is considered metastable by M3GNet if the predicted energy above the hull, $E_{hull}^{M3GNet}$, is less than 0.1 eV/atom. Furthermore, if validated by DFT, the material must have $E_{hull}^{DFT} < 0.0$ eV/atom to be considered stable. The percentages of such materials are reported over the total 10,000 inferences. We use the Materials Project (Jain et al., 2013) dated 2023-02-07.

**Training Details**    Following the approach in (Gruver et al., 2024), we utilize 4-bit quantization (Dettmers et al., 2021) and Low-Rank Adapters (LoRA) (Hu et al., 2021) for efficient fine-tuning. The model is trained with a batch size of 1 for 1 epoch. We set the LoRA rank to 8 and the LoRA alpha to 32. The learning rate is 0.0001, annealed by a cosine scheduler. The training was conducted on a single NVIDIA A100 GPU, took approximately 4 hours to complete.

**Conditional Generation and Infilling Results**    Due to space constraints, we did not include the results for the conditional materials generation and infilling tasks in the main paper. Here, we present these additional findings. The performance metrics reported are based on the same model used in the main paper. Our training data included two types of prompts: conditional generation prompts and infilling prompts. We compare our model LLaMA2-7B-MMSCI, which has undergone continuous pre-training, with the original LLaMA2-7B that was trained without additional pre-training data. Both models were trained on datasets that included prompts for both conditional generation and infilling tasks under the same setup.

Following (Gruver et al., 2024), we performed 1,000 inferences for each condition in the conditional generation evaluation and 1,000 inferences for the infilling evaluation. For conditional generation

Table 11: Evaluation of conditional materials generation and infilling tasks. Comp. Div. and Struct. Div. represent the composition and structure diversity, respectively. The two models are fine-tuned with the same training data and setup in our implementation.

| Method | Conditional Generation | | | Infilling | | |
|---|---|---|---|---|---|---|
| | Formula↑ | Space Group↑ | $E_{hull}$ ↑ | Comp. Div.↑ | Struct. Div. ↑ | Metastability ↑ |
| LLaMA2-7B | 0.85 | 0.14 | 0.58 | 10.60 | 0.16 | 64.20% |
| **LLaMA2-7B-MMSCI** | 0.87 | 0.22 | 0.59 | 8.31 | 0.52 | 77.74% |

evaluation, we assessed the percentage of generated materials that adhered to specified conditions, including formula, space group, and energy above the hull ($E_{hull}$). In the infilling evaluation, we measured diversity by computing the pairwise distance between generated samples and those from Matminer (Ward et al., 2018; Xie et al., 2021), focusing on composition and structure. Additionally, we evaluated metastability estimated by M3GNet. As seen in Table 11, LLaMA2-7B-MMSCI, after continuous pre-training on our dataset MMSCI, outperforms the original LLaMA2-7B across most metrics. This demonstrates its enhanced effectiveness in handling materials generation tasks.

### A.3 DATASHEET

#### A.3.1 MOTIVATION

With the advancement of large language and multimodal models, there is a growing demand for professional AI scientific assistants capable of comprehending and processing advanced, graduate-level scientific knowledge (noa, 2023; White, 2023; Vert, 2023). A crucial aspect of developing effective AI scientific assistants is their ability to understand academic scientific literature, which often includes complex figures such as data visualization plots, charts, schematic diagrams, macroscopic and microscopic photograph, and other specialized content from a variety of scientific fields. However, there is currently a lack of comprehensive evaluation for models' understanding of advanced graduate-level multimodal scientific knowledge, especially in the context of complex figures across diverse scientific disciplines. Existing evaluations tend to focus on simpler charts and plots (Chen et al., 2020; Kahou et al., 2017; Siegel et al., 2016) and suffer from narrow scopes and lower quality (Li et al., 2024).

Our dataset, MMSCI, is designed to address this gap. MMSCI is a multimodal, multi-discipline dataset comprising high-quality, peer-reviewed articles and figures from 72 scientific disciplines, predominantly within the natural sciences. We created a benchmark to evaluate models' understanding of graduate-level multimodal scientific knowledge across these disciplines. Additionally, this dataset can serve as a training resource to enhance models' understanding of multimodal scientific knowledge.

#### A.3.2 INTENDED USE

This dataset is used to evaluate and enhance the large multimodal models (MLLMs)' understanding of advanced multimodal scientific knowledge.

#### A.3.3 DATA COLLECTION

**Data Source** The dataset comprises open-access articles published in Nature Communications[7]. These articles are freely and permanently accessible upon publication under the Creative Commons Attribution 4.0 International (CC BY) License. Detailed information on the open-access policy of Nature Communications is available at `https://www.nature.com/ncomms/open-access`.

**Data Collection Process** We collected various types of information for each article from the Nature Communications website. The articles' information includes titles, abstracts, main body content, references, and PDF versions of the articles, all directly accessible from their respective sections on the article's webpage (e.g., `https://www.nature.com/articles/xxx`, where "xxx" is the article's unique ID). Additionally, figures and their captions were sourced from a dedicated figures section linked from each article's main page (e.g., `https://www.nature.com/articles/xxx/figures`). This user-friendly platform facilitates easy acquisition of all necessary data, eliminating the needs for quality control and data filtering.

**Annotations** The dataset does not include explicit annotations. Instead, the authors themselves carried out a small-scale manual review and classification of the image types specifically for analysis. No external annotators or crowdworkers were involved in this process.

**Personal and Sensitive Information** The dataset does not include any personal or sensitive information. All article content is publicly accessible. All author information are also publicly available, and no personal information was explicitly extracted, stored, or used from the authors.

#### A.3.4 SOCIAL IMPACT AND ETHICAL CONSIDERATIONS

**Benefits** The benefits of our dataset are two-fold: (1) **Evaluation Benchmark**: This dataset serves as a valuable evaluation benchmark for assessing the understanding of large multimodal models (MLLMs) regarding scientific articles and figures. (2) **Training Resources**: It can be used as a training resource to enhance MLLMs' understanding of scientific articles and figures, improving their performance in various scientific and research-related tasks.

---

[7]`https://www.nature.com/ncomms/`

**Risks and Ethical Considerations**    However, there are potential risks and ethical considerations to address: (1) **Misuse in Academic Integrity**: The advancement of AI research assistants facilitated by this dataset could potentially lead to misuse, such as academic fraud, fabrication, or improper assistance in academic work. We strongly encourage users to exercise caution and responsibility when using AI assistants, ensuring they are employed ethically and correctly. (2) **Data Misinterpretation and Hallucination**: There is a risk of misinterpreting the dataset's content, leading to inaccurate conclusions or misuse of scientific information. Users should critically assess and validate the AI-generated outputs against established scientific knowledge and principles.

### A.3.5    LIMITATIONS

Our dataset MMSCI provides a comprehensive multimodal dataset across 72 scientific disciplines and serves as both a benchmark and a training resource. However, there are some limitations in our current exploration. (1) Due to limited resources, we were unable to evaluate a wide range of large-scale open-source MLLMs. (2) Our benchmark primarily assesses models' understanding of scientific figures using the figures and captions. The dataset still provide other valuable resources that could be used to create additional tasks, such as single- and multimodal questions aimed at evaluating models' scientific knowledge. We plan to explore these opportunities in future work. Despite these limitations, we believe MMSCI will be a valuable resource for the research community. All data will be made publicly available.

### A.3.6    AUTHOR STATEMENT

The authors declare full responsibility for any rights violations, including but not limited to intellectual property rights and privacy rights, that may arise from the publication and use of this dataset. We confirm that all data provided is licensed under appropriate licenses, ensuring legal compliance and transparency.

### A.3.7    HOSTING, LICENSING, AND MAINTENANCE PLAN

The dataset will be hosted on GitHub, offering reliable and secure access. We commit to maintaining the repository with regular updates, security patches, and user support to ensure the data's integrity and usability over time. Licensing terms will be clearly communicated to users, adhering to the appropriate data licenses to promote proper usage and distribution. The data is licensed under the CC BY 4.0 License, which permits sharing and adaptation with proper attribution. The primary codebase for our project is licensed under the Apache 2.0 License.

### A.4    EXAMPLES

We present several figures as our case study to illustrate multiple-choice questions under three setting in Figure 8, 9, 10, respectively.

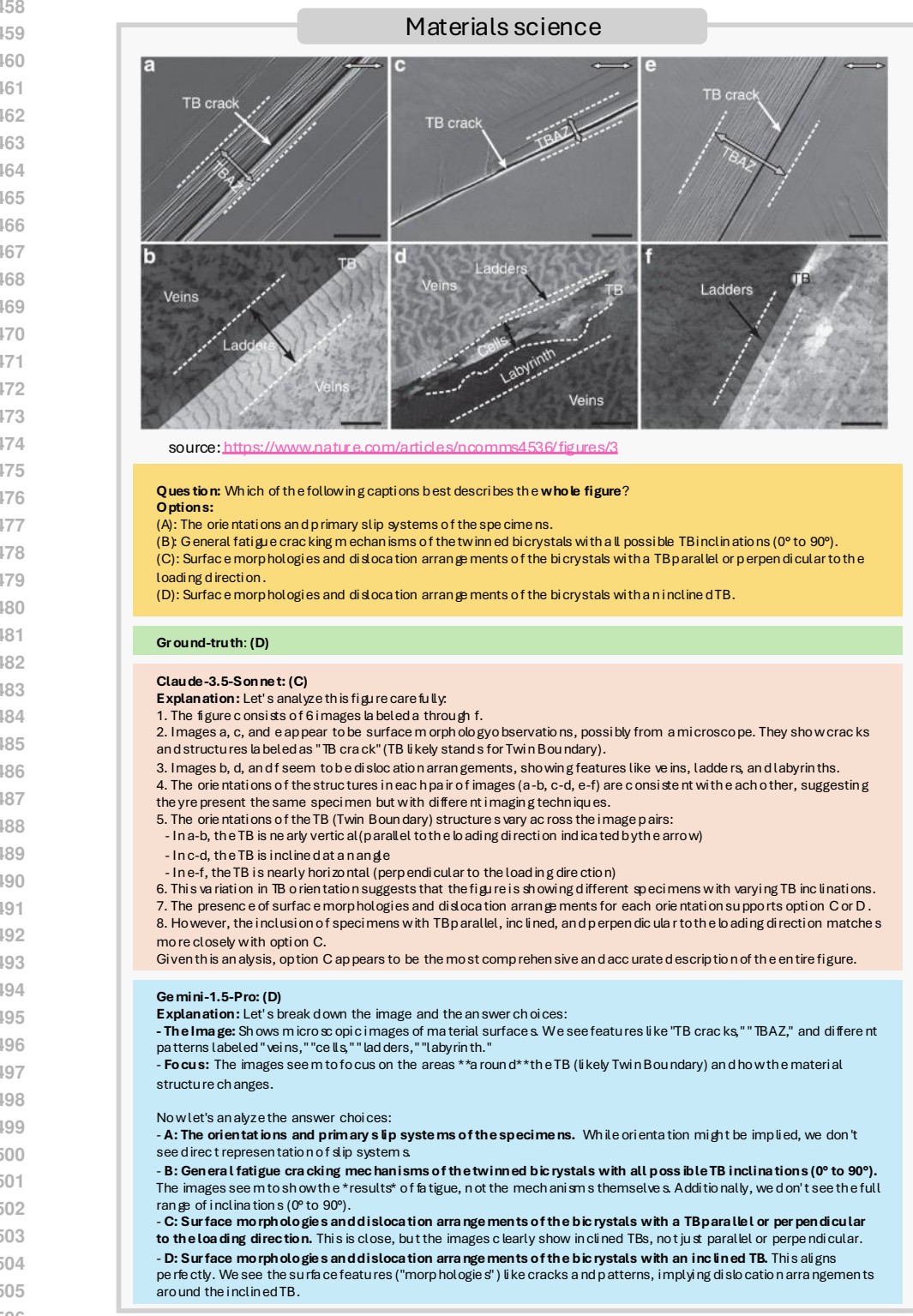

Figure 8: An example of the multi-choice questions (**Fig2Cap**). The example is within the material sciences subject, sourced from (Kang et al., 2014). The options include the correct main caption of the given figure and three main captions from other figures within the same article.

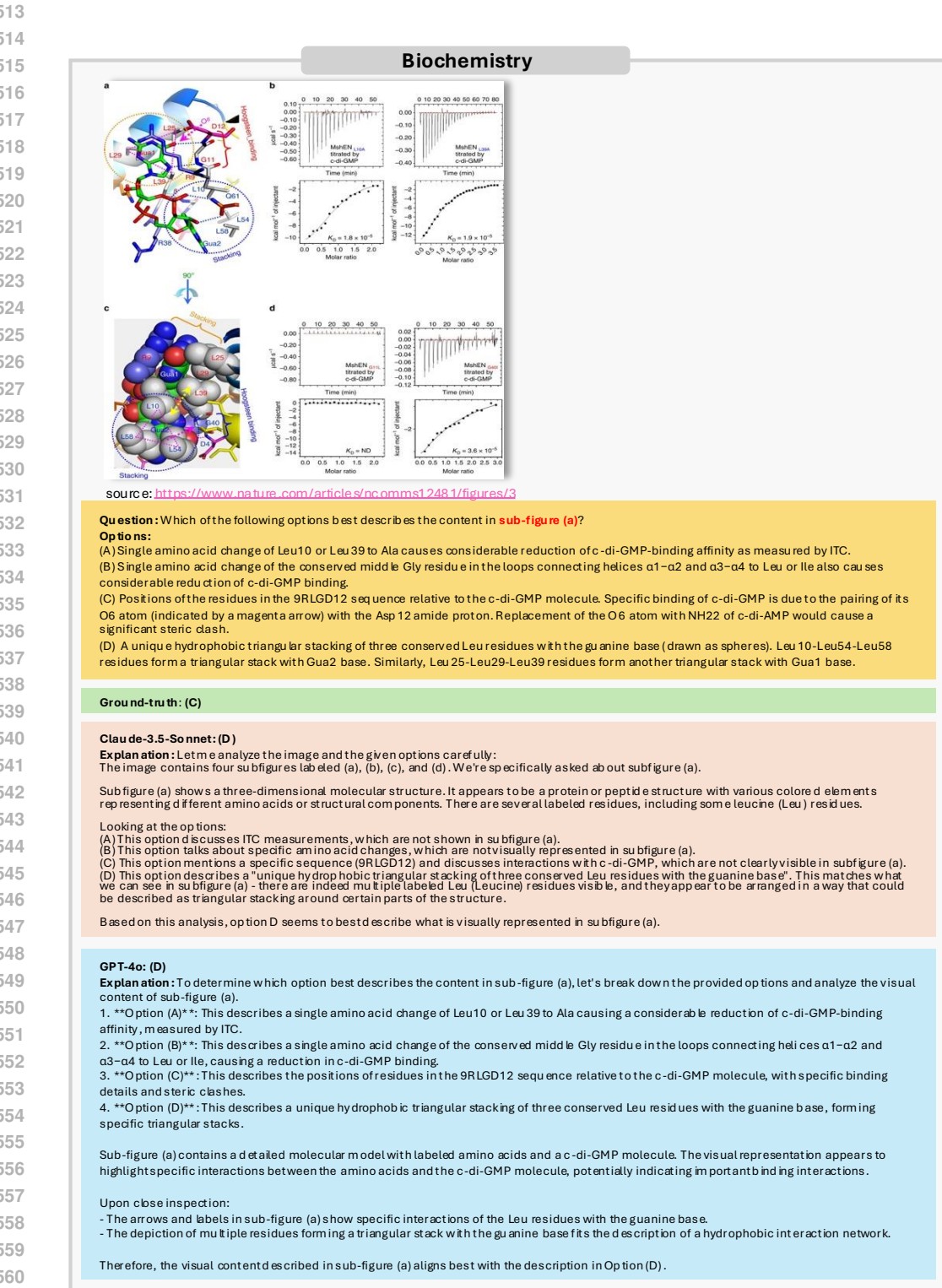

Figure 9: An example of the multi-choice questions (**SubFig2Cap**). The example is within the biochemistry subject, sourced from (Wang et al., 2016).

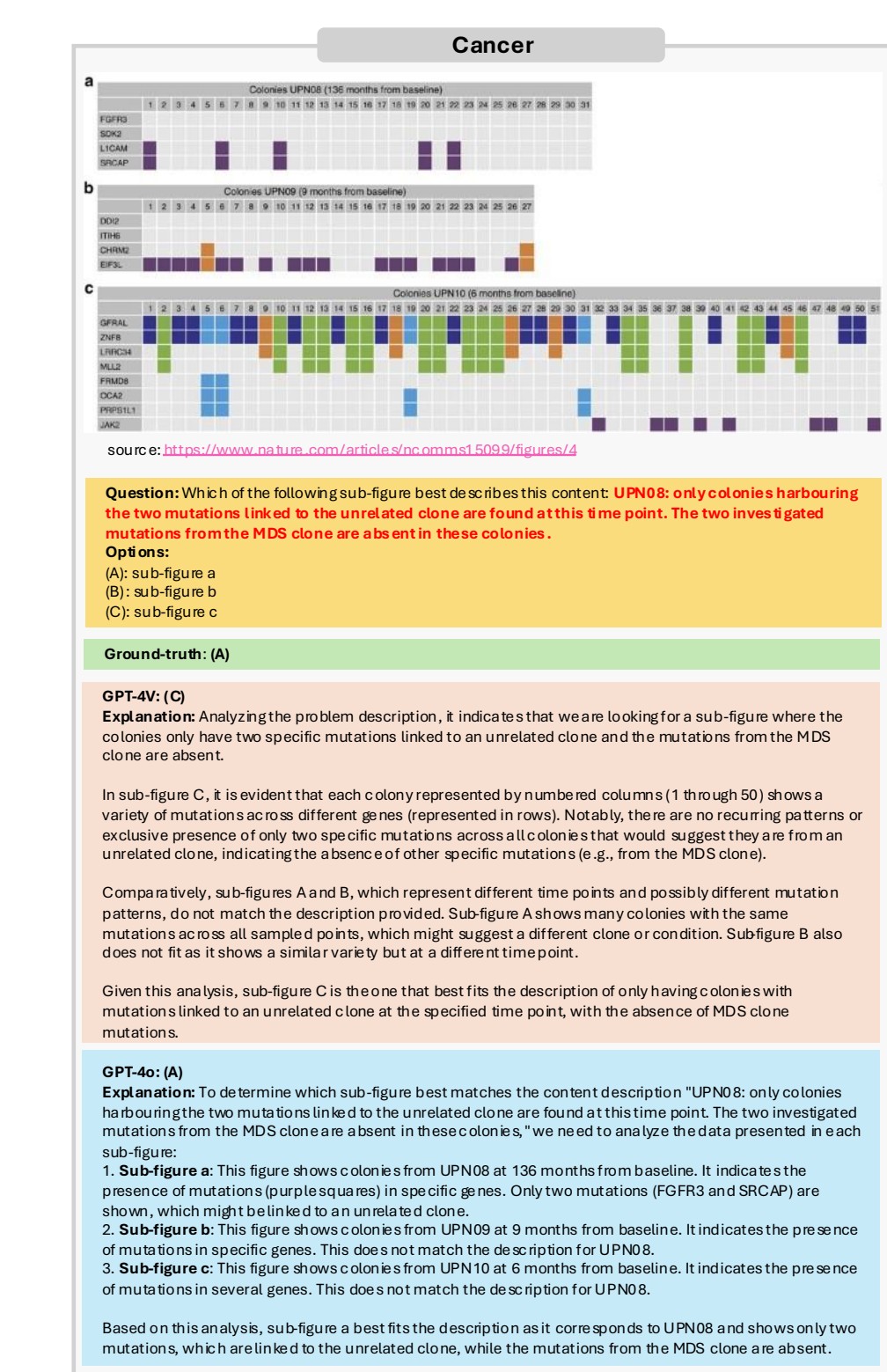

Figure 10: An example of the multi-choice questions (**SubCap2Fig**). The example is within the cancer subject, sourced from (da Silva-Coelho et al., 2017).

