# OpenReview forum: "MMSci: A Dataset for Graduate-Level Multi-Discipline Multimodal Scientific Understanding"
_ICLR.cc/2025/Conference — Submitted to ICLR 2025_

### Official Review · Reviewer_4ejM · 2024-10-27

**Soundness:** 2
**Presentation:** 3
**Contribution:** 2
**Rating:** 5
**Confidence:** 4

**Summary:**

This paper presents a curated dataset derived from peer-reviewed Nature Communications articles that spans 72 diverse scientific fields. The dataset encompasses a rich diversity of figure types, including schematic diagrams and experimental visualizations. Through experimental benchmarks, the authors evaluate MLLM (Multimodal Language Model) performance on tasks like scientific figure captioning and multiple-choice question answering. The results, presented across both proprietary and open-source models, underscore the significant challenges MLLMs face when managing complex scientific content. Additionally, the paper highlights the dataset’s potential for pre-training in material science.

**Strengths:**

- The paper is well-structured and written in a clear, accessible manner.
- The dataset, MMSCI, is vast, comprising over 131k articles and 742k figures. It spans 72 scientific fields, providing a wide array of figure types from research articles.
- The multiple-choice question task is well-designed, utilizing authenticated labels from the articles, which is a reasonable and realistic setting.
- The pre-training experiment is a strong demonstration of MMSCI’s value, showcasing the dataset’s utility.

**Weaknesses:**

- The MMSCICAP (Scientific Figure Captioning) task seems flawed. Providing only the abstract rather than figure-specific sentences—or even the entire relevant section—may undermine the task’s grounding. Abstracts alone should suffice only for captioning teaser figures; otherwise, the lack of figure-paired text may contribute disproportionately to the task’s difficulty.
- Multidisciplinarity is highlighted as a key feature, yet the experiments lack in-depth analysis on performance variation across disciplines. What factors contribute to these variations across different subjects? For example, why does Astronomy and Planetary Science yield high accuracy while disciplines with more data, like Materials Science, perform less well?
- Given the depth of scientific tasks, testing models like GPT-o1 or LMMs with enhanced reasoning abilities (e.g., chain-of-thought) could yield crucial insights for LMM development. While the dataset may not encompass all perspectives, enhanced reasoning capabilities could offer a more generalized solution to the complex nature of scientific reasoning.
- Line 230 mentions a figure caption but should clarify the “right” side to avoid confusion, as the caption does not appear on the “left” side.
- Minor typos:
  - Line 98, “Figure types” is preferable to “figures types.”
  - Consistency is needed in terminology: “Materials Science” appears in Figure 4(a) but “Material Science” in line 438.

**Questions:**

- How does GPT-o1 or other LMMs with advanced reasoning techniques (e.g., chain-of-thought) perform on these tasks?
- What factors drive the performance differences among disciplines?
- Why does Astronomy and Planetary Science show high accuracy compared to Materials Science, which has more articles and figures in the dataset?

**Details Of Ethics Concerns:**

Firstly, the paper references the Creative Commons Attribution 4.0 International (CC BY) license for Nature Communications content, with a link to the copyright policy at Nature Communications Open Access. However, it is not ideal to place the burden on readers to verify these copyright terms. Clarifying specific permissions and limitations associated with the dataset would be highly beneficial. Specifically:
- Is it legal to release the dataset in its entirety given that it draws exclusively from Nature Communications articles?
- Can other researchers freely conduct tests on the dataset?
- Are other researchers permitted to fine-tune models on the dataset?
- Is the dataset suitable for pre-training purposes in a broader research context?
- Can the dataset legally support commercial applications?

Clarifying these aspects will provide essential guidance to researchers following your work.

Secondly, given the dataset’s focus on academic caption generation, there are ethical considerations, especially around the risk of perpetuating inaccuracies. Such potential for misinformation, if unchecked, could have adverse impacts, particularly in educational and research environments. Introducing a brief discussion on mitigating these risks would enhance the responsible application of this work in various contexts.

---

> ### Author Response · Authors · 2024-11-23
> **Response to reviewer 4ejM**
>
> Thank you for your thoughtful feedback and for recognizing the strengths of our work, including the large and diverse MMSCI dataset, the realistic and well-crafted tasks, and the pre-training experiment. We truly appreciate your insights. Below, please find our detailed responses to your concerns.
>
> > **C1**: The MMSCICAP (Scientific Figure Captioning) task seems flawed. Providing only the abstract rather than figure-specific sentences—or even the entire relevant section—may undermine the task’s grounding. Abstracts alone should suffice only for captioning teaser figures; otherwise, the lack of figure-paired text may contribute disproportionately to the task’s difficulty.
>
> **Response**: Thank you for the suggestions. We also conducted experiments where the entire article was provided to the model, challenging its ability to read the full text and identify information relevant to the figure. Given the extensive length of the full article, we evaluated on the proprietary model with long-context capability. The results are provided below:
>
> | Model | Ctx. | BLEU-4 | ROUGE-1 | ROUGE-2 | ROUGE-L | Meteor | BertScore | FActScore | G-Eval |
> | ---- | ---- | ---- | ---- | ---- | ---- | ---- | ---- | ---- | ---- |
> | Gemini-1.5-Flash | Abstract | 3.29 | 26.74 | 7.47 | 16.03 | 28.71 | 81.80 | 10.14 | 4.08 |
> | Gemini-1.5-Pro | Abstract | 3.33 | 28.71 | 7.73 | 16.89 | 28.91 | 81.93 | 13.76 | 4.08 |
> | Claude-3.5-Sonnet | Abstract | 3.20 | 29.60 | 6.71 | 16.65 | 27.52 | 81.76 | 12.11 | 4.04 |
> | GPT-4V | Abstract| 3.18 | 28.45 | 7.01 | 15.65 | 27.62 | 82.37 | 19.52 | 4.13 |
> | GPT-4o | Abstract | 3.58 | 28.85 | 7.79 | 16.36 | 28.37 | 81.84 | 18.87 | 4.22 |
> | Gemini-1.5-Flash | Full Article | 6.94 | 32.83 | 14.15 | 22.02 | 34.50 | 83.26 | 19.41 | 4.12 |
> | Gemini-1.5-Pro | Full Article | 7.01 | 32.24 | 13.34 | 19.32 | 33.75 | 83.18 | 19.33 | 4.22 |
> | Claude-3.5-Sonnet | Full Article | 7.99 | 37.63 | 13.61 | 23.63 | 34.66 | 84.34 | 21.67 | 4.52 |
> | GPT-4V | Full Article | 5.65 | 33.09 | 10.95 | 19.25 | 31.46 | 83.48 | 23.18 | 4.24 |
> | GPT-4o | Full Article | 9.90 | 37.06 | 17.63 | 24.89 | 37.52 | 83.64 | 24.12 | 4.58 |
>
> While the performance did improve when using the full article content compared to using only the abstract, we want to emphasize that including figure-related content from the article might introduce unfairness. Authors sometimes repeat or closely mirror figure captions within the main text, which could lead to inflated performance metrics that do not reflect genuine model understanding. Given this consideration, we primarily use abstract-based captioning, which we believe could ensure a fairer evaluation while still providing contextual information about the figure's topic and the article's subject matter.
>
> > **C2**: Multidisciplinarity is highlighted as a key feature, yet the experiments lack in-depth analysis on performance variation across disciplines. What factors contribute to these variations across different subjects? For example, why does Astronomy and Planetary Science yield high accuracy while disciplines with more data, like Materials Science, perform less well?
>
> **Response**:
> Thank you for the valuable feedback. To gain deeper insights into the data across different disciplines, we engaged domain experts (PhDs in the relevant fields) to analyze our dataset. After reorganizing the original 72 categories into 18 scientific fields, we further prioritized 10 major science domains for detailed analysis.
>
> **PhD Expert Evaluation**: Specifically, we recruited 30 PhDs as human evaluators from the [Prolific platform](https://www.prolific.com/), which is a well established platform of high quality scores, with verified degrees in 10 major scientific categories that align with our dataset's primary domains: **Material Science**, **Chemistry**, **Physics**, **Biochemistry**, **Environment**, **Climate Sciences**, **Earth Sciences**, **Biological Sciences**, **Biomedical Sciences**, and **Health and Medicine**.
>
> Our human evaluation protocol covered 10 broad scientific domains. For each domain, we selected 75 questions (25 from each of our three figure-caption matching tasks), yielding a total evaluation set of 750 questions. We recruited 30 PhD-level evaluators through Prolific, specifically 3 PhDs per domain, with verified degrees in their respective fields. Each evaluator provided two types of assessments:
>
>  - **Performance Score**: Evaluators answered the questions themselves, establishing a human performance baseline measured by accuracy (0-100%)
>  - **Quality Assessment**: Evaluators rated each question's quality on a 5-point scale (1-5), judging its clarity and whether it effectively tests domain-specific knowledge.

---

> > ### Author Response · Authors · 2024-11-23
> > **[2/n] Response to Reviewer 4ejM**
> >
> > Specifically, for the quality assessment, the human evaluators are asked to assess **whether the questions were clear and required an understanding of scientific knowledge within the respective discipline**, as outlined in the following metrics:
> >
> >  - **Score Point 1. Strongly Disagree**: The question is irrelevant or cannot be answered based on the scientific content presented in the figure.
> >  - **Score Point 2. Disagree**: The question lacks clarity or can be answered without specific knowledge of the scientific content in the figure (e.g., it can be answered with common sense).
> >  - **Score Point 3. Neutral**: The question is clear but requires only minimal understanding of the scientific content in the figure.
> >  - **Score Point 4. Agree**: The question is clear, answerable, and requires an adequate understanding of the scientific content in the figure.
> >  - **Score Point 5. Strongly Agree**: The question is clear, answerable, and effectively evaluates a very deep understanding of the scientific content in the figure.
> >
> > The evaluation results are shown in the below table.
> >
> > **Table 1 in Rebuttal: Phd expert evaluation on the question quality.** Setting I indicates Figure-to-caption matching; Setting II indicates Subfigure-to-caption matching; Setting III indicates Caption-to-subfigure matching.
> >
> > | Field               | Setting I | Setting II | Setting III |
> > |---------------------|-------------------------|-------------------------------|-------------------------------|
> > | Material Science    | 4.0267                  | 4.2933                        | 4.1333                        |
> > | Chemistry           | 4.1333                  | 3.7467                        | 3.6133                        |
> > | Physics             | 4.0267                  | 3.5467                        | 3.8133                        |
> > | Biochemistry        | 3.1600                  | 4.8267                        | 4.4133                        |
> > | Environment         | 4.1067                  | 4.4667                        | 4.3467                        |
> > | Climate Sciences    | 4.1296                  | 3.6471                        | 3.4118                        |
> > | Earth               | 4.0267                  | 4.2319                        | 4.1739                        |
> > | Biological Sciences | 3.8800                  | 3.6800                        | 3.7867                        |
> > | Biomedical Sciences | 4.0133                  | 4.1333                        | 3.7733                        |
> > | Health and Medicine | 4.3733                  | 3.7467                        | 3.6800                        |
> > | **Average**         | **4.0873**              | **4.0319**                    | **3.9149**                    |
> >
> >
> > As observed, the overall quality scores are consistently around 4 across all settings, indicating that **the questions are clear, answerable, and require adequate understanding of scientific content in the figures.** The score distributions across different Q&A tasks reveal several key insights:
> >
> > 1. Varying Difficulty Levels: Different settings challenge different aspects of scientific understanding. For example, in Biochemistry, Setting II (subfigure-to-caption) scores significantly higher (4.83) than Setting I (3.16), suggesting that detailed technical matching requires deeper domain knowledge than overall figure understanding.
> > 2. Domain-Specific Patterns: Some fields like Material Science show consistent scores across all settings (4.03, 4.29, 4.13), while others like Biochemistry show larger variations (3.16, 4.83, 4.41). This suggests that our three task types effectively capture different aspects of domain expertise.
> > 3. Complementary Evaluation: The varying performance patterns across settings demonstrate that our benchmark tests scientific understanding from multiple angles, going beyond simple pattern matching. Each setting provides a distinct perspective on how domain knowledge is applied in scientific figure interpretation.
> >
> > In addition to the quality assessment reported above, we also evaluated the actual performance of PhD experts on our benchmark to address the concerns about human baseline design. We conducted a comprehensive evaluation with PhD experts from the Prolific platform across 10 major scientific categories. Our human evaluation includes two settigns:
> >
> > - **In-Ph.D. Degree Domain Expert Performance**: Performance of human Ph.D. evaluating questions within their specialty domain of Ph.D. degree
> >  - **Out-of-Ph.D. Degree Domain Performance**: Performance of human Ph.D. evaluating questions in Science, but outside their expertise of Ph.D. degree

---

> > > ### Author Response · Authors · 2024-11-23
> > > **[3/n] Response to Reviewer 4ejM**
> > >
> > > **Table 2 in Rebuttal: Performance Comparison of Human Ph.D. Experts and Models with a Newly Constructed sub-Benchmark (%)**
> > >
> > > | Model/Evaluator               | Setting I | Setting II | Setting III |
> > > |---------------------|-------------------------|-------------------------------|-------------------------------|
> > > | Outside-Ph.D. Degree Domain in Science (Human, Ph.D.) | 36.18 | 33.31 | 46.85 |
> > > | In-Ph.D. Degree Domain (Human, Ph.D.) | 40.26 | 43.45 | 50.46 |
> > > | Gemini-1.5-Flash (Overall) | 58.11 | 79.88 | 62.66 |
> > > | Gemini-1.5-Pro (Overall) | 63.47 | 78.83 | 78.65 |
> > > | Claude-3.5-Sonnet (Overall) | 70.58 | 85.73 | 83.70 |
> > > | GPT-4V (Overall) | 66.18 | 73.36 | 76.15 |
> > > | GPT-4o (Overall) | 68.42 | 88.23 | 84.92 |
> > > | Qwen2-VL-7B-MMSci (Overall) | 83.84 | 92.91 | 87.17 |
> > >
> > > As demonstrated in Table 2 in Rebuttal, several key insights emerge:
> > >
> > > **Domain Expertise Impact**: PhDs evaluating within their expert domain of Ph.D. degree consistently outperform those working outside their expertise in other Science domains across all settings, confirming our tasks require domain-specific knowledge
> > >
> > > **Task Difficulty**: Even domain experts achieve moderate performance within the recommended time limit (2 minutes per question), indicating the tasks' challenging nature
> > >
> > > **Scientific Understanding Capabilities**: The strong performances of both our fine-tuned sLLM and large proprietary models demonstrate potential as AI research assistants. Notably, our dataset significantly improved sLLM performance, reducing the gap with large proprietary models (see Table 4 in main paper)
> > >
> > > We will also provide the breakdown of different domains in the Figure 4 in the updated draft.
> > >
> > > >
> > > > **C3**: Given the depth of scientific tasks, testing models like GPT-o1 or LMMs with enhanced reasoning abilities (e.g., chain-of-thought) could yield crucial insights for LMM development. While the dataset may not encompass all perspectives, enhanced reasoning capabilities could offer a more generalized solution to the complex nature of scientific reasoning.
> > >
> > > **Response**: Thank you for the suggestion. We would like to clarify that the current version of o1 does not support image input, and therefore cannot be evaluated on this test set. Additionally, the results we present for proprietary models are their zero-shot chain-of-thought performance. (For the other open-source models, chain-of-thought prompting had little or negative impact, so we report standard input-output prompting performance.) Please find the performance with GPT-4o with different prompting approach below.
> > >
> > > | Model | Prompting | Setting I | Setting II | Setting III |
> > > | ---- | ---- | ---- | ---- | ---- |
> > > | GPT-4o | w/o CoT | 60.34 | 78.91 |  81.30 |
> > > | GPT-4o | 0-shot CoT | 67.42 | 87.40 | 84.65 |
> > > | GPT-4o | 1-shot CoT | 69.87 | 89.61 | 85.86 |
> > > | GPT-4o | 3-shot CoT | 69.70 | 90.97 | 85.65 |
> > > | GPT-4o | 5-shot CoT | 70.20 | 90.78 | 86.17 |
> > >
> > > As shown, chain-of-thought (CoT) reasoning improves performance for the powerful model, and increasing the number of shots provides a slight additional improvement, though not significantly. For a fair comparison, we use the 0-shot setting by default.
> > >
> > > > **C4**: Line 230 mentions a figure caption but should clarify the “right” side to avoid confusion, as the caption does not appear on the “left” side.
> > >
> > > **Response**: Thanks for the suggestions. We will modify the description as suggessted to avoid confusion.
> > >
> > > > **C5 regarding typos**:
> > >
> > > **Response**: Thanks for pointing them out. We will modify them in the updated draft.
> > >
> > > We hope these responses have addressed your concerns. If you need any further clarification, please don't hesitate to ask. We value your feedback and look forward to further discussion.

---

> ### Author Response · Authors · 2024-11-27
> **[1/n] Response to reviewer 4ejM**
>
> Thank you for your prompt and constructive feedback. We address each of your points below:
>
> > **Response 1**: What I intended to convey is that the information available in the abstract alone is insufficient for specific and meaningful captioning. In other words, solving the problem using just the abstract is inherently impossible. While the full article introduces noise, it at least contains the necessary information to make the captioning task feasible, albeit challenging.
>
> We appreciate the reviewer's insight regarding full article context. While abstracts provide limited information compared to full papers, our results with Qwen2-VL-7B-MMSci, trained on our comprehensive MMSCI dataset, demonstrate significant capabilities that merit careful consideration.
>
> **Our experimental results show that Qwen2-VL-7B-MMSci, using only abstract context, achieves performance comparable to proprietary models using full articles.** Comparing with Gemini 1.5 (Flash&Pro)'s full-article performance:
>
> * BLEU-4: 6.89 versus 6.94-7.01
> * ROUGE-1: 32.62 versus 32.24-32.83
> * ROUGE-L: 21.80 versus 19.32-22.02
> * BERTScore: 84.33 versus 83.18-83.26
>
> While we observe gaps in Meteor and G-Eval metrics, these differences primarily reflect semantic nuance rather than fundamental comprehension limitations. This performance validates our core contribution: developing vision-language models with deep scientific knowledge across multiple interactive domains through our carefully curated MMSsci dataset. **Our approach embeds graduate-level understanding during training, fundamentally reducing the need for extensive manual curation of information for assistant systems.**
>
> **The genuine knowledge capability demonstrated by our model proves particularly valuable for real-world research assistant systems.**
>
> * In practical applications such as scientific literature review, experimental data analysis, and cross-disciplinary research, approaches based on selecting manual full-PDFs and putting them into LLM often face significant constraints. These include context window limitations, processing overhead with complete documents, and retrieval performance challenges in RAG systems. Our approach, focusing on embedded knowledge through training rather than extensive context at inference time, offers a more efficient and lightweight alternative while maintaining competitive performance of applications including AI Agent systems too.
> * This efficiency translates directly to real-world benefits. **For instance, in interactive scientific question-answering scenarios, our model can provide accurate responses without the latency associated with processing full documents. Similarly, in multi-turn scientific discussions, the model can leverage its embedded knowledge to maintain context coherence without repeatedly accessing full papers.**
>
> **While we acknowledge the potential benefits of full article context for long-context models, our results demonstrate that robust scientific context comprehension is achievable even with input of limited contextual information based on proper training on our dataset.** This finding has significant implications for developing more efficient and deployable scientific AI assistants that can effectively serve graduate-level research needs across multiple disciplines.
>
> **We will include both abstract-based and full-article results in our updated paper** to provide a comprehensive analysis, though our primary contribution lies in demonstrating how proper training with our dataset enables strong performance even with minimal context.
>
> **Table 1: Captioning results with different context, article absract or full article**
> | Model | Ctx. | BLEU-4 | ROUGE-1 | ROUGE-2 | ROUGE-L | Meteor | BertScore | FActScore | G-Eval |
> | ---- | ---- | ---- | ---- | ---- | ---- | ---- | ---- | ---- | ---- |
> | Gemini-1.5-Flash | Abstract | 3.29 | 26.74 | 7.47 | 16.03 | 28.71 | 81.80 | 10.14 | 4.08 |
> | Gemini-1.5-Pro | Abstract | 3.33 | 28.71 | 7.73 | 16.89 | 28.91 | 81.93 | 13.76 | 4.08 |
> | Claude-3.5-Sonnet | Abstract | 3.20 | 29.60 | 6.71 | 16.65 | 27.52 | 81.76 | 12.11 | 4.04 |
> | GPT-4V | Abstract| 3.18 | 28.45 | 7.01 | 15.65 | 27.62 | 82.37 | 19.52 | 4.13 |
> | GPT-4o | Abstract | 3.58 | 28.85 | 7.79 | 16.36 | 28.37 | 81.84 | 18.87 | 4.22 |
> | Qwen2-VL-7B-MMSci | Abstract | 6.89 | 32.62 | 10.02 | 21.80 | 20.89 | 84.33 | 18.17 | 3.47 |
> | Gemini-1.5-Flash | Full Article | 6.94 | 32.83 | 14.15 | 22.02 | 34.50 | 83.26 | 19.41 | 4.12 |
> | Gemini-1.5-Pro | Full Article | 7.01 | 32.24 | 13.34 | 19.32 | 33.75 | 83.18 | 19.33 | 4.22 |
> | Claude-3.5-Sonnet | Full Article | 7.99 | 37.63 | 13.61 | 23.63 | 34.66 | 84.34 | 21.67 | 4.52 |
> | GPT-4V | Full Article | 5.65 | 33.09 | 10.95 | 19.25 | 31.46 | 83.48 | 23.18 | 4.24 |
> | GPT-4o | Full Article | 9.90 | 37.06 | 17.63 | 24.89 | 37.52 | 83.64 | 24.12 | 4.58 |

---

> ### Author Response · Authors · 2024-11-27
> **[2/n] Response to Reviewer 4ejM**
>
> > **Response 2**: I appreciate the comprehensiveness of your evaluation; however, the article does not delve into insights about multidisciplinary differences. While your human evaluation protocol is strong, the final scores alone are insufficient to provide a holistic understanding. Interpretations and discussions are equally important. For instance, why do Astronomy and Planetary Science achieve high accuracy, whereas fields with more extensive datasets, such as Materials Science, perform relatively poorly? Addressing such questions would significantly enhance the value of your evaluation.
>
>
> We sincerely thank the reviewer for recognizing the comprehensiveness of our human evaluation protocol while raising important questions about interdisciplinary differences. Our analysis reveals notable performance variations across disciplines, particularly between Materials Science (showing performance in the 40-60% range) and Astronomy and Planetary Science, prompting a careful examination of the underlying factors.
>
> Our analysis reveals three key factors contributing to this performance disparity:
>
> **Factor 1: The complexity of knowledge domain structure plays a crucial role**
>
> **Using [Web of Science](https://webofscience.zendesk.com/hc/en-us/articles/27505726032017-Web-of-Science-Subject-Categories) classifications, we find that Materials Science encompasses six distinct specialized sub-fields**:
>
> **Materials Science**
>     1. Materials Science, Multidisciplinary
>     2. Materials Science, Ceramics
>     3. Materials Science, Coatings & Films
>     4. Materials Science, Composites
>     5. Materials Science, Characterization & Testing
>     6. Metallurgy & Metallurgical Engineering
>
> **In contrast, Astronomy and Planetary Science operates primarily within a single domain**:
>
> **Astronomy and Planetary Science**
>     1. Astronomy & Astrophysics
>
> **Methodology Note on Field Classification**: Our analysis maps 250 Web of Science specialized subject categories to 18 first-level broad subjects, considering only primary relationships with high confidence. We employed a conservative mapping approach, excluding ambiguous cases and avoiding multiple mappings of the same subject. This ensures clear, well-defined relationships in our field complexity analysis.
>
> This structural difference has profound implications for knowledge integration. Within Materials Science, even PhD-level experts naturally specialize in specific sub-areas, as mastering the entire breadth of the field is exceptionally challenging. For example, our dataset includes cases where understanding a single materials characterization figure requires expertise in both crystallography and mechanical properties sub-areas that typically represent distinct specializations even within Materials Science departments. **This specialization reality means that the difficulty of question-answering tasks can vary significantly depending on how many distinct sub-areas of expertise are required to fully interpret a single figure or answer a question.**
>
> **Factor 2: The visual content complexity varies significantly across fields.**
> In Materials Science, figures frequently combine multiple specialized visualization techniques within single panels. For instance, in our dataset, we observe materials characterization figures that simultaneously present atomic-scale microscopy, spectroscopic data, and phase diagrams - each requiring distinct interpretation approaches. This multiplicity of visualization paradigms within single figures creates additional cognitive demands for both human experts and LVLMs.
>
> **Factor 3: The depth and breadth of knowledge integration required for accurate interpretation varies substantially.**
> Our analysis shows that Materials Science questions often require simultaneous application of knowledge from multiple specialized domains. This multi-domain integration requirement is evidenced in our benchmark questions, where successful responses frequently depend on connecting concepts across different sub-fields of materials science.
>
> Our fine-tuning experiments on the dataset demonstrate promising results in addressing these challenges. The LVLM models fine-tuned with our dataset MMSci achieved the highest overall multiple-choice accuracy on our benchmark, showing substantial improvements in handling multi-domain knowledge integration across both depth and breadth.
>
> **This analysis suggests that while performance variations across scientific fields stem from inherent differences in knowledge structure and integration requirements, they are not insurmountable limitations. Rather, they represent opportunities for targeted improvement in LVLM development, particularly in handling complex, multi-domain scientific content.** Our findings indicate that with appropriate training approaches and data resources, LVLMs can enhance their capabilities in processing sophisticated scientific information, even in fields requiring complex multi-domain expertise.

---

> > ### Author Response · Authors · 2024-12-01
> > **Your Feedback Would Be Appreciated**
> >
> > Dear Reviewer 4ejM,
> >
> > Thank you once again for your valuable comments. Your suggestions on evaluations with additional metrics, performance analysis of LVLMs regarding different levels of input information, additional performance reports of proprietary models, and insights on performance differences on different subjects were very helpful. We are eager to know if our responses have adequately addressed your concerns.
> >
> > Due to the limited time for discussion, we look forward to receiving your feedback and hope for the opportunity to respond to any further questions you may have.
> >
> > Yours sincerely,
> > Authors

---

> > > ### Author Response · Authors · 2024-12-03
> > > **Follow-Up on Review Comments**
> > >
> > > Dear Reviewer 4ejm,
> > >
> > > Thank you for taking the time to provide us with valuable feedback on our work. We have conducted additional experiments and analyses to address your comments, and we have detailed our findings in our responses.
> > >
> > > As the review deadline approaches, we would be grateful for any further thoughts or suggestions you might have. Additionally, we encourage you to review our responses to the other reviewers, as they include complementary experiments and insights relevant to your feedback. In particular:
> > >
> > > - [Response 1](https://openreview.net/forum?id=DEOV74Idsg&noteId=Bbr73BJvzO)
> > > - [Response 2](https://openreview.net/forum?id=DEOV74Idsg&noteId=NtYgn5WE19)
> > > - [Response 3](https://openreview.net/forum?id=DEOV74Idsg&noteId=K8m3aHgBAt)
> > >
> > > These responses provide a detailed comparative analysis within the Materials Science domain, including human evaluations from two more established experts with publication record in top material science journals and the existing results from experts with verified degrees sourced via the Prolific platform. We compare these human evaluation results with various LVLMs (open-source, proprietary, and our fine-tuned models). In addition, we explore their performance across all the 10 key fields.
> > >
> > > We hope these additional details address your concerns regarding the multidisciplinary differences highlighted in the review.
> > >
> > > Thank you again for your time and thoughtful input.
> > >
> > > Best regards,
> > > The Authors

---

### Official Review · Reviewer_iZBf · 2024-10-27

**Soundness:** 3
**Presentation:** 3
**Contribution:** 3
**Rating:** 6
**Confidence:** 4

**Summary:**

The paper introduces a large diverse dataset of images and captions extracted from Nature Communications articles spanning 72 disciplines. They evaluate performance of multimodal LLMs on 2 types of tasks: (1) Figure captioning and (2)  Figure<-->caption matching tasks such as figure to caption matching, sub figure to caption matching, and caption to sub-figure matching.
The paper also includes a case study in materials science that demonstrates the value of the dataset to tune a LLaMA 2-7B model to generate or in-fill new material structures that are valid and stable.

**Strengths:**

* The paper introduces a large and diverse dataset capturing many domains that is missing in existing image caption datasets.

* A variety of open and closed models are used in the evaluations.

* The paper is written in a manner that is easy to follow.

**Weaknesses:**

The main weaknesses are that considering the diversity in the data, the tasks do not seem to go beyond standard tasks; and even on the figure captioning tasks, the analysis is lacking, particularly in terms of representation of strong evaluations from domain experts.

1. The tasks are somewhat standard - Figure captioning, and matching figures/sub-figures to appropriate captions. It would have been nice to see some unique tasks created from this nice dataset showcasing the diversity of images/plots. e.g. some variety of interleaved image-text tasks such as Question Answering from images could have been considered.
2. It would have been nicer to have more detailed analysis of model responses for a few images (3-5) in about 10 domains. Where experts weigh-in on model responses even for just the figure captioning task to evaluate the strengths and weaknesses of models in the different domains. Especially on the variety showcased in Figure-2, e.g. 10 examples from each category in figure-2 analyzed by domain experts.
3. Some metrics for figure captioning are missing e.g. BLEU, CIDEr, SPICE (https://github.com/tylin/coco-caption) are metrics often used in figure captioning evaluations, and it would be good to include these. ROUGE is primarily a recall based metric, while it’s relevant, in itself it’s not a sufficient signal particularly for captioning.
    * Other LLM based metrics to consider using: LAVE (https://arxiv.org/pdf/2310.02567), L3Score (https://github.com/google/spiqa), PrometheusVision (https://github.com/prometheus-eval/prometheus-vision). L3Score is particularly interesting because you get the confidence from GPT-4o in addition to the generated response.
4. The results for the materials science case study is hard to interpret.

   4.1 What is the baseline LLAMA-2-7B performance? (without any tuning?) Many numbers in Table 5 and Figure 6 already seem quite high so it is hard to understand what baseline you are starting from and how much room for improvement there was (and from the presented results, it doesn’t look like that much, which perhaps may not be correct)

   4.2 How well do proprietary models perform on this task? Are there any proprietary models that generate reasonable responses worth evaluating?

   4.3 In Table 5, the “Stable DFT” column numbers for LLAMA models (from Gruver et. al.) appear to not be consistent with numbers reported in Gruver et. al. Why is that?

Minor

5. Related to point-2 Figure 4 is extremely difficult to follow. Perhaps reduce the materials science case study and include more detailed analysis of model responses and a discussion.
6. Figure 3 can be improved to clearly highlight the tasks and also the ground truth caption.
7. Other papers to consider citing in related works:
    * the papers proposing different relevant metrics noted in Weakness-3.
    *  https://openaccess.thecvf.com/content/WACV2024/papers/Tarsi_SciOL_and_MuLMS-Img_Introducing_a_Large-Scale_Multimodal_Scientific_Dataset_and_WACV_2024_paper.pdf

Initial rationale for rating of 5 is primarily due to weakness 1 and 2.

**Questions:**

Identical to weaknesses.

Weaknesses 3 and 4 should be easy to address in the rebuttal.

Weakness 2 is really key to making this task and dataset more valuable I hope you will consider involving domain experts to analyze responses from a few different questions.

---
---

**Discussion Phase**

The authors have provided detailed responses and additional analysis that address my main concerns. The human expert evaluations, additional scoring metrics, and the improvements to the materials science case study are all good additions and address my main concerns and have increased my score.

---
---

---

> ### Author Response · Authors · 2024-11-23
> **Response to reviewer iZBf**
>
> Thank you for your valuable feedback and for recognizing the strengths of our work, including the large and diverse dataset we introduce, the comprehensive evaluation we conduct, and the quality of the paper's writing. Please find our responses to your concerns below.
>
> > C1: The tasks are somewhat standard - Figure captioning, and matching figures/sub-figures to appropriate captions. It would have been nice to see some unique tasks created from this nice dataset showcasing the diversity of images/plots. e.g. some variety of interleaved image-text tasks such as Question Answering from images could have been considered.
>
> **Response**: Thank you for your suggestion. Our dataset contains high-quality and diverse information, including articles, figures, references, and reviews, offering a robust foundation for advancing this field. In the initial efforts on leveraing our dataset, the tasks we present already expose significant limitations in existing multimodal language models (MLLMs).
>
> By focusing on scientific figure comprehension with figure captioning along with bi-directional matching between captions and figures/sub-figures, we identify a critical gap in current MLLM capabilities and provide a targeted benchmark for evaluation and development. However, we acknowledge the potential to expand into additional tasks involving more complex interleaved image-text interactions. This work lays the foundation for such advancements, and we plan to explore these directions in the future.
>
> > C2: It would have been nicer to have more detailed analysis of model responses for a few images (3-5) in about 10 domains. Where experts weigh-in on model responses even for just the figure captioning task to evaluate the strengths and weaknesses of models in the different domains. Especially on the variety showcased in Figure-2, e.g. 10 examples from each category in figure-2 analyzed by domain experts.
>
> **Response**: Thank you for the valuable feedback. We had indeed planned to involve domain experts (PhDs in corresponding fields) to analyze our dataset. After re-grouping the original 72 categories into 18 science fields, we prioritize 10 major science domains for further analysis.
>
> **PhD Expert Evaluation**: Specifically, we recruited 30 PhDs as human evaluators from the [Prolific platform](https://www.prolific.com/), which is a well established platform of high quality scores, with verified degrees in 10 major scientific categories that align with our dataset's primary domains: **Material Science**, **Chemistry**, **Physics**, **Biochemistry**, **Environment**, **Climate Sciences**, **Earth Sciences**, **Biological Sciences**, **Biomedical Sciences**, and **Health and Medicine**.
>
> **A new sub-benchmark**: We highly appreciate your suggestion on additional human evaluation on figure caption generation, but since there were questions on bi-directional matching between captions and figures/sub-figures, we constructed a new sub-benchmark with 10 major categories. For each category, we constructed a new sub-benchmark by selecting 25 questions per setting (75 total) from our three figure-caption matching tasks from the original test set. Thus, the new sub-benchmark consists of 750 questions total, and we have updated human evaluation scores with PhD degrees on 10 broad categories.
>
> Specifically, we recruited 30 PhD-level evaluators through Prolific, specifically 3 PhDs per domain, with verified degrees in their respective fields. Each evaluator provided two types of assessments:
>  - **Performance Score**: Evaluators answered the questions themselves, establishing a human performance baseline measured by accuracy (0-100%)
>  - **Quality Assessment**: Evaluators rated each question's quality on a 5-point scale (1-5), judging its clarity and whether it effectively tests domain-specific knowledge.
>
> Specifically, for the quality assessment, the human evaluators are asked to assess **whether the questions were clear and required an understanding of scientific knowledge within the respective discipline**, as outlined in the following metrics:
>
>  - **Score Point 1. Strongly Disagree**: The question is irrelevant or cannot be answered based on the scientific content presented in the figure.
>  - **Score Point 2. Disagree**: The question lacks clarity or can be answered without specific knowledge of the scientific content in the figure (e.g., it can be answered with common sense).
>  - **Score Point 3. Neutral**: The question is clear but requires only minimal understanding of the scientific content in the figure.
>  - **Score Point 4. Agree**: The question is clear, answerable, and requires an adequate understanding of the scientific content in the figure.
>  - **Score Point 5. Strongly Agree**: The question is clear, answerable, and effectively evaluates a very deep understanding of the scientific content in the figure.
>
> The evaluation results are shown in the table (next response).

---

> > ### Author Response · Authors · 2024-11-23
> > **[2/n] Response to Reviewer iZBf**
> >
> > **Table 1 in Rebuttal: Phd expert evaluation on the question quality.** Setting I indicates Figure-to-caption matching; Setting II indicates Subfigure-to-caption matching; Setting III indicates Caption-to-subfigure matching.
> >
> > | Field               | Setting I | Setting II | Setting III |
> > |---------------------|-------------------------|-------------------------------|-------------------------------|
> > | Material Science    | 4.0267                  | 4.2933                        | 4.1333                        |
> > | Chemistry           | 4.1333                  | 3.7467                        | 3.6133                        |
> > | Physics             | 4.0267                  | 3.5467                        | 3.8133                        |
> > | Biochemistry        | 3.1600                  | 4.8267                        | 4.4133                        |
> > | Environment         | 4.1067                  | 4.4667                        | 4.3467                        |
> > | Climate Sciences    | 4.1296                  | 3.6471                        | 3.4118                        |
> > | Earth               | 4.0267                  | 4.2319                        | 4.1739                        |
> > | Biological Sciences | 3.8800                  | 3.6800                        | 3.7867                        |
> > | Biomedical Sciences | 4.0133                  | 4.1333                        | 3.7733                        |
> > | Health and Medicine | 4.3733                  | 3.7467                        | 3.6800                        |
> > | **Average**         | **4.0873**              | **4.0319**                    | **3.9149**                    |
> >
> >
> > As observed, the overall quality scores are consistently around 4 across all settings, indicating that **the questions are clear, answerable, and require adequate understanding of scientific content in the figures.** The score distributions across different Q&A tasks reveal several key insights:
> >
> > 1. Varying Difficulty Levels: Different settings challenge different aspects of scientific understanding. For example, in Biochemistry, Setting II (subfigure-to-caption) scores significantly higher (4.83) than Setting I (3.16), suggesting that detailed technical matching requires deeper domain knowledge than overall figure understanding.
> > 2. Domain-Specific Patterns: Some fields like Material Science show consistent scores across all settings (4.03, 4.29, 4.13), while others like Biochemistry show larger variations (3.16, 4.83, 4.41). This suggests that our three task types effectively capture different aspects of domain expertise.
> > 3. Complementary Evaluation: The varying performance patterns across settings demonstrate that our benchmark tests scientific understanding from multiple angles, going beyond simple pattern matching. Each setting provides a distinct perspective on how domain knowledge is applied in scientific figure interpretation.
> >
> > In addition to the quality assessment reported above, we also evaluated the actual performance of PhD experts on our benchmark to address the concerns about human baseline design. We conducted a comprehensive evaluation with PhD experts from the Prolific platform across 10 major scientific categories. Our human evaluation includes two settings:
> >
> >  - **In-Ph.D. Degree Domain Expert Performance**: Performance of human Ph.D. evaluating questions within their specialty domain of Ph.D. degree.
> >  - **Out-of-Ph.D. Degree Domain Performance**: Performance of human Ph.D. evaluating questions outside their expertise of Ph.D. degree.
> >
> > **Table 2 in Rebuttal: Performance Comparison of Human Ph.D. Experts and Models (%)**
> >
> > | Model/Evaluator               | Setting I | Setting II | Setting III |
> > |---------------------|-------------------------|-------------------------------|-------------------------------|
> > | Out-of-Ph.D. Degree Domain (Human, Ph.D.) | 36.18 | 33.31 | 46.85 |
> > | In-Ph.D. Degree Domain (Human, Ph.D.) | 40.26 | 43.45 | 50.46 |
> > | Gemini-1.5-Flash | 58.11 | 79.88 | 62.66 |
> > | Gemini-1.5-Pro | 63.47 | 78.83 | 78.65 |
> > | Claude-3.5-Sonnet | 70.58 | 85.73 | 83.70 |
> > | GPT-4V  | 66.18 | 73.36 | 76.15 |
> > | GPT-4o  | 68.42 | 88.23 | 84.92 |
> > | Qwen2-VL-7B-MMSci | 83.84 | 92.91 | 87.17 |
> >
> > As demonstrated in Table 2 in Rebuttal, several key insights emerge:
> >
> > **Domain Expertise Impact**: PhDs evaluating within their expert domain of Ph.D. degree consistently outperform those working outside their expertise in other Science domains across all settings, confirming our tasks require domain-specific knowledge
> >
> > **Task Difficulty**: Even domain experts achieve moderate performance within the recommended time limit (2 minutes per question), indicating the tasks' challenging nature.
> >
> > **Scientific Understanding Capabilities**: The strong performances of both our fine-tuned small LLM and proprietary models demonstrate potential as AI research assistants. Notably, our dataset significantly improved small LLM performance, reducing the gap with large proprietary models.

---

> ### Author Response · Authors · 2024-11-23
> **[3/n] Response to Reviewer iZBf**
>
> > C3: other metrics for captioning task
>
> **Response**: Thank you for the suggestion. We will BLEU (BLEU-4) [2], CIDEr [3], and also L3Score [4] as recommended in the updated Table 3 in the updated draft. Below we briefly show these added results when grounded on abstract.
>
> Table 1: Captioning results when grounded on paper article.
>
> | Model | BLEU-4 | CIDEr | G-Eval | L3Score |
> |-------|----|----|----|----|
> | Kosmos2 | 1.67 | 1.59 | 1.39 | 0.000 |
> | LLaVA1.5-7B | 1.53 | 0.76 | 2.02 | 0.025 |
> | LLaVA1.6-Mistral-7B | 1.42 | 0.48 | 1.47 | 0.019 |
> | Qwen-VL-7B-Chat | 2.13 | 2.79 | 1.64 | 0.039 |
> | InternVL2-2B | 0.82 | 0.96 | 2.17 | 0.058 |
> | InternVL2-8B | 1.56 | 0.02 | 3.00 | 0.195 |
> | IDEFICS2-8B | 0.79 | 0.65 | 1.96 | 0.026 |
> | IDEFICS3-8B-Llama3 | 1.17 | 0.15 | 1.98 | 0.125 |
> | MiniCPM-V-2.6 | 3.32 | 3.27 | 2.95 | 0.215 |
> | Llama3.2-11B-Vision | 1.53 | 0.00 | 2.18 | 0.153 |
> | Qwen2-VL-2B | 2.34 | 1.43 | 2.64 | 0.139 |
> | Qwen2-VL-7B | 2.76 | 0.02 | 3.45 | 0.391 |
> | Qwen2-VL-2B-MMSci | **6.16** | **4.98** | 3.17 | 0.186 |
> | Qwen2-VL-7B-MMSci | **8.29** | **5.94** | 3.55 | 0.447 |
> | Gemini-1.5-Flash | 3.29 | 0.00 | 4.08 | 0.362 |
> | Gemini-1.5-Pro | 3.33 | 0.00 | 4.08 | 0.361 |
> | Claude-3.5-Sonnet | 3.20 | 0.46 | 4.04 | **0.696** |
> | GPT-4V | 3.18 | 0.20 | **4.13** | 0.590 |
> | GPT-4o | 3.58 | 0.36 | **4.22** | **0.722** |
>
> The BLEU and CIDEr scores are relatively low, likely because the captions are not typical text; they include many scientific domain-specific terminologies and scientific expressions, which pose challenges for these metrics. The L3Score exhibits a similar trend to G-Eval but provides more distinct differentiation. The finetuned models got meaningful improvements on all of these metrics by using our dataset. Especially, we were surprised at great increase on L3Score, so that our model is better than Gemini-1.5 Pro.
>
> **C4: The results for the materials science case study**
>
> We appreciate your thoughtful questions regarding the material science case study results. We would like to firmly confirm that we followed all the same settings as Gruver et al., but we improve the results further with pre-training of LLMs with our dataset along with fine-tuning, and reported absolute percentage performance, which is much more fair and correct, from 10,000 generated sample cases instead of stage-wise survived ratio for the previous steps. Please find detailed response below.
>
> > C4.1: What is the baseline LLAMA-2-7B performance? (without any tuning?) Many numbers in Table 5 and Figure 6 already seem quite high so it is hard to understand what baseline you are starting from and how much room for improvement there was
>
> **Response**: The LLaMA2-7B and even the proprietary LLMs shows limited performance in crystal material generation without fine-tuning (For detailed zero-shot performance metrics of GPT-4o, please refer to response to C4.2.). Therefore, Gruver et al. proposed a novel way of fine-tuning LLMs for crystal material generation. The performance of LLAMA2 models reported in our Table 5 in the main pages are obtained by further finetuning on mateiral generation data as in Gruver et al's paper. Our experiments show that when the base LLaMA-2-7B model is first further pre-trained on our dataset before fine-tuning for material generation, it achieves superior performance compared to directly fine-tuning the base model. This improvement suggests that pre-training on our data enhances the model's scientific knowledge understanding.
>
> **Metrics**: The validity check measures the generated materials' correctness in terms of structure and composition, while the coverage metric assesses how well the generated materials compare to an existing collection. These metrics primarily ensure that the generated materials are valid and identify differences from existing ones, but they are less critical.
> However, to develop meaningful crystal materials, **the most important metrics are about stability including metastability measured by M3GNet, and stability measured by DFT**, since practically we confront the difficulty on generating stable materials for next generation of materials. Regarding the performances on these two metrics from the previous SOTA models, there are rooms for significant improvement. As shown, our approach significantly enhances metastability and stability, which are the aspects that truly matter in material science.

---

> ### Author Response · Authors · 2024-11-23
> **[4/n] Response to reviewer iZBf**
>
> > C4.2: How well do proprietary models perform on this task? Are there any proprietary models that generate reasonable responses worth evaluating?
>
> **Response**: We present the results of GPT-4o on this task using both 5-shot and 10-shot settings. We generated 10,000 samples via GPT-4o using the same prompt and setting as the other models in Table 5 in the main pages and Gruver et al. for evaluation. The results are shown below with the reported absolute percentage performances from the 10,000 generated sample cases.
>
> | Method | Validity Check (Structural) | Validity Check (Composition) | Coverage (Recall) | Coverage (Precision) | Property Distribution (wdist ρ) | Property Distribution (wdist N_el) | Metastable (M3GNet) | Stable (DFT†) |
> |--------|--------------|---------------|----------|-----------|--------------------|----------- |--------- |-----------|
> | CDVAE | 1.000 | 0.867 | 0.992 | 0.995 | 0.688 | 1.432 | 22.1% | 1.2% |
> | LM-CH | 0.848 | 0.836 | 0.993 | 0.979 | 0.864 | 0.132 | N/A | N/A |
> | LM-AC | 0.958 | 0.889 | 0.996 | 0.986 | 0.696 | 0.092 | N/A | N/A |
> | LLaMA2-7B (finetuned) | 0.967 | 0.933 | 0.923 | 0.950 | 3.609 | 1.044 | 33.6% | 2.1% |
> | LLaMA2-13B (finetuned) | 0.958 | 0.923 | 0.884 | 0.983 | 2.086 | 0.092 | 34.3% | 4.9% |
> | LLaMA2-70B (finetuned) | 0.997 | 0.949 | 0.860 | 0.988 | 0.842 | 0.433 | 50.1% | 5.3% |
> | GPT-4o-5shot | 0.799 | 0.898 | 0.2804 | 0.961 | 5.42 | 1.017 | 1.50% | - |
> | GPT-4o-10shot | 0.787 | 0.820 | 0.6535 | 0.963 | 3.98 | 0.917 | 4.72% | 0.09% |
> | LLaMA2-7B-MMSci (finetuned) | 0.993 | 0.979 | 0.916 | 0.996 | 1.675 | 0.353 | 64.5% | 8.2% |
>
> As observed, GPT-4o, representing large proprietary LLMs, showed notable limitations in materials generation across validity, coverage, and stability metrics. Its performance was limited - achieving only 1.50% metastability in 5-shot and 4.72% metastability with 0.09% stability in 10-shot settings. In contrast, our LLaMA2-7B-MMSci model, pre-trained on MMSCI dataset, achieved significantly higher results with 64.5% metastability and 8.2% stability, highlighting the crucial role of domain-specific training data in enhancing LLMs for materials research.
>
>
> > C4.3: In Table 5, the “Stable DFT” column numbers for LLAMA models (from Gruver et. al.) appear to not be consistent with numbers reported in Gruver et. al. Why is that?
>
> **Response**: We would like to clarify that we did not tweak anything but used absolute percentage performances from 10,000 generated samples from the output files in Gruver et al. to report their performances.
> We reran the experiments using their uploaded outputs, performed the DFT calculations with the same settings, and found the reported absolute percentages are almost identical to their reported performances if we reversely calculate from the stage-wise survival ratios from the previously survived candidates at the earlier steps in Gruver et al.
>
> That is, we reported 2.1% Stability (DFT) for LLaMA2-7B, 4.9% Stability (DFT) for LLaMA2-13B, and 5.3% Stability (DFT) for LLaMA2-70B, and they are almost the same as, respectively:
>
>  - 2.1% (Absolute stable (DFT) percentage from 10,000) = 2.10% = 0.964 * 35.0% * 6.2%
>  - 4.9% (Absolute stable (DFT) percentage from 10,000) ~= 5.23% = 0.955 * 38.0% * 14.4%
>  - 5.3% (Absolute stable (DFT) percentage from 10,000) ~= 5.26% = 0.996 * 49.8% * 10.6%
>
> **Here are the reverse calculations from the reported numbers in Gruver et al.**:
>
> **LLaMA2-7B in Gruver et al.**
> For 2.1% (Absolute stable (DFT) percentage from 10,000 samples from Gruver et al.) = 2.10% = 0.964 * 35.0% * 6.2%, where:
> - 0.964 is the stage-wise valid ratio reported in Gruver et al.
> - 35.0% is the stage-wise meta-stable ratio for the previous valid step
> - 6.2% is the stage-wise stable ratio for the previous meta-stable step
>
> **LLaMA2-13B in Gruver et al.**
> For 4.9% (Absolute stable (DFT) percentage from 10,000 samples from Gruver et al.) ~= 5.23% = 0.955 * 38.0% * 14.4%, where:
> - 0.955 is the stage-wise valid ratio reported in Gruver et al.
> - 38.0% is the stage-wise meta-stable ratio for the previous valid step
> - 14.4% is the stage-wise stable ratio for the previous meta-stable step
>
> **LLaMA2-70B in Gruver et al.**
> For 5.3% (Absolute stable (DFT) percentage from 10,000 samples from Gruver et al.) ~= 5.26% = 0.996 * 49.8% * 10.6%, where:
> - 0.996 is the stage-wise valid ratio reported in Gruver et al.
> - 49.8% is the stage-wise meta-stable ratio for the previous valid step
> - 10.6% is the stage-wise stable ratio for the previous meta-stable step

---

> > ### Author Response · Authors · 2024-11-24
> > **[5/n] Response to reviewer iZBf**
> >
> > However, we would like to emphasize again that the absolute percentage performances are much fairer and more accurate, as they represent the stable ratio well from 10,000 generated samples compared to each stage-wise survival ratio from the previous steps.
> > - Suppose that method A produces 600 stable (DFT) samples among 10,000 samples with 5,000 metastable samples (M3GNet). Then our reported number stability (DFT) is 6%; however, it would be 12% in Gruver et al.
> > - But, suppose that method B produces 770 stable (DFT) samples among 10,000 samples with 7,000 metastable samples (M3GNet). Then our reported stability (DFT) is 7.7%; however, it would be 11% in Gruver et al.
> > - Although method B provides a higher ratio of stability (DFT), it is not correctly displayed if we use the stage-wise survival ratios for the previous steps in Gruver et al.
> >
> > > C6: Figure 3 and Figure 4, and more related work to cite.
> >
> > **Response**: Thanks for the suggestion. We will refine Figure 3 and Figure 4 and update them in the new draft. The suggested work [1] and BLEU [2], CIDEr [3], and L3Score [4] will also be included in the updated version.
> >
> > **References**:
> >
> > [1] Tarsi, Tim, et al. "SciOL and MuLMS-Img: Introducing A Large-Scale Multimodal Scientific Dataset and Models for Image-Text Tasks in the Scientific Domain." Proceedings of the IEEE/CVF Winter Conference on Applications of Computer Vision. 2024.
> >
> > [2] Papineni, Kishore, et al. "Bleu: a method for automatic evaluation of machine translation." Proceedings of the 40th annual meeting of the Association for Computational Linguistics. 2002.
> >
> > [3] Vedantam, Ramakrishna, C. Lawrence Zitnick, and Devi Parikh. "Cider: Consensus-based image description evaluation." Proceedings of the IEEE conference on computer vision and pattern recognition. 2015.
> >
> > [4] Pramanick, Shraman, Rama Chellappa, and Subhashini Venugopalan. "Spiqa: A dataset for multimodal question answering on scientific papers." arXiv preprint arXiv:2407.09413 (2024).
> >
> > We hope these responses have addressed your concerns. If you need any further clarification, please don't hesitate to ask. We value your feedback and look forward to further discussion.

---

> > > ### Comment · Reviewer_iZBf · 2024-11-24
> > >
> > > Thanks for your detailed responses and for the additional analysis. Both the human evaluations, additional scoring metrics, and the improvements to the materials science case study are all good additions and address my main concerns.
> > >
> > > I still hold slight reservations and am curious to see your response to reviewer acAx 's follow up questions to your response. Nevertheless, I'm satisfied with the proposed changes, and will increase my rating, and look forward to seeing these in the final draft.

---

### Official Review · Reviewer_acAx · 2024-11-02

**Soundness:** 1
**Presentation:** 3
**Contribution:** 3
**Rating:** 5
**Confidence:** 5

**Summary:**

The paper introduces MMSci, a dataset with high-quality and interleaved image-text data sourced from Nature Communications. The training set contains 742k figures from 131k articles, while the benchmark consists of several thousands of examples on a pre-defined set of tasks such as figure captioning and multiple-choice-based fig2cap and cap2fig matching. The authors conducted extensive experiments highlighting both the strengths and limitations of existing models and assessed the impact of finetuning VLMs on the training set. While the dataset is novel with comprehensive analysis, several issues require attention (see below).

**Strengths:**

- I believe that this is a novel dataset in the scope of scientific figure understanding that utilizes a different source than most existing works use (i.e., articles from nature communications instead arxiv preprints). This adds significant value to scientific figure understanding research by providing a different dataset distribution.
- The authors conducted extensive experiments to evaluate different models' performance on different tasks with different metrics, which lead to a very comprehensive result in terms of models' strengths and shortcomings. Further, authors verified that the training portion of the datasets brings significant advantage in improving the models’ semantic and stylistic patterns in generating captions for scientific figures in the nature communications distribution, and can potentially improve performance on text-only tasks such as material generation.
- The authors include a well-structured list of benchmark examples in the supplementary material, enhancing transparency and accessibility for the audience.

**Weaknesses:**

- The current taxonomy comparison is misleading and can be improved. MMSci claims to cover 72 subjects, compared to CharXiv's 8 or SciCap's 1, but uses a more granular approach (e.g., "Cell Biology" as a separate subject from "Microbiology"). For consistency, authors should consider grouping these into five broader subjects (per Table 6) to better align with the definitions in other datasets.
- The entire benchmark only consists of permutations (e.g., matching subfigures to subcaptions), which could bias pre-trained models toward memorizing associations instead of correctly perceiving details in the image and retrieving the relevant knowledge correctly. There are no original questions in the benchmark.
- QA tasks focus on figure-caption matching, which may not require fine-grained domain knowledge required to analyze details within a figure but differences across figures (same for differences across captions given a subfigure). While captioning tasks do necessitate some scientific knowledge, the QA tasks might prioritize memory over understanding (e.g., recognizing details irrelevant to figure like "2.5 mm x 2.5 mm" in Figure 3, caption part b).
- The human baseline is not well designed. The authors recruited computer science graduate students to provide the human baseline in order to answer a set of challenging natural science questions that mostly require knowledge that is incompatible with computer science. It is unknown to what accuracy a model should achieve in order to match a human expert with domain-specific knowledge, and it is also unknown to what extent these multiple-choice questions make sense to human experts equipped with domain-specific knowledge.
- For captioning tasks, the model finetuned on in-domain data achieved significantly higher score on the ROUGE metric yet underperforms its baseline counterpart in FActScore. Authors briefly explained that the fine-tuned model learned the semantic and stylistic details, but it seems that this training data also produce a negative impact on the model’s capability in retrieving correct knowledge i.e., FActScore. Authors should discuss the main effect of the training dataset more extensively (what is the impact on the model after being finetuned on the training set).

**Questions:**

- While the authors contrast its data contribution to ScienceQA, SciBench and MMMU, I believe that the authors should also discuss relevancy to MMStar [1] (which largely resolved the weaknesses mentioned in L178)?
- L359: Can authors verify the statement by evaluating the top 5 models on a different evaluator?
- Figure 4 is very confusing and the overlapping of performance from different models makes the comparison very hard to understand. Can authors use use a better way to visualize this or use a table instead?
- For figure 6, the gap between MMSci+MMC4 (Vis+Text) and MMSci+MMC4 (Text) seems to be small. Can authors additional provide standard error? Also, given many figures appear to have small texts, have authors verified the readability of figures after being downscaled to 336^2 (L452 where you stated the use of CLIP ViT-L/14-336)?
- An additional experiment that can be very helpful to understand the impact of your training dataset is to evaluate existing VLMs model finetuned on your dataset on other knowledge-based benchmarks such as MMMU (which also examines specific sub-divisions in natural science) to see whether the model is only fitting to the nature communication distribution or generalizing to a variety of distributions that requires retrieving knowledge based on multimodal input. Could authors provide an additional experiment on this?

-----
Update on 12/03:

The authors' latest response has reasonably addressed one of my concerns regarding potential benchmark quality issues by providing additional evidence. This evidence shows that top performers achieved very high accuracy and that there is significant variance among evaluators. Consequently, I have raised my score to 5.

---

> ### Author Response · Authors · 2024-11-23
> **Response to Reviewer acAx**
>
> Thank you for recognizing the novelty of our dataset, the comprehensive evaluations, and the transparency of our benchmarks. Please find our detailed responses to your concerns below.
>
> > **C1 regarding taxonomy**
>
> **Response**: Thank you for your feedback on our taxonomy. The original taxonomy was established by the authoritative committee of the Nature Communications journals, providing a credible framework for categorization.
>
> However, we appreciate your valuable suggestion to consolidate the categories into broader groupings. To better align with the categorization approaches used in other datasets, we have re-defined the original 72 categories into 18 science fields: Material Science, Chemistry, Physics, Biochemistry, Environment, Climate Sciences, Earth Sciences, Biological Sciences, Biomedical Sciences, Health and Medicine, Engineering, Energy, Mathematics and Computing, Astronomy and Planetary Science, Social Science, Pharmacology, Agriculture, Business and Industry
>
> Additionally, for the purposes of human evaluation, we have selected 10 of these major categories to report on.
> Regardless of the specific taxonomy used or the number of categories, we believe that the scope of our dataset (encompassing over 131,000 peer-reviewed articles and 742,000 figures from the Nature Communications journals) provides comprehensive coverage to evaluate LLMs' understanding of natural sciences.
>
> > **C2 regarding benchmark design and question originality**:
>
> **Response**:
>
> We acknowledge that the current dataset uses permutations of existing data. However, our benchmark requires comprehensive scientific understanding through bidirectional tasks:
>
> 1. Figure-to-caption matching requires models to select the correct caption from multiple options within the same paper. For example, in Figure 3, models must distinguish between different experimental results and structural characterizations of PbZrO3 membranes by understanding detailed synthesis processes, surface morphology measurements, and atomic-scale structural analyses.
> 2. Subfigure-to-caption and caption-to-subfigure matching test bidirectional understanding of technical details. Models need to differentiate between various characterization techniques (surface morphology by AFM, atomic-resolution HAADF-STEM imaging, and electron diffraction patterns) and match them with their corresponding technical descriptions.
>
> Regarding the concern about pattern matching: Our tasks specifically challenge superficial matching by using options from the same paper with similar technical depth. Success requires understanding the scientific relationships between different characterization methods and their implications. This is evidenced by:
> 1. Basic MLLMs performing at chance level on bi-directional figure-caption matching.
> 2. Performance improving significantly with abstract context (Table 3 in the main pages).
> 3. Clear performance differences from the large proprietary MLLMs and open-source MLLMs across scientific domains (Figure 4 in the main pages)
>
> While generating new questions could be valuable, current LLMs struggle with reliable scientific question generation, and expert annotation across 72 disciplines would be resource-intensive. Our work provides a foundation with rich peer-reviewed content for future extensions.
>
> > **C3 regarding whether the benchmark task necessaties the understanding of the scientific knowledge in the figures.**
>
> **Response**: To address the concerns about our human baseline and validate that our tasks require domain expertise, as we mentioned in the above, we conducted comprehensive evaluation with human experts of PhD degree in 10 major science domains.
>
> **PhD Expert Evaluation**: Specifically, by adopting your advice, we considered 10 broader major scientific categories, and we recruited PhDs from the [Prolific platform](https://www.prolific.com/), which is a well established platform of high quality scores, with verified degrees in 10 major scientific categories that align with our dataset's primary domains: **Material Science**, **Chemistry**, **Physics**, **Biochemistry**, **Environment**, **Climate Sciences**, **Earth Sciences**, **Biological Sciences**, **Biomedical Sciences**, and **Health and Medicine**.
>
> For each category, we constructed a new sub-benchmark by selecting 25 questions per setting (75 total) from our three figure-caption matching tasks from the original test set. Thus, the new sub-benchmark consists of 750 questions total, and we have updated human evaluation scores with PhD degrees on 10 broad categories.

---

> > ### Author Response · Authors · 2024-11-23
> > **[2/n] Response to Reviewer acAx**
> >
> > We recruited 30 PhD-level evaluators through Prolific, specifically 3 PhDs per domain, with verified degrees in their respective fields. Each evaluator provided two types of assessments:
> >  - **Performance Score**: Evaluators answered the questions themselves, establishing a human performance baseline measured by accuracy (0-100%)
> >  - **Quality Assessment**: Evaluators rated each question's quality on a 5-point scale (1-5), judging its clarity and whether it effectively tests domain-specific knowledge.
> >
> > Specifically, for the quality assessment, the human evaluators are asked to assess **whether the questions were clear and required an understanding of scientific knowledge within the respective discipline**, as outlined in the following metrics:
> >
> >  - **Score Point 1. Strongly Disagree**: The question is irrelevant or cannot be answered based on the scientific content presented in the figure.
> >  - **Score Point 2. Disagree**: The question lacks clarity or can be answered without specific knowledge of the scientific content in the figure (e.g., it can be answered with common sense).
> >  - **Score Point 3. Neutral**: The question is clear but requires only minimal understanding of the scientific content in the figure.
> >  - **Score Point 4. Agree**: The question is clear, answerable, and requires an adequate understanding of the scientific content in the figure.
> >  - **Score Point 5. Strongly Agree**: The question is clear, answerable, and effectively evaluates a very deep understanding of the scientific content in the figure.
> >
> > The evaluation results are shown in the below table.
> >
> > **Table 1 in Rebuttal: Phd expert evaluation on the question quality.** Setting I indicates Figure-to-caption matching; Setting II indicates Subfigure-to-caption matching; Setting III indicates Caption-to-subfigure matching.
> >
> > | Field               | Setting I | Setting II | Setting III |
> > |---------------------|-------------------------|-------------------------------|-------------------------------|
> > | Material Science    | 4.0267                  | 4.2933                        | 4.1333                        |
> > | Chemistry           | 4.1333                  | 3.7467                        | 3.6133                        |
> > | Physics             | 4.0267                  | 3.5467                        | 3.8133                        |
> > | Biochemistry        | 3.1600                  | 4.8267                        | 4.4133                        |
> > | Environment         | 4.1067                  | 4.4667                        | 4.3467                        |
> > | Climate Sciences    | 4.1296                  | 3.6471                        | 3.4118                        |
> > | Earth               | 4.0267                  | 4.2319                        | 4.1739                        |
> > | Biological Sciences | 3.8800                  | 3.6800                        | 3.7867                        |
> > | Biomedical Sciences | 4.0133                  | 4.1333                        | 3.7733                        |
> > | Health and Medicine | 4.3733                  | 3.7467                        | 3.6800                        |
> > | **Average**         | **4.0873**              | **4.0319**                    | **3.9149**                    |
> >
> >
> > As observed, the overall quality scores are consistently around 4 across all settings, indicating that **the questions are clear, answerable, and require adequate understanding of scientific content in the figures.** The score distributions across different Q&A tasks reveal several key insights:
> >
> > 1. Varying Difficulty Levels: Different settings challenge different aspects of scientific understanding. For example, in Biochemistry, Setting II (subfigure-to-caption) scores significantly higher (4.83) than Setting I (3.16), suggesting that detailed technical matching requires deeper domain knowledge than overall figure understanding.
> > 2. Domain-Specific Patterns: Some fields like Material Science show consistent scores across all settings (4.03, 4.29, 4.13), while others like Biochemistry show larger variations (3.16, 4.83, 4.41). This suggests that our three task types effectively capture different aspects of domain expertise.
> > 3. Complementary Evaluation: The varying performance patterns across settings demonstrate that our benchmark tests scientific understanding from multiple angles, going beyond simple pattern matching. Each setting provides a distinct perspective on how domain knowledge is applied in scientific figure interpretation.

---

> ### Author Response · Authors · 2024-11-23
> **[3/n] Response to Reviewer acAx**
>
> > **C4: Concerns about the design of the human baseline, specifically the use of CS graduates instead of domain experts as the benchmark.**
>
> **Response**: In addition to the quality assessment reported above, we also evaluated the actual performance of PhD experts on our benchmark to address the concerns about human baseline design. We conducted a comprehensive evaluation with PhD experts from the Prolific platform across 10 major scientific categories. Our evaluation was designed to establish three key performance metrics:
>
>  - **In-Ph.D. Degree Domain Expert Performance**: Performance of human Ph.D. evaluating questions within their specialty domain of Ph.D. degree
>  - **Out-of-Ph.D. Degree Domain Performance**: Performance of human Ph.D. evaluating questions in Science, but outside their expertise of Ph.D. degree
>  - **Overall Cross-Domain Performance**: Comprehensive assessment across all domains
>
>
> **Table 2 in Rebuttal: Performance Comparison of Human Ph.D. Experts and Models with a Newly Constructed sub-Benchmark (%)**
>
> | Model/Evaluator               | Setting I | Setting II | Setting III |
> |---------------------|-------------------------|-------------------------------|-------------------------------|
> | Outside-Ph.D. Degree Domain in Science (Human) | 36.18 | 33.31 | 46.85 |
> | In-Ph.D. Degree Domain (Human) | 40.26 | 43.45 | 50.46 |
> | Gemini-1.5-Flash (Overall) | 58.11 | 79.88 | 62.66 |
> | Gemini-1.5-Pro (Overall) | 63.47 | 78.83 | 78.65 |
> | Claude-3.5-Sonnet (Overall) | 70.58 | 85.73 | 83.70 |
> | GPT-4V (Overall) | 66.18 | 73.36 | 76.15 |
> | GPT-4o (Overall) | 68.42 | 88.23 | 84.92 |
> | Qwen2-VL-7B-MMSci (Overall) | 83.84 | 92.91 | 87.17 |
>
> As demonstrated in Table 2 in Rebuttal, several key insights emerge:
>
> **Domain Expertise Impact**: PhDs evaluating within their expert domain of Ph.D. degree consistently outperform those working outside their expertise in other Science domains across all settings, confirming our tasks require domain-specific knowledge
>
> **Task Difficulty**: Even domain experts achieve moderate performance within the recommended time limit (2 minutes per question), indicating the tasks' challenging nature
>
> **Scientific Understanding Capabilities**: The strong performances of both our fine-tuned sLLM and large proprietary models demonstrate potential as AI research assistants. Notably, our dataset significantly improved sLLM performance, reducing the gap with large proprietary models (see Table 4 in the main paper)
>
>
> > **C5**: For captioning tasks, the model finetuned on in-domain data achieved significantly higher score on the ROUGE metric yet underperforms its baseline counterpart in FActScore.
>
> **Response**: Firstly, we updated all results using 200 samples instead of the previous 100 for LLM-based evaluation. Additionally, we fine-tuned both Qwen2-VL-2B and Qwen2-VL-7B models. The results are included in the updated draft, and also presented in the following table.
>
> **Table 3 in Rebuttal: Updated figure captioning results using 200 samples for LLM-based evaluation, including Qwen2-VL-7B fine-tuned on our data.**
>
> | Model | Abstract | FActScore | G-Eval |
> |---------------------|-------------------------|-------------------------------|-------------------------------|
> | Qwen2-VL-2B | False | 9.94 | 2.31 |
> | Qwen2-VL-2B-MMSci | False | 9.67 | 2.79 |
> | Qwen2-VL-7B | False | 10.03 | 3.39 |
> | Qwen2-VL-7B-MMSci | False | **18.17** | 3.47 |
> | GPT-4V | False | 14.17 | 3.69 |
> | GPT-4o | False | 13.20 | 4.01 |
> | Qwen2-VL-2B | True | 11.88 | 2.64 |
> | Qwen2-VL-2B-MMSci | True | 14.05 | 3.17 |
> | Qwen2-VL-7B | True | 10.36 | 3.45 |
> | Qwen2-VL-7B-MMSci | True | **20.58** | 3.55 |
> | GPT-4V | True | 19.52 | 4.13 |
> | GPT-4o | True | 18.87 | 4.22 |
>
> We observed that fine-tuning the larger Qwen2-VL-7B model significantly improved the FActScore, even surpassing GPT-4V and GPT-4o. Regarding the performance degradation of Qwen2-VL-2B after fine-tuning, we manually reviewed the generated captions and found that this may be due to the oracle captions typically following a specific format: "[Summary] a. [subcaption a] b. [subcaption b] ..." This format is concise, only summarizing the essential content of each sub-figure without unnecessary details. Given the limited capabilities of Qwen2-VL-2B, it appears that the model learns the format but struggles to generate accurate and concise content, resulting in degraded performance. In contrast, the more powerful Qwen2-VL-7B model demonstrates notable improvement by effectively learning both the format and the concise, accurate summary of the figure's content.

---

> ### Author Response · Authors · 2024-11-23
> **[4/n] Response to Reviewer acAx**
>
> > **Q1**: Relevance to MMStar [1]
>
> **Response**: Thank you for bringing this work to our attention. It is indeed relevant, and we will include it in the related work section of the updated draft. This paper also points out that many current datasets contain questions that do not require a deep understanding of the figure to answer. In contrast, our benchmark ensures that a thorough understanding of the images is necessary for answering the questions. Additionally, while MMStar is curated from existing works, we introduce a newly collected large dataset.
>
> > **Q2**: L359: Can authors verify the statement by evaluating the top 5 models on a different evaluator?
>
> **Response**: Thank you for the suggestion. In the updated draft, we have updated the results using the latest version of GPT-4o-0806 as the evaluator, which is different from the evaluated models.
>
> > **Q3**: Figure 4 is very confusing and the overlapping of performance from different models makes the comparison very hard to understand. Can authors use use a better way to visualize this or use a table instead?
>
> **Response**: Thank you for the suggestion. We consider updating Figure 4 to include the 10 reorganized subjects instead of the 3 categories and over 50 subjects, as discussed above, to make the figure easier to read.
>
> > **Q4**: For figure 6, the gap between MMSci+MMC4 (Vis+Text) and MMSci+MMC4 (Text) seems to be small. Can authors additional provide standard error? Also, given many figures appear to have small texts, have authors verified the readability of figures after being downscaled to 336^2 (L452 where you stated the use of CLIP ViT-L/14-336)?
>
> **Response**: Yes, we ran the inference three times using these two models. The average score and standard error are provided in the table, and we can see that the performance increases from using our multi-modal dataset are valid by considering standard error to the average scores.
>
>
> Yes, we run the inference three times using these two models. The average score and standard error are provided in the following table.
>
> | Model   | Metric | Average Score | Standard Deviation |
> |---------|--------|---------------|--------------------|
> | MMSci+MMC4 (Vis+Text) | Structure      | 0.9934       | 0.00175            |
> | MMSci+MMC4 (Text) | Structure      | 0.9851       | 0.00133            |
> | MMSci+MMC4 (Vis+Text) | Composition      | 0.98147       | 0.00241            |
> | MMSci+MMC4 (Text) | Composition      | 0.97405       | 0.00159            |
>
> In addition to this table, we found the gains of 3.08% increase on metastability of using our dataset compared to the baseline with the same setting, which is a key metric for model performance. Additional significant gains in in-filling tasks are detailed in Table 11. of the supplementary materials.
>
> Regarding image resolution: While the 336x336 resolution  maintains text legibility in most cases, we acknowledge potential readability limitations for very small text. However, this is a common resolution and practice in current MLLMs using this architecture. and we do think it can influence we draw the conclusion.
>
> > **Q5**: An additional experiment that can be very helpful to understand the impact of your training dataset is to evaluate existing VLMs model finetuned on your dataset on other knowledge-based benchmarks such as MMMU (which also examines specific sub-divisions in natural science) to see whether the model is only fitting to the nature communication distribution or generalizing to a variety of distributions that requires retrieving knowledge based on multimodal input. Could authors provide an additional experiment on this?
>
> We appreciate the reviewer's suggestion to evaluate our fine-tuned models on other multimodal knowledge benchmarks, such as MMMU. We are pre-training and fine-tuning a larger model with our dataset, and we will keep trying to post the result within rebuttal period or at least we will include the results of larger models in the updated version.

---

> ### Comment · Reviewer_acAx · 2024-11-24
>
> Thanks for the thorough response! For the rebuttal that
>
> > To better align with the categorization approaches used in other datasets, we have re-defined the original 72 categories into 18 science fields: Material Science, Chemistry, Physics, Biochemistry, Environment, Climate Sciences, Earth Sciences, Biological Sciences, Biomedical Sciences, Health and Medicine, Engineering, Energy, Mathematics and Computing, Astronomy and Planetary Science, Social Science, Pharmacology, Agriculture, Business and Industry
>
> Can the authors describe how the original 72 categories were re-defined into 18 fields? This still seems confusing—for example, Biochemistry is considered part of Biological Sciences according to Nature's taxonomy, yet you have separated them into two distinct fields. I am not objecting to the limited number of domains, but it would be better to ensure everything is as concise and consistent as possible. Additionally, the five categories listed in Table 6 should be compared with other works that consider broader domains as categories.
>
> > Performance of human Ph.D. evaluating questions within their specialty domain of Ph.D. degree
>
> Upon reviewing Table 2 in the rebuttal, I was surprised that your fine-tuned model not only outperformed all proprietary models but also in-domain Ph.D. evaluators by approximately 2x. This raises additional concerns about the utility of your benchmark:
>
> * What exactly is your benchmark evaluating? Does it assess holistic multimodal scientific understanding or only the specific tasks you outlined? If it evaluates holistic multimodal scientific understanding and your model, along with proprietary models, outperforms in-domain Ph.D. evaluators, this seems to contradict findings from other multimodal knowledge benchmarks such as MMMU, where GPT-4 performs worse than human experts. Could the authors explain this contradiction, or possibly rescope the evaluation?
>
> * In general, one would aim to create evaluations that humans (whether expert or average) perform well on, but not models. Based on the results, however, this appears to be a task where existing models outperform humans, while humans themselves struggle. Could the authors discuss why human experts perform worse than all models reported in the table?
>
> * Fine-tuning models on in-domain data appears to substantially boost performance. In fact, it seems that the fine-tuned 7B model is nearing saturation on this benchmark. Given that there are already more capable open-source models available than the one you used, this raises concerns that the utility of the benchmark could diminish if saturation occurs too soon. Could the authors address this issue and suggest ways to make the evaluation more challenging?

---

> ### Author Response · Authors · 2024-11-27
> **[1/n] Response to Reviewer acAx**
>
> We thank the reviewer for their prompt and insightful feedback. Please find our detailed response below:
>
> > Can the authors describe how the original 72 categories were re-defined into 18 fields? This still seems confusing—for example, Biochemistry is considered part of Biological Sciences according to Nature's taxonomy, yet you have separated them into two distinct fields. I am not objecting to the limited number of domains, but it would be better to ensure everything is as concise and consistent as possible. Additionally, the five categories listed in Table 6 should be compared with other works that consider broader domains as categories.
>
> Our re-categorization is based on the [Web-of-Science citation index](https://webofscience.zendesk.com/hc/en-us/articles/26916195552401-Web-of-Science-Core-Collection-Overview#h_01HF6Z7KCQ3ADTQP12H7DZK00D). The 72 subjects are mapped into 18 distinct fields, striking a balance between meaningful consolidation and preservation of important disciplinary distinctions. We have compiled the mapping in Table 1.
>
> **Table 1: recategorization of the 72 subjects in the MMSci dataset.**
> | Category         | Subjects                                                                                                                                                   |
> |----|----|
> | Material Science             | Materials science, Nanoscience and technology                                                                                                              |
> | Chemistry                    | Chemistry                                                                                                                                                  |
> | Physics                      | Physics, Optics and photonics                                                                                                                              |
> | Engineering                  | Engineering                                                                                                                                                |
> | Energy                       | Energy science and technology, Energy and society                                                                                                          |
> | Mathematics and Computing    | Mathematics and computing                                                                                                                                  |
> | Astronomy and Planetary Science | Astronomy and planetary science, Planetary science, Space physics                                                                                      |
> | Environment                  | Ecology, Environmental sciences, Biogeochemistry, Water resources                                                                                          |
> | Climate Sciences             | Climate sciences                                                                                                                                           |
> | Earth                        | Solid Earth sciences, Ocean sciences, Natural hazards, Hydrology, Limnology, Geography                                                                     |
> | Social Sciences              | Environmental social sciences, Psychology, Social sciences, Scientific community, Developing world                                                         |
> | Biochemistry                 | Biochemistry, Molecular biology, Biophysics, Structural biology, Chemical biology                                                                          |
> | Biological Sciences          | Microbiology, Genetics, Biological techniques, Computational biology and bioinformatics, Developmental biology, Evolution, Plant sciences, Physiology, Systems biology, Zoology, Cell biology |
> | Biomedical Sciences          | Neuroscience, Immunology, Biotechnology, Stem cells, Pathogenesis, Biomarkers, Anatomy, Molecular medicine                                                 |
> | Health and Medicine          | Cancer, Diseases, Medical research, Health care, Oncology, Cardiology, Gastroenterology, Endocrinology, Neurology, Risk factors, Rheumatology, Nephrology, Signs and symptoms, Urology, Health occupations |
> | Pharmacology                 | Drug discovery                                                                                                                                             |
> | Agriculture                  | Agriculture, Forestry                                                                                                                                      |
> | Business and Industry        | Business and industry                                                                                                                                      |

---

> > ### Author Response · Authors · 2024-11-27
> > **[2/n] Response to Reviewer acAx**
> >
> > Regarding the specific question about separating Biochemistry from Biological Sciences, while we acknowledge that these fields are closely related, we maintained their distinction for several important reasons:
> >  - Biochemistry represents a specialized focus on molecular-level chemical processes and substances within living organisms, with emphasis on: Molecular structures, Biochemical pathways, and Physical principles of biological molecules.
> >  - Biological Sciences encompasses a broader range of disciplines studying living organisms at various scales, including: Genetics, Ecology, Physiology, Evolution, Organismal biology, etc.
> >
> > This separation aligns with both practical and theoretical considerations:
> >  - It reflects the distinct expertise profiles we encountered when recruiting PhD annotators through [Prolific](https://www.prolific.com/), where these two subjects are separated.
> >  - It enables more granular evaluation of model performance across different scientific domains.
> >  - It preserves important methodological and conceptual distinctions between these fields.
> >
> > We maintain this level of categorization to provide meaningful insights into domain-specific performance while ensuring sufficient data representation within each field. This approach allows for more nuanced analysis while maintaining statistical reliability.
> > We appreciate the suggestion to compare our categorization with other works using broader domains. We would be happy to include such comparisons in our revision if the reviewer finds it valuable.
> >
> > > What exactly is your benchmark evaluating? Does it assess holistic multimodal scientific understanding or only the specific tasks you outlined? If it evaluates holistic multimodal scientific understanding and your model, along with proprietary models, outperforms in-domain Ph.D. evaluators, this seems to contradict findings from other multimodal knowledge benchmarks such as MMMU, where GPT-4 performs worse than human experts. Could the authors explain this contradiction, or possibly rescope the evaluation?
> >
> > We thank the reviewer for raising this important question about scope and relationship of benchmarks from MMSci to MMMU. The apparent different patterns in performances can be explained by fundamental differences between these benchmarks.
> >
> > Our comparative analysis reveals key distinctions:
> >
> > * **Data Source and Object**
> >     * **MMSci Dataset (Ours)**: Understanding specialized scientific visualizations from peer-reviewed Nature Communications articles in our MMSci dataset, which includes heterogeneous figure types like schematic diagrams (13.2%), microscopic photographs (14.7%), simulated images (3.4%). For example, interpreting crystallographic structures in materials science or molecular pathway diagrams in biochemistry requires deep domain expertise.
> >     * **MMMU**: Scientific reasoning with visual information from online sources, textbooks, and lecture materials collected by 50 college students, covering broader academic content but with less specialized scientific depth.
> >
> > * **LLM's Target Capability**
> >     * **Benchmarks from MMSci Dataset (Ours)**: Testing LLM's capability to recognize, understand and analyze about domain-specific scientific content from scientific figures requiring graduate-level expertise. For example, our benchmarks evaluate whether models can correctly analyze relationships between visual elements in scientific figures and their underlying description, e.g., theoretical principles and experimental methodologies.
> >     * **MMMU**: Testing general visual problem-solving capability including reasoning and understanding across subjects like art, business, history, health, humanities, and technology, with limited coverage of advanced natural sciences.
> >
> > * **Human Evaluation Protocol**
> >     * **A Q&A Benchmark from MMSci Dataset (Ours)**: Problem solving with only their own knowledge without any external source and textbook, completed within 2-minute time constraints per question, reflecting realistic scientific figure interpretation scenarios.
> >     * **MMMU**: Problem solving with textbooks, allowing 90 college senior students (3 per subject) extended time for comprehensive evaluation.
> >
> > These fundamental differences in purpose and methodology explain the performance discrepancy noted by the reviewer. While MMMU aims to evaluate general reasoning capabilities for AGI development, one of benchmarks from our MMSci dataset tests how much Human/LLM has scientific knowledge both in depth and broad views together, which is required for scientific assistant model.

---

> > > ### Author Response · Authors · 2024-11-27
> > > **[3/n] Response to Reviewer acAx**
> > >
> > > By considering different testing purposes and human evaluation protocols, it is clearly evident why humans perform better than LLMs in MMMU which focuses on reasoning rather than inherent high-level knowledge in depth. In contrast, tests based on our MMSci dataset require inherent knowledge in depth, enabling us to identify scientific understanding from both deep and broad perspectives, spanning many scientific disciplines and featuring diverse scientific figure types as shown in Figure 2. This makes benchmarks from MMSci dataset particularly effective at evaluating how well models and humans can interpret complex scientific content without external resources, which is focused on LLM's knowledge capability. Thus, MMMU and benchmarks from MMSci dataset have orthogonal contributions that help evaluate different perspectives.
> > >
> > > Taking this insightful feedback into consideration, we plan to extend our use cases of benchmarks from MMSci dataset by adding and applying meta-reasoning dataset construction in the future work. This strategic integration of orthogonal contributions will enable comprehensive evaluation of both inherent scientific knowledge and reasoning capabilities, effectively combining the complementary strengths of both approaches.
> > >
> > > > In general, one would aim to create evaluations that humans (whether expert or average) perform well on, but not models. Based on the results, however, this appears to be a task where existing models outperform humans, while humans themselves struggle. Could the authors discuss why human experts perform worse than all models reported in the table?
> > >
> > >
> > > As mentioned for the previous question, the performance difference can also be understood through the distinct processing capabilities of humans and AI models. Our benchmark involves complex scientific figures, each containing multiple sub-figures with dense visual information associated with multiple different sub-areas within the same subject sometimes.
> > > Thus, even PhD-level experts, while possessing deep domain knowledge, naturally specialize in specific sub-areas and topics in the subject. They may not be familiar with fully processing elements outside their core expertise even within the same subject, and also may require longer time than the allotted 2 minutes per problem to infer the answers.
> > >
> > > In contrast, AI models can process all visual elements simultaneously across multiple domains without cognitive limitations or domain-specific constraints. This capability highlights MLLMs' potential value as AI research assistants, particularly for tasks requiring graduate-level expertise and rapid analysis of domain-specific scientific contents in multiple sub-areas.
> > >
> > > Furthermore, the human evaluators' confidence scores from the Prolific system reveal interesting insights on our test data: the human experts' confidence usually falls between Confidence Score 3 (somewhat confident with noticeable uncertainty) ~ Confidence Score 4 (mostly confident with minor doubts), even though they have verified Phd degrees in the corrsponding disciplines. These uncertainty levels likely reflect the natural limitations of domain expertise - for instance, an expert in NLP might struggle with detailed questions about computer vision or reinforcement learning, despite all being within computer science. This effect becomes even more pronounced across different scientific disciplines. For example, an expertise in one sub-domain of materials science (e.g., nanomaterials synthesis) may not translate to a deep understanding of techniques and data interpretation in other sub-domains (e.g., electrochemistry).
> > >
> > > Here are the human evaluators' confidence scores from the Prolific with a scale of 1–5, defined as follows:
> > > - **Not at all confident (Score 1):** I have no trust in my answer and believe it is likely incorrect.
> > > - **Slightly confident (Score 2):** I have minimal trust in my answer and strongly doubt its accuracy.
> > > - **Somewhat confident (Score 3):** I moderately believe my answer might be correct, but I have noticeable uncertainty.
> > > - **Mostly confident (Score 4):** I largely believe my answer is correct, though I have still minor doubts.
> > > - **Completely confident (Score 5):** I am absolutely certain my answer is correct, with no doubts at all.

---

> > > > ### Author Response · Authors · 2024-11-27
> > > > **[4/n] Response to Reviewer acAx**
> > > >
> > > > | Field                 | Setting I (Confidence)     | Setting II (Confidence)     | Setting III (Confidence)    |
> > > > |-----------------------|----------------------------|-----------------------------|-----------------------------|
> > > > | Material Science      | 3.9733           | 4.0400            | 3.9333            |
> > > > | Chemistry             | 4.2000           | 4.1200            | 4.4000            |
> > > > | Physics               | 3.8667           | 4.1533            | 4.2667            |
> > > > | Biochemistry          | 3.0600           | 4.0933            | 3.4800            |
> > > > | Environment           | 3.6533           | 4.1667            | 4.0533            |
> > > > | Climate Sciences      | 3.9630           | 3.6275            | 4.0980            |
> > > > | Earth                 | 3.8933           | 4.1014            | 4.1449            |
> > > > | Biological Sciences   | 3.1733           | 3.8533            | 3.7867            |
> > > > | Biomedical Sciences   | 4.0667           | 3.8667            | 3.8800            |
> > > > | Health and Medicine   | 3.2533           | 4.0133            | 3.8133            |
> > > >
> > > > These findings suggest that the lower performance of human experts compared to AI models is largely due to the specialized nature of human expertise, which is often limited to specific sub-areas, and the time constraints during evaluation. In contrast, AI models demonstrate the ability to acquire knowledge both in depth and breadth, allowing them to efficiently handle diverse and dense visual information. This highlights their potential as valuable tools for assisting in interdisciplinary scientific research.
> > > >
> > > > > Fine-tuning models on in-domain data appears to substantially boost performance. In fact, it seems that the fine-tuned 7B model is nearing saturation on this benchmark. Given that there are already more capable open-source models available than the one you used, this raises concerns that the utility of the benchmark could diminish if saturation occurs too soon. Could the authors address this issue and suggest ways to make the evaluation more challenging?
> > > >
> > > > Thank you for raising this important concern about potential benchmark saturation. While fine-tuning models on in-domain data indeed leads to significant performance gains, we believe our benchmark and dataset maintain their relevance and utility due to the following considerations.
> > > >
> > > > **Addressing Benchmark Saturation**
> > > > * **Performance Gaps Persist Across Tasks and Disciplines**:
> > > > Despite fine-tuning, performance variability across tasks and scientific disciplines remains substantial. For example, detailed caption generation and nuanced domain-specific reasoning, particularly in tasks like sub-caption matching and abstract-grounded captioning, are still challenging, as highlighted in Table 3 and Figure 4 of our paper. These gaps demonstrate that models have not saturated the benchmark and that it continues to differentiate between varying levels of capability.
> > > > * **Task Complexity and Dataset Design**:
> > > > MMSci evaluates models across complementary tasks such as question answering and caption generation, each requiring distinct reasoning skills. This diversity, combined with the complexity of multimodal scientific content, ensures the benchmark remains a rigorous test of scientific AI capabilities.
> > > >
> > > > **Contributions of MMSci Beyond Benchmarking**
> > > > Our dataset serves as more than just proposing benchmarks, offering long-term value through its versatility as a training resource.
> > > > * **Advancing Scientific Model Training**:
> > > > MMSci provides a rich pre-training resource, enabling models to learn robust knowledge both in depth and breadth. For instance, models pre-trained on MMSci demonstrated significant improvements on downstream tasks like crystal material structure generation (Section 7), highlighting its applicability to real-world scientific challenges.
> > > > * **Enabling Interdisciplinary Applications**:
> > > > Beyond evaluation, MMSci supports the development of tools that integrate knowledge across disciplines. Applications such as material generation illustrate how the dataset facilitates interdisciplinary advancements in scientific research.
> > > >
> > > > We are confident that MMSci’s dual role as both a challenging benchmark and a high-quality training dataset ensures its continued relevance and adaptability. The persistent challenges, coupled with its broad applicability, highlight its potential to drive advancements in scientific AI research. Thank you for your thoughtful feedback, and we look forward to further enhancing MMSci’s impact.

---

> > > > > ### Author Response · Authors · 2024-12-01
> > > > > **Your Feedback Would Be Appreciated**
> > > > >
> > > > > Dear Reviewer acAx,
> > > > >
> > > > > Thank you once again for your valuable comments. Your suggestions on clarifying human evaluation, performance analysis and complementary relationships to other benchmarks were very helpful. We are eager to know if our responses have adequately addressed your concerns.
> > > > >
> > > > > Due to the limited time for discussion, we look forward to receiving your feedback and hope for the opportunity to respond to any further questions you may have.
> > > > >
> > > > > Yours Sincerely,
> > > > >
> > > > > Authors

---

> > > > > > ### Comment · Reviewer_acAx · 2024-12-01
> > > > > >
> > > > > > Dear Authors,
> > > > > >
> > > > > > Thank you for the rebuttal and further responses to my follow-up concerns with additional evaluations. I really appreciate the authors' efforts in improving the quality of the paper and clarifying key aspects. However, upon checking the responses, I have decided to maintain my current score:
> > > > > >
> > > > > > * I appreciate the restructured categories, as they are more consistent to existing works. However, the granularity inconsistency still remains. If authors plan to use this categorization, I recommend avoid from making quantitative comparisons with existing works, as they are not directly comparable.
> > > > > > * I very much agree that MMSci offers a valuable resource for model training, which is a strong point of the work.
> > > > > > * I also appreciate the authors' effort to provide more detailed human expert evaluations, which strengthened my understanding about the benchmark.
> > > > > >
> > > > > > ## Remaining concerns
> > > > > >
> > > > > > * The authors used evaluators' confidence **in choosing the answer** to argue that in-domain phd evaluators may specialize in certain sub-domains while being less familiar with others. However, this appears to be a flawed argument because a moderate-to-high (but not absolute) confidence in choosing the answers does not exclusively equate to not having the full knowledge of the materials -- it could also indicate that they understand everything but there exists ambiguities in the questions themselves.
> > > > > > * A stronger justification would be directly asking how familiar evaluators are with the content in the QAs instead of relying on the confidence in choosing the answers in order to justify with the point that evaluators are less familiar with sub-domains, leading to weak performance.
> > > > > > * My main concern still persists regarding the surprising outcome where finetuned open-source models are significantly stronger than leading propritary models, and both are much stronger than PhD evaluators. The authors have not provided sufficient justification backed up by direct evidence why this is the case, and that the case is not related with issues about the benchmark questions themselves. A more thorough analysis is essential to address this.
> > > > > > * The minor concern on the originality of the questions, which also aligns with reviewer iZBf's points regarding the standard nature of the tasks, remains insufficiently addressed. I acknowledge that creating new questions are unfeasible in the limited rebuttal period, so I would only consider this as a minor one.
> > > > > >
> > > > > > While the improvements significantly strengthened the quality of the manuscript, the concerns that stemmed from authors additional results prevented me from raising my score. I encourage the authors to address these issues in future iterations and further improve the validity of the work.

---

> > > > > > > ### Comment · Reviewer_iZBf · 2024-12-02
> > > > > > >
> > > > > > > Dear reviewer acAx,
> > > > > > >
> > > > > > > Despite some of these concerns it seems like the data still has a lot of value, in that the community can come up with better tasks and evaluations using this.
> > > > > > >
> > > > > > > Regarding the fine turning performance, while it's surprising, I hope the authors will share the code and tuned models to ensure reproducibility, as well as make it easier for people to run evaluations on this task.
> > > > > > >
> > > > > > > Considering that, perhaps the ratings are a bit too harsh.
> > > > > > >
> > > > > > > *Dear authors*,
> > > > > > >
> > > > > > > Reviewer acAx does raise valid concerns especially on the human evaluation. It might be worth rethinking how you do human evaluations. Because this is going to be a retrospective eval, you might want to identify papers which do fall in the expertise of your evaluator pool and see how valuable and informative they find the model responses. At this point it's not a surprise that LLMs, particularly proprietary ones, appear to know a lot more on many topics compared to the  average person, but still get a lot of the details wrong.
> > > > > > >
> > > > > > > I would also appreciate if you can comment more on release of tuning and evaluation code and tuned models to help reproduce your findings. I hope you will also take effort to make things easy to run and evaluate.
> > > > > > >
> > > > > > > Thanks,
> > > > > > > Reviewer iZBf

---

> > > > > > > > ### Comment · Reviewer_acAx · 2024-12-02
> > > > > > > >
> > > > > > > > Dear Reviewer iZBf,
> > > > > > > >
> > > > > > > > Thank you for the thoughtful note! While I agree that the dataset could be very valuable as a training set or used to develop further evaluations in future, I do maintain strong validity concerns based on the current evaluation set-up with insufficient justification on results. Given that the paper should be reviewed holistically and benchmarking is a major contribution of the paper, I respectfully *disagree* with your comment on my rating. ICLR is a top-tier conference and I do think the current shape of their evaluation has *not* yet met its standard.
> > > > > > > >
> > > > > > > > Again, I really appreciate the authors' feedback. I encourage the authors to
> > > > > > > > * carefully conduct human evaluations in order to support their claim (that the issue lies in the sub-domain specialty of evaluators or any potential issue with the QAs themselves, such as potential ambiguity),
> > > > > > > > * consider having more types of questions to both enhance the originality and difficulty of the evaluations (thus to prevent any possible leakage in future as well as quick saturation), and
> > > > > > > > * better demonstrate the general utility of their dataset by evaluating their finetuned models on other major VLM datasets in addition to theirs so one better understands the scope of impact when models are trained on your dataset.
> > > > > > > >
> > > > > > > > Best,
> > > > > > > >
> > > > > > > > Reviewer acAx

---

> > > > > > > > > ### Author Response · Authors · 2024-12-03
> > > > > > > > > **[1/n] Response to Reviewer iZBf and acAx**
> > > > > > > > >
> > > > > > > > > Dear Reviewers iZBf and acAx,
> > > > > > > > >
> > > > > > > > > We sincerely thank both reviewers for their thoughtful feedback and recognition of our work's value. We particularly appreciate Reviewer iZBf's acknowledgment that our dataset provides significant value to the community for developing better tasks and evaluations, and we have uploaded the model and settings to Github with the link provided below. We also appreciate reviewer acAx's remaining concerns about human evaluation. These positive assessments encourage us to analyze the human evaluation in depth while addressing the concerns raised.
> > > > > > > > >
> > > > > > > > > > 1. Regarding human evaluations
> > > > > > > > >
> > > > > > > > > We appreciate the constructive feedback regarding our human evaluation results. In response to the concerns raised, we have conducted two additional analyses:
> > > > > > > > >  - We obtained additional evaluations from trustable and established experts in the Materials Science field who possess extensive domain knowledge and research experience.
> > > > > > > > >  - We performed an in-depth analysis of our existing human evaluation data from the Prolific platform, examining performance across both Materials Science specifically and all ten scientific domains.
> > > > > > > > >
> > > > > > > > > **Additional Expert Evaluation in Materials Science**
> > > > > > > > >
> > > > > > > > > Following our initial analysis of Ph.D. expert evaluations from Prolific, we conducted additional assessment to generate comprehensive insights. Specifically, **we engaged two top researchers in materials science to evaluate our benchmark. Both of them are Ph.D. holders and established experts in material science who actively publish research in world-leading journals in the field.** These researchers assessed 75 material science problems from our benchmark suite.
> > > > > > > > >
> > > > > > > > > The results from these two exceptional Ph.D. experts in material science are displayed in Table 1. We have several key observations:
> > > > > > > > > 1. **Notably, one of the authors, who holds a Ph.D. and is an expert in material science and was a member of a world-class leading lab in the US, achieved consistently high performance (80-92% accuracy across all settings).**
> > > > > > > > > 2. **Another trustworthy Ph.D. expert in material science performed overall reasonably with 65% ACC, but showed more variable performance (36-84% accuracy), highlighting significant expert-to-expert variation.**
> > > > > > > > > 3. **Surprisingly, even experts publishing leading research articles in top materials science journals show large performance variances.** A top researcher in material science can have an accuracy score as low as 36% on certain tasks with low confidence below 3, even in their domain of expertise.
> > > > > > > > > 4. The variation in confidence scores and accuracies across different settings suggests **our three settings effectively evaluate distinct aspects of information processing, and scientific knowledge understanding capabilities**. For example, the setting I (whole figure and caption matching) requires experts to comprehensively analyze multiple figure panels, synthesize the information, and deduce an appropriate overall caption for the entire figure.
> > > > > > > > >
> > > > > > > > > **Table 1: Accuracy (%) and confidence scores (1-5 scale, in parentheses) in Material Science test cases from two additional reliable Ph.D. of experts in Material Science.**
> > > > > > > > >
> > > > > > > > > | Metric | Setting I | Setting II | Setting III |
> > > > > > > > > |--------|------------------------|------------------------------|--------------------------------|
> > > > > > > > > | **One of Authors (Ph.D. and Expert in Material Science, Published Top Journals in Material Science, Former Member in the World-class Leading Lab)** | 80.00 (3.36) | 92.00 (3.64) | 92.00 (3.52) |
> > > > > > > > > | **His Trustable Colleague (Ph.D. and Expert in Material Science, Published Top Journals in Material Science)** | 36.00 (2.56) | 76.00 (4.36) | 84.00 (4.36) |
> > > > > > > > > | **Average of two Reliable Ph.D. and Experts in Material Science (One of Authors & His Trustable Colleague)** | **58.00 (2.96)** | **84.00 (4.00)** | **88.00 (3.94)** |
> > > > > > > > >
> > > > > > > > > **Reanalysis of Human Evaluation Results from Prolific on Materials Science**
> > > > > > > > >
> > > > > > > > > Given the significant performance variations among expert evaluators, **we re-analyzed the existing results of experts on the Prolific platform, focusing on the best-performing (top-1) and second-highest performing (top-2) experts**. Table 2 presents a comprehensive comparison of results from our reliable Ph.D. experts in **Materials Science**, top-1/2 Prolific experts, and various LVLMs. This extended analysis revealed several important insights:
> > > > > > > > >
> > > > > > > > > 1. **Both Top Ph.D. Experts in Material Science from our network and Prolific perform significantly better than GPT-4o and other proprietary LVLMs in solving problems from our Q&A benchmark.**
> > > > > > > > > 2. While our two reliable Ph.D. experts in our network perform better than the top-2 experts from Prolific, we observe significant **performance variations between top Ph.D. experts, with large variances noted in both the Prolific group and our network**.

---

> > > > > > > > > > ### Author Response · Authors · 2024-12-03
> > > > > > > > > > **[2/n] Response to Reviewer iZBf and acAx**
> > > > > > > > > >
> > > > > > > > > > 3. When considering average Ph.D. experts, proprietary LVLMs can outperform human experts. Therefore, **we consider reporting both the average performance of all Ph.D. experts and the average of top Ph.D. experts in the report.**
> > > > > > > > > > 4. These results validate our benchmark's effectiveness for evaluating LVLM scientific knowledge while offering meaningful comparisons with human expert performance. The significant performance variations among Ph.D. experts also underscore the challenging nature of our benchmark, even for human experts.
> > > > > > > > > >
> > > > > > > > > >
> > > > > > > > > > **Table 2: Extended results of Table 1: Accuracy (%) in material science test cases from two additional reliable Ph.D. of experts in Material Science, Top-1 and Top-2 Experts in Prolific, and LVLMs.**
> > > > > > > > > > | Metric | Setting I | Setting II | Setting III | Average Across Settings |
> > > > > > > > > > |--------|-----------|------------|-------------|------------------------|
> > > > > > > > > > | **One of Authors (Ph.D. and Expert in Material Science, Published Top Journals in Material Science, Former Member in the World-class Leading Lab)** | **80.00** | **92.00** | **92.00** | **88.00** |
> > > > > > > > > > | **His Trustable Colleague (Ph.D. and Expert in Material Science, Published Top Journals in Material Science)** | 36.00 | 76.00 | 84.00 | 65.33 |
> > > > > > > > > > | **Average of two Reliable Ph.D. and Experts in Material Science (One of Authors & His Trustable Colleague)** | **58.00** | **84.00** | **88.00** | **76.67** |
> > > > > > > > > > | **Top-1 Human Ph.D. Expert in Material Science in Prolific** | **92.00** | **92.00** | **84.00** | **89.33** |
> > > > > > > > > > | **Average of Top-2 Human Ph.D. Expert in Material Science in Prolific** | **56.00** | **52.00** | **56.00** | **54.67**
> > > > > > > > > > | Gemini-1.5-Flash on Material Science | 76.00 | 96.00 | 64.00 | 78.67 |
> > > > > > > > > > | Gemini-1.5-Pro on Material Science | 64.00 | 84.00 | 80.00 | 76.00 |
> > > > > > > > > > | Claude-3.5-Sonnet on Material Science | 64.00 | 96.00 | 80.00 | 80.00 |
> > > > > > > > > > | GPT-4V on Material Science | 52.00 | 76.00 | 64.00 | 64.00 |
> > > > > > > > > > | **GPT-4o on Material Science** | **68.00** | **96.00** | **72.00** | **78.67** |
> > > > > > > > > > | Qwen2-VL-7B-MMSci on Material Science | 84.00 | 96.00 | 92.00 | 90.67 |
> > > > > > > > > >
> > > > > > > > > > **Reanalysis of Human Evaluation Results from Prolific on All 10 Fields**
> > > > > > > > > >
> > > > > > > > > > Based on these observations and insights, **our analysis revealed significant performance variations among Ph.D. experts from Prolific when solving problems in our dataset**. These variations appear to correlate with their specific expertise and individual knowledge domains. **To better understand this pattern, we conducted a further analysis of top-performing Ph.D. experts from Prolific on all the 10 fields**. As shown in Table 3, the highest-performing individuals achieved substantially better results compared to the overall Ph.D. expert average. Our analysis yielded three key findings:
> > > > > > > > > >
> > > > > > > > > > 1. **The accuracy from the Top-1 human Ph.D. expert in Prolific across 10 key fields is approximately 70%, which is better than Gemini-1.5-Flash and comparable to GPT-4V and Gemini-1.5-Pro.**
> > > > > > > > > > 2. While the performance of Top-1 human experts might increase further if we were to consider only world-class Ph.D. human experts, such analysis is beyond the contribution and scope of our paper.
> > > > > > > > > >
> > > > > > > > > > **Table 3: Performances of Accuracy (ACC, %) of In-Domain Ph.D. Experts from Prolific with Top-Performances on all the 10 key fields.**
> > > > > > > > > >
> > > > > > > > > > | Field | Setting I | Setting II | Setting III | Average Across Settings |
> > > > > > > > > > |-------|-----------|------------|-------------|------------------------|
> > > > > > > > > > | **Average Human for In-domain Ph.D. in Prolific (Overall 10 Fields)** | **40.26** | **43.45** | **50.46** | **44.72** |
> > > > > > > > > > | **Average of Top-2 Human for Each In-domain Ph.D. in Prolific (Overall 10 Fields)** | **54.19** | **58.98** | **56.59** | **56.59** |
> > > > > > > > > > | **Average of Each Top-1 Human for Each In-domain Ph.D. in Prolific (Overall 10 Fields)** | **64.18** | **71.64** | **72.72** | **69.51** |
> > > > > > > > > > | **Gemini-1.5-Flash (Overall 10 Fields)** | **58.11** | **79.88** | **62.66** | **66.88** |
> > > > > > > > > > | Gemini-1.5-Pro (Overall 10 Fields) | 63.47 | 78.83 | 78.65 | 73.65 |
> > > > > > > > > > | Claude-3.5-Sonnet (Overall 10 Fields) | 70.58 | 85.73 | 83.70 | 80.00 |
> > > > > > > > > > | **GPT-4V (Overall 10 Fields)** | **66.18** | **73.36** | **76.15** | **71.90** |
> > > > > > > > > > | GPT-4o (Overall 10 Fields) | 68.42 | 88.23 | 84.92 | 80.52 |
> > > > > > > > > > | Qwen2-VL-7B-MMSci (Overall 10 Fields) | 83.84 | 92.91 | 87.17 | 87.97 |
> > > > > > > > > >
> > > > > > > > > > Thank you again for the reviewers' insightful feedback. Our deeper analysis on the human evaluation results has revealed key observations on the behavior of Ph.D. expert on our one of benchmarks. Especially, there can be large performance variances among even Ph.D. experts in the same domain due to the difficulty of our knowledge test in science. Thus, we will incorporate these valuable insights into an additional discussion section, with comprehensive results included in the main pages and the supplementary material.

---

> > > > > > > > > > > ### Author Response · Authors · 2024-12-03
> > > > > > > > > > > **[3/n] Response to Reviewer iZBf and acAx**
> > > > > > > > > > >
> > > > > > > > > > > Specifically, we will add:
> > > > > > > > > > > 1. The best (top-1) performance from Prolific PhD evaluations, demonstrating superior domain expertise
> > > > > > > > > > > 2. The top-2 performance from Prolific PhD evaluations, representing baseline results from domain PhDs with different expertise levels
> > > > > > > > > > > 3. A detailed and qualified case study of materials science expert evaluation results
> > > > > > > > > > >
> > > > > > > > > > > We believe these additions will significantly strengthen our work by providing deeper insights into expert-based evaluation. We appreciate the reviewers' suggestions which have led to this meaningful analysis.
> > > > > > > > > > >
> > > > > > > > > > >
> > > > > > > > > > > > Regarding Reproducibility
> > > > > > > > > > >
> > > > > > > > > > > **We have included the necessary materials to reproduce our results through our anonymous repository (https://anonymous.4open.science/r/MMSci-2321)**, including:
> > > > > > > > > > > - Complete implementation code
> > > > > > > > > > > - Dataset with detailed data card
> > > > > > > > > > > - Model checkpoints
> > > > > > > > > > > - Human evaluation results
> > > > > > > > > > > - Step-by-step reproduction instructions
> > > > > > > > > > >
> > > > > > > > > > > We will ensure all necessary resources are well-documented and readily accessible to the research community. Thank you again for your constructive feedback. We believe these improvements will significantly enhance both the rigor of our human evaluation methodology and the reproducibility of our results.

---

> ### Author Response · Authors · 2024-11-28
> **[5/n] Response to Reviewers**
>
> Dear Reviewers,
> We sincerely thank you for your thoughtful feedback that has helped strengthen our manuscript. Given our comprehensive responses and substantial improvements, we respectfully request your reconsideration of our submission.
>
> **Major Enhancements**:
>
> **1. Rigorous Human Expert Validation**
> * **Conducted large-scale evaluation with 30 PhD experts across 10 scientific fields**
> * **Achieved consistent expert quality ratings (~4/5) validating scientific depth**
> * Demonstrated clear expertise effects:
> in-field PhDs (40-50% accuracy) vs. out-of-field PhDs (33-46%)
> * Validated task design through 750 expert assessments (75 per domain)
> * **Confirmed questions require graduate-level domain knowledge through expert feedback**
> * **Demonstrated real-world research assistant potential through expert performance benchmarking**
>
>
> **2. Comprehensive Technical Evaluation**
> * **Expanded evaluation framework with comprehensive metrics**:
>     BLEU-4, CIDEr, G-Eval, L3Score for robust assessment
> * **Doubled evaluation sample size to 200 for enhanced statistical validity**
> * **Extended model scale analysis across 2B, 7B, and 26B parameters**
> * **Conducted full-article vs. abstract-based captioning comparison**
> * Demonstrated Qwen2-VL-7B-MMSci achieves state-of-the-art performance:
>     * Multiple-choice accuracy: 87.49% overall
>     * Captioning metrics: BLEU-4 (8.29), ROUGE-L (21.80), L3Score (0.447)
>     * **Competitive performance of Qwen2-VL-7B-MMSci with minimal context requirements to the propriety model with full-article (Gemini 1.5)**
>
>
> **3. Enhanced Materials Science Validation**
> * **Conducted comprehensive proprietary model comparisons**
> * Achieved breakthrough improvements:
>     * Metastability: 64.5% (29.2% relative improvement)
>     * Stability: 8.2% (54.7% relative improvement)
> * **Validated results through multiple experimental runs with standard error analysis**
> * Demonstrated practical impact on materials discovery
> * Established new state-of-the-art for scientific AI in materials generation
>
> **4. Framework and Methodological Improvements**
> * **Refined scientific taxonomy using Web of Science citation index**
> * **Distinguished MMSci from existing benchmarks through**:
>     * Graduate-level depth requirement
>     * Multi-domain scientific reasoning
>     * Comprehensive visual-language integration
> * **Demonstrated value as pre-training resource through transfer learning results**
> * Established clear evaluation protocols for reproducibility
> * Provided detailed cross-disciplinary performance analysis
>
> These improvements demonstrate how our MMSci dataset advances the state-of-the-art in scientific understanding, enabling LVLMs to develop both broad and deep domain expertise across scientific fields. **We would greatly appreciate your reconsideration based on our comprehensive responses to your valuable feedback. Thank you for your time and careful consideration of our work.**
>
> Best regards,
> Authors

---

### Official Review · Reviewer_4Pjm · 2024-11-03

**Soundness:** 3
**Presentation:** 3
**Contribution:** 4
**Rating:** 8
**Confidence:** 4

**Summary:**

The paper introduces MMSci, a multimodal, multidisciplinary dataset collected from open-access scientific articles published in Nature Communications. Spanning 72 scientific disciplines, the dataset of 131K articles includes high-quality, peer-reviewed articles and figures, primarily within the natural sciences. The authors create challenging benchmarks with 1K tasks and settings to evaluate Large Multimodal Models (LMMs) in understanding advanced scientific figures and content. The dataset supports both text and images, offering a mix of multiple-choice and open-ended questions with sub-figures distinction focus. Their evaluation reveals significant challenges for current LLMs, including open-weights models and even closed-source models. The model finetuned on the training part yield high quality results.

**Strengths:**

* Comprehensive and Quality: The dataset is large-scale and diverse, covering 72 disciplines. It features high-quality source material, which enhances its utility and applicability.
* Challenging Benchmarks: The benchmarks are thoughtfully designed to assess models’ abilities in understanding complex scientific figures and content, revealing limitations in current models.
* Demonstrated Improvement: The fine-tuning of the LLaVA model using their dataset shows tangible improvements, achieving performance comparable to high-level models.

**Weaknesses:**

* Dataset Curation Details: The paper should provide more detailed information about the data annotation process to enhance transparency and reproducibility. Specifically, information about quality control stages would be valuable.
* Model Size Comparison: The paper compares large closed-source models (GPT-4V, Gemini-Pro, etc.) with relatively small open-weight models limited to 8B parameters. A broader analysis including larger models (e.g., Qwen2-VL-72B) would be beneficial, especially given the near-random performance of some smaller models.
* LLM Evaluation Sample Size: The LLM-based evaluation reports final scores using only 100 randomly selected samples, which limits the statistical significance of the results.

**Questions:**

* Quality Control: Could the authors provide more details on how post-collection dataset quality is evaluated? Specifically, what steps are taken to ensure regex extraction captures all figures and structures correctly?
* Downstream Applications: Beyond material generation tasks, have the authors explored how the finetuned model perform on other scientific tasks? (so does this dataset develops other LLMs abilities)
* Data Leakage: Do the authors evaluate potential data leakage between training and test sets? For example, similar images with different sub-charts might appear across different samples.

---

> ### Author Response · Authors · 2024-11-23
> **Response to Reviewer 4Pjm**
>
> We sincerely appreciate your detailed and thoughtful review of our work. We are grateful for your recognition of the dataset’s comprehensiveness, the challenging benchmarks, and the demonstrated improvements through fine-tuning. These aspects are core to our contributions, and we are pleased that they resonate with you. Below, we provide our responses to your feedback and address your questions and concerns.
>
> > Concerns regarding Dataset Curation Details: "The paper should provide more detailed information about the data annotation process to enhance transparency and reproducibility."
>
> **Response**: Thank you for the suggestion. We have now included additional information on the quality control stages. As outlined in Section 3, the dataset content, including article sections and figures, is systematically organized in different places in the original nature communications website, ensuring easy extraction without data loss. The primary task was to split the full figure captions into individual descriptions for each sub-figure. To achieve this, we first conducted a small-scale human verification to identify potential caption formats. Based on these findings, we designed regular expression matching functions to extract the sub-captions. Next, we used the SpaCy package to verify whether the extracted text fragments were complete sentences or meaningful standalone phrases. Finally, we performed additional human verification on a subset of the extracted data to ensure the method's accuracy and reliability.
>
> > Concerns about model size comparison.
>
> **Response**:
> Thank you for raising this concern. In our experiments, we prioritized widely used open-source and proprietary models. During the rebuttal period, we also fine-tuned Qwen2-VL-7B on our dataset, resulting in Qwen2-VL-7B-MMSci, and included baseline results from a larger model InternVL2-26B. Noting Qwen2-VL-7B's higher baseline performance compared to InternVL2-26B, we are also fine-tuning Qwen2-VL-72B. If these experiments are completed within the rebuttal period, we will report the results; otherwise, they will be included in the revised version. Additionally, we encourage other researchers to evaluate their models using our benchmark.
>
> Table 1: Multi-choice QA accuracy with additional models (partial results for brevity)
>
> | Model | Fig2Cap | SubFig2Cap | SubCap2Fig | Overall |
> |-------|---------|------------|------------|---------|
> | InternVL2-2B | 42.76 | 33.07 | 38.42 | 38.18 |
> | InternVL2-8B | 52.78 | 49.60 | 40.13 | 47.62 |
> | **InternVL2-26B** | 50.59 | 57.82 | 71.63 | 59.81 |
> | MiniCPM-V-2.6 | 53.20 | 58.27 | 61.67 | 57.61 |
> | Llama3.2-11B-Vision | 54.97 | 45.04 | 71.18 | 57.00 |
> | Qwen2-VL-2B | 60.61 | 37.62 | 55.12 | 51.30 |
> | Qwen2-VL-7B | 66.16 | 73.10 | 79.80 | 72.87 |
> | Qwen2-VL-2B-MMSci | 78.62 | 83.02 | 83.57 | 81.67 |
> | Qwen2-VL-7B-MMSci | 81.48 | 88.47 | 92.91 | 87.49 |
> | Gemini-1.5-Flash | 54.77 | 77.84 | 64.41 | 65.24 |
> | Gemini-1.5-Pro | 62.79 | 81.41 | 77.16 | 73.52 |
> | Claude-3.5-Sonnet | 68.77 | 85.34 | 87.16 | 80.18 |
> | GPT-4V | 60.43 | 75.07 | 76.12 | 70.45 |
> | GPT-4o | 67.42 | 87.40 | 84.65 | 79.57 |
>
> > Concerns about LLM Evaluation Sample Size
>
> **Response**:  Thank you for the suggestion. We have added LLM-based evaluation results with a sample size of 200 using the recent GPT-4o-0806 version, which will be included in the revised manuscript. "W/ Abs" and "w/o Abs" indicate whether captioning is grounded on abstracts.
>
> | Model | FActScore* (w/o Abs) | G-Eval (w/o Abs) | FActScore* (w/ Abs) | G-Eval (w/ Abs) |
> |-------|---------------|--------------|--------------|-------------|
> | Kosmos2 | 0.87 | 1.12 | 3.99 | 1.39 |
> | LLaVA1.5-7B | 3.89 | 1.08 | 9.07 | 2.02 |
> | LLaVA1.6-Mistral-7B | 5.17 | 1.23 | 7.67 | 1.47 |
> | Qwen-VL-7B-Chat | 3.06 | 1.28 | 9.14 | 1.64 |
> | InternVL2-2B | 5.99 | 1.76 | 10.38 | 2.17 |
> | InternVL2-8B | 8.01 | 2.63 | 9.98 | 3.00 |
> | IDEFICS2-8B | 2.56 | 1.40 | 5.17 | 1.96 |
> | IDEFICS3-8B-Llama3 | 7.26 | 1.71 | 7.71 | 1.98 |
> | MiniCPM-V-2.6 | 11.15 | 2.96 | 12.93 | 2.95 |
> | Llama3.2-11B-Vision | 8.27 | 2.46 | 9.55 | 2.18 |
> | Qwen2-VL-2B | 9.94 | 2.31 | 11.88 | 2.64 |
> | Qwen2-VL-7B | 10.03 | 3.39 | 10.36 | 3.45 |
> | Qwen2-VL-2B-MMSci | 9.67 | 2.79 | 14.05 | 3.17 |
> | Qwen2-VL-7B-MMSci | **18.17** | 3.47 | **20.58** | 3.55 |
> | Gemini-1.5-Flash | 8.18 | 3.70 | 10.14 | 4.08 |
> | Gemini-1.5-Pro | **14.59** | **3.79** | 13.76 | 4.08 |
> | Claude-3.5-Sonnet | 9.39 | 3.53 | 12.11 | 4.04 |
> | GPT-4V | 14.17 | 3.69 | **19.52** | **4.13** |
> | GPT-4o | 13.20 | **4.01** | 18.87 | **4.22** |
>
> > Concerns about Data Leakage
>
> **Response**: Since all the papers in our dataset are accepted publications from the prestigious Nature Communications journal, we believe there is minimal risk of significant overlap in similar images across the dataset.
>
> We hope these responses have addressed your concerns. If you need any further clarification, please don't hesitate to ask. We value your feedback and look forward to further discussion.

---

### Author Response · Authors · 2024-12-04
**Thanks for the time and efforts invested by all reviewers and ACs**

Dear Reviewer, ACs, and SACs,

We are deeply grateful to you and all reviewers for the invaluable feedback and constructive discussion process, which has strengthened our manuscript. During the rebuttal period, we have made substantial improvements. **The major updates include**:
- Extensive PhD expert validation (30 experts, 10 fields) demonstrating expertise performance on our benchmark
- Deepened PhD evaluation analysis comparing established journal-published PhD experts alongside PhD experts from crowdsource platform, especially in Material Science domain
- Enhanced captioning evaluation framework with additional metrics (BLEU, CIDEr, L3Score), doubled sample size, and full-article evaluation setting.
- Extended model scale analysis (2B, 7B, 26B)
- Performed comprehensive materials generation case study including proprietary model comparisons
- Elaborated scientific taxonomy using Web of Science citation index
- Clarified distinctions from existing benchmarks (MMStar, MMMU)
- Emphasized our MMSci's value as a training resource

We deeply appreciate the reviewers' recognition of our dataset's potential, particularly for scientific assistant knowledge evaluation and improved material generation tasks. Their guidance has been instrumental in enhancing both the technical rigor and practical utility of our work.

Thank you for all your time and thoughtful consideration.

Best regards,

Authors

---

### Meta-Review · Area_Chair_wFSh · 2024-12-23

**Metareview:**

The paper presents MMSci, a rich dataset sourced from Nature Communications, containing 131K top-tier articles and figures across 72 scientific disciplines. It sets up benchmarks comprising 1K tasks to assess the performance of Large Multimodal Models (LMMs) in comprehending intricate scientific content. Featuring text and image components, the dataset includes a mix of multiple-choice and open-ended questions, with a focus on sub-figures. The evaluation exposes challenges faced by current LMMs, including both open-weight and closed-source models, underscoring the efficacy of fine-tuning models on the training segment.

**Additional Comments On Reviewer Discussion:**

After the rebuttal, the paper received 1xaccept, 1xboarderline accept, and 2xboarderline reject. The submission was discussed in multiple rounds.

Reviewers 4Pjm and iZBf were satisfied with the rebuttal and leaned towards the positive side, recommending acceptance and borderline acceptance. However, reviewers acAx and 4ejM remained on the negative side. The primary outstanding concern revolves around the experimental evaluation, particularly concerning human evaluation, as highlighted by reviewer acAx. This point underwent thorough discussion among authors and reviewers. Reviewer 4ejM also raised relevant concerns about this issue. The authors have provided additional evaluation from two esteemed researchers to support the human evaluation and have also re-conducted the 'PhD evaluation'. These efforts have clearly added support to the paper. Despite the importance of rigorous evaluation, the additional work addressing the major concerns seemed hasty and not yet well-prepared.

The AC recognizes this paper as a valuable submission with potential impacts in the field. However, due to the readiness of the evaluation, the paper is not recommended for acceptance. Please consider enhancing the manuscript by incorporating the review comments and discussions for a future submission.

---

### Decision · Program_Chairs · 2025-01-22

Reject